# DNA-sensing inflammasomes cause recurrent atherosclerotic stroke

Jiayu Cao[1,22], Stefan Roth[1,22 ✉], Sijia Zhang[1], Anna Kopczak[1,2], Samira Mami[1], Yaw Asare[1], Marios K. Georgakis[1,2,3], Denise Messerer[4], Amit Horn[5], Ruth Shemer[5], Charlene Jacqmarcq[6], Audrey Picot[6], Jack P. Green[7], Christina Schlegl[1], Xinghai Li[8], Lukas Tomas[4], Alexander Dutsch[8], Thomas G. Liman[9], Matthias Endres[9,10,11], Saskia R. Wernsdorf[1], Christina Fürle[1], Olga Carofiglio[1], Jie Zhu[1], David Brough[7], DEMDAS Study Group*, Veit Hornung[12], Martin Dichgans[1,2,13,14], Denis Vivien[6,15], Christian Schulz[4,13,16], Yuval Dor[5], Steffen Tiedt[1], Hendrik B. Sager[8,13], Gerrit M. Grosse[17] & Arthur Liesz[1,2 ✉]

The risk of early recurrent events after stroke remains high despite currently established secondary prevention strategies[1]. Risk is particularly high in patients with atherosclerosis, with more than 10% of patients experiencing early recurrent events[1,2]. However, despite the enormous medical burden of this clinical phenomenon, the underlying mechanisms leading to increased vascular risk and recurrent stroke are largely unknown. Here, using a novel mouse model of stroke-induced recurrent ischaemia, we show that stroke leads to activation of the AIM2 inflammasome in vulnerable atherosclerotic plaques via an increase of circulating cell-free DNA. Enhanced plaque inflammation post-stroke results in plaque destabilization and atherothrombosis, finally leading to arterioarterial embolism and recurrent stroke within days after the index stroke. We confirm key steps of plaque destabilization also after experimental myocardial infarction and in carotid artery plaque samples from patients with acute stroke. Rapid neutrophil NETosis was identified as the main source of cell-free DNA after stroke and NET–DNA as the causative agent leading to AIM2 inflammasome activation. Neutralization of cell-free DNA by DNase treatment or inhibition of inflammasome activation reduced the rate of stroke recurrence after experimental stroke. Our findings present an explanation for the high recurrence rate after incident ischaemic events in patients with atherosclerosis. The detailed mechanisms uncovered here provide clinically uncharted therapeutic targets for which we show high efficacy to prevent recurrent events. Targeting DNA-mediated inflammasome activation after remote tissue injury represents a promising avenue for further clinical development in the prevention of early recurrent events.

Stroke is the second-leading cause of death worldwide[3] and a leading cause of long-term disability, with incidence rates rising due to demographic ageing[4]. An important factor contributing to this sociomedical burden is the high risk of recurrent vascular events such as stroke and myocardial infarction. A systematic review and meta-analysis reported a pooled recurrent stroke risk of 11.1% at 1 year[1,2]. The risk of myocardial infarction is likewise substantially increased in survivors of stroke[5]. The precise mechanisms underlying this increased vascular risk after an ischaemic event are unclear. It is equally unexplained why the hazards for recurrence are dependent on time and particularly pronounced in the early phase after the event. Previous epidemiological studies have indicated that vascular risk is particularly increased in patients with large-artery atherosclerosis and to a lesser degree in patients with other stroke aetiologies[6]. This suggests an abundant residual vascular risk in patients with atherosclerosis, which is not effectively prevented by current secondary prevention measures.

[1]Institute for Stroke and Dementia Research (ISD), LMU University Hospital, LMU Munich, Munich, Germany. [2]Munich Cluster for Systems Neurology (SyNergy), Munich, Germany. [3]Programme in Medical and Population Genetics, Broad Institute of MIT and Harvard, Cambridge, MA, USA. [4]Medizinische Klinik und Poliklinik I, LMU University Hospital, LMU Munich, Munich, Germany. [5]Department of Developmental Biology and Cancer Research, Hebrew University of Jerusalem, Faculty of Medicine, Jerusalem, Israel. [6]Normandie University, UNICAEN, INSERM UMR-S U1237, Physiopathology and Imaging of Neurological Disorders (PhIND), GIP Cyceron, Institute Blood and Brain @ Caen-Normandie (BB@C), Caen, France. [7]Geoffrey Jefferson Brain Research Centre, The Manchester Academic Health Science Centre, Northern Care Alliance NHS Group, University of Manchester, Manchester, UK. [8]Department of Cardiology, German Heart Centre Munich, Technical University of Munich, Munich, Germany. [9]Center for Stroke Research Berlin (CSB), Charité-Universitätsmedizin Berlin, Berlin, Germany. [10]Department of Neurology, Charité-Universitätsmedizin Berlin, Berlin, Germany. [11]German Center for Cardiovascular Research (DZHK), Partner Site Berlin, Berlin, Germany. [12]Gene Center and Department of Biochemistry, LMU Munich, Munich, Germany. [13]German Center for Cardiovascular Research (DZHK), Partner Site Munich Heart Alliance, Munich, Germany. [14]German Center for Neurodegenerative Diseases (DZNE), Munich, Germany. [15]Research Clinical Department, Caen Normandie University Hospital, Caen, France. [16]Department of Immunopharmacology, Mannheim Institute for Innate Immunoscience (MI3), Medical Faculty Mannheim, Heidelberg University, Mannheim, Germany. [17]Department of Neurology, Hannover Medical School, Hannover, Germany. [22]These authors contributed equally: Jiayu Cao, Stefan Roth. *A list of authors and their affiliations appears at the end of the paper. ✉e-mail: stefan.roth@med.uni-muenchen.de; arthur.liesz@med.uni-muenchen.de

Atherosclerosis is a chronic inflammatory disease, in which immune cells in atherosclerotic plaques contribute to the progression and evolving vulnerability[7]. We and others have previously shown that stroke leads to a systemic inflammatory response, which can contribute to progression of vascular inflammation and plaque load in established atherosclerosis[8–10]. Epidemiological data on inflammatory biomarkers and atheroprogression in patients with stroke have suggested that inflammation can be associated with progression of atherosclerosis and even with recurrent ischaemic events[11,12].

On the basis of these findings, we formulated the hypothesis that stroke might facilitate the occurrence of subsequent vascular events by inflammatory mechanisms. However, the detailed mechanisms along this supposed brain–immune–vascular axis are currently unknown and therefore specific immunological targets to potentially reduce the rate of early recurrence after stroke are missing. A potential reason for this lack of information on such a pressing biomedical question might be the lack of suitable animal models to study recurrent ischaemic events. Commonly used models for atherosclerosis in rodents differ from the situation in high-risk patients with cardiovascular disease in that the atherosclerotic plaques are less complex, not prone to rupture and barely affect the cerebrovascular circulation.

Here we developed an adapted mouse model of rupture-prone high-risk plaques of the carotid artery in combination with contralateral experimental stroke or with myocardial infarction. We found that both stroke and myocardial infarction induced a destabilization of atherosclerotic plaques, leading to recurrent ischaemic events. We identified activation of the AIM2 inflammasome by cell-free DNA (cfDNA) derived from neutrophil extracellular traps (NETs) as the immunological mechanism leading to plaque destabilization via matrix metalloproteinase (MMP) activation, finally leading to atherothrombosis and arterial embolism. The same pathophysiological processes were confirmed in human atherosclerotic plaques obtained from patients within the first days after an acute ischaemic stroke. Targeting this immunological pathway efficiently prevented recurrent ischaemic events after experimental stroke.

## Stroke recurrence in atherosclerosis

As previous epidemiological data on stroke recurrence by stroke aetiology are more than 20 years old[6] and clinical practice has substantially changed in this time period, we assessed the recurrence rate of cerebrovascular events (stroke and transient ischaemic attack) using pooled data ($n = 1,798$ patients with stroke) from two ongoing clinical cohorts with information on aetiology of the first stroke according to TOAST criteria[13,14]. Focusing on the early phase after stroke (days 0–30 after the index event), recurrence rates were markedly higher in patients with large-artery atherosclerotic (LAA) stroke than in other stroke aetiologies and nearly as high as for the entire period from days 31–360 combined (Fig. 1a and Extended Data Fig. 1a). This indicates that current strategies for secondary stroke prevention fail to efficiently reduce early recurrent events in patients with LAA stroke. This clinical notion was further confirmed by treating atherosclerotic mice undergoing experimental stroke with high-dose rosuvastatin and aspirin—the current standard scheme for secondary stroke prevention in patients. This high-dose treatment had no effect on mortality, vascular inflammation and plaque load within the first week after experimental stroke (Extended Data Fig. 1b–f), confirming the inefficiency of current secondary prevention therapy on early atherosclerotic plaque progression within the first week after stroke. To study the mechanisms underlying the increased risk of recurrent events in patients with atherosclerosis, we established an animal model of unilateral highly stenotic and haemodynamically relevant atherosclerotic plaques of the common carotid artery (CCA). This involved the induction of turbulent flow by a stenotic tandem ligation (Extended Data Fig. 2a–f). We used this model to test the effects of experimentally induced acute ischaemic stroke in the hemisphere contralateral to the CCA tandem stenosis on atherosclerotic plaque morphology and destabilization (Fig. 1b). We screened for the occurrence of secondary ischaemic events in the brain hemisphere supplied by the stenotic atherosclerotic CCA (that is, contralateral to the experimentally induced stroke) by magnetic resonance imaging (MRI) and histological analysis of cell loss (Fluoro Jade C and TUNEL staining) and microgliosis (Fig. 1c). We found that experimental stroke resulted in secondary brain ischaemia in 30% of the animals with secondary lesions mainly located in the middle cerebral artery (MCA) territory (Extended Data Fig. 2g–i). By contrast, this was not observed in any of the animals with CCA tandem stenosis and sham surgery or atherosclerotic animals without CCA tandem stenosis (Fig. 1c and Extended Data Fig. 2j–l). This observation suggested stroke-induced rupture of remote CCA plaques leading to secondary stroke events. This hypothesis was further supported by the identification of plaque rupture and intravascular thrombus formation at the site of the CCA stenosis, which was significantly associated with the occurrence of secondary brain lesions after stroke (Fig. 1d). Correspondingly, the plaque vulnerability index for the stenotic CCA (a previously established score of several morphological markers indicating plaque risk to rupture[15]; Extended Data Fig. 3a–f) was significantly increased in mice after experimental stroke and was even further exacerbated in mice with secondary events after the primary stroke (Fig. 1e). Fibrotic cap thickness, which contributes to plaque stability, was significantly decreased after stroke (Extended Data Fig. 3d), and histological indication of plaque rupture was observed in 40% of mice after stroke compared with 10% of mice undergoing sham surgery (Extended Data Fig. 3g). We further validated the concept of secondary remote plaque rupture in a model of myocardial infarction. Similar to stroke, myocardial infarction increased vulnerability of the remote CCA plaque (Fig. 1f and Extended Data Fig. 4a–e). Flow cytometry analysis of CCA plaques revealed increased cellular plaque inflammation after stroke compared with sham surgery (Extended Data Fig. 4f,g). We further analysed the mechanism of enhanced immune cellularity either by invasion or proliferation using antibody labelling of circulating immune cells or histological assessment of the proliferation marker Ki67, respectively. We found that both the de novo recruitment and local proliferation of leukocytes were increased after stroke (Fig. 1g–i) and that the local proliferation rate was associated with the occurrence of recurrent events (Fig. 1i).

## Post-stroke DNA activates AIM2 in plaques

Further analyses of the inflammatory milieu in atherosclerotic CCA plaques revealed a substantial increase in local IL-1β production after stroke, suggesting stroke-induced inflammasome activation within atherosclerotic plaques (Fig. 2a). Local inflammasome activation was confirmed by western blot analysis of cleaved caspase-1—the effector enzyme of the inflammasome—in the atherosclerotic plaque and flow cytometric analysis using FAM, a fluorescent molecule that selectively binds to activated caspase-1, for cell-based analysis of inflammasome activation in plaque macrophages (Fig. 2b) and independently verified by histological analysis with increased caspase-1 expression in CCA plaques after stroke (Extended Data Fig. 5a–d). Blocking inflammasome activity by a caspase-1 inhibitor (VX765) prevented the proliferation of immune cells within CCA plaques after stroke and attenuated the invasion of pro-inflammatory circulating monocytes (Fig. 2c,d). Consequently, inflammasome inhibition significantly reduced plaque vulnerability after experimental stroke to levels comparable with sham-operated mice (Fig. 2e and Extended Data Fig. 5e–g). We next investigated which specific inflammasome mediates this effect of stroke on CCA plaque exacerbation by applying specific pharmacological inhibitors of the NLRP3 (MCC950) or a novel AIM2 inflammasome inhibitor[16] (4-sulfonic calixarene), which we found to dose-dependently displace DNA, which is the ligand of

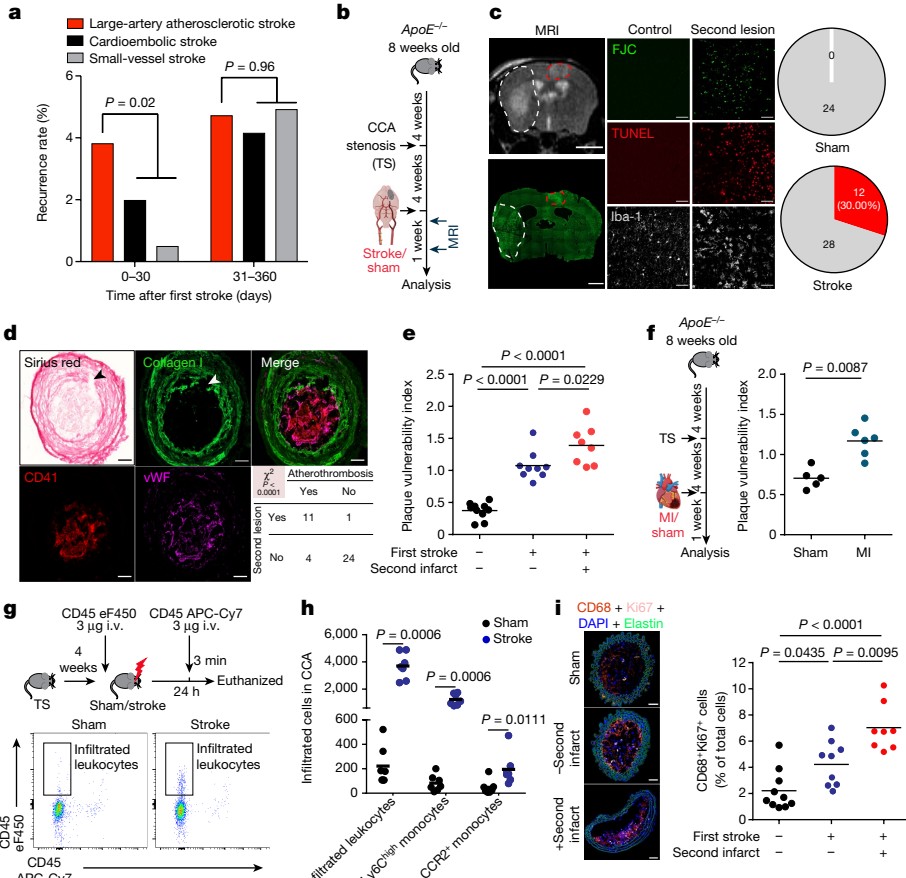

**Fig. 1 | Ischaemic events induce recurrent stroke and exacerbate plaque vulnerability. a**, Recurrence rates by incident stroke aetiology in the first month (days 0–30) and months 2–12 (days 31–360) in a population of 1,798 patients with stroke (log-rank test in Kaplan–Meier curves; Extended Data Fig. 1a). **b**, Experimental design. Eight-week-old high-cholesterol diet (HCD)-fed *ApoE*[−/−] mice underwent tandem stenosis (TS) surgery, and stroke or sham surgery 4 weeks later. The recurrence of secondary ischaemia in the contralateral hemisphere was examined by MRI and histological analysis. **c**, Representative MRI image (the white dashed line denotes the primary stroke area; the red dashed line indicates recurrent contralateral stroke). Fluoro Jade C (FJC; lower left panel) staining corresponding to MRI. Scale bars, 3 mm. Representative images of histological stainings (FJC, TUNEL and Iba-1) from control mice or mice with a secondary lesion (middle panels). Scale bars, 50 μm. Pie charts for stroke recurrence 7 days after sham or stroke surgery (*n* = 24 (sham) and *n* = 40 (stroke); red denotes with secondary lesion, and grey indicates without

secondary lesion) are also shown (right panels). **d**, Representative images of a plaque rupture (the arrowheads in the Sirius red and collagen I stainings) with contiguous CCA thrombus stained for von Willebrand factor (vWF) and platelets (CD41). Chi-squared test (*P* < 0.0001) for occurrence of CCA thrombi and secondary brain lesions (*n* = 12 per group). Scale bars, 50 μm. **e**, Quantification of plaque vulnerability in CCA sections 7 days after sham or stroke surgery (analysis of variance (ANOVA); *n* = 9–10 per group). **f**, Experimental design as shown in panel **b**, but with induction of myocardial infarction (MI) instead of stroke (left), and quantification of plaque vulnerability (right; *U*-test; *n* = 5–6 per group). **g**,**h**, Experimental design and representative FACS plots (**g**) for quantification (**h**) of infiltrated leukocytes and Ly6C[high] and CCR2[+] monocyte subpopulations in CCA 24 h after sham or stroke surgery (*U*-test; *n* = 6 per group). i.v., intravenous. **i**, Representative images (left) and quantification (right) of proliferating macrophages (ANOVA; *n* = 8–10 per group). Scale bars, 50 μm. **e**,**f**,**h**,**i**, Bars indicate the mean.

---

the AIM2 inflammasome (Extended Data Fig. 6a,b). We focused on these two abundant inflammasome subtypes, because the NLRP3 inflammasome has previously been implicated in the development of atherosclerosis, whereas stroke leads to systemic AIM2 inflammasome activation[17,18]. Both the inhibition of AIM2 and NLRP3 prevented inflammasome activation in the CCA plaque after stroke, with greater efficacy of the AIM2 inhibitor, whereas only the AIM2 inhibitor significantly prevented the post-stroke increase in IL-1β levels in the CCA plaque (Fig. 2f,g and Extended Data Fig. 6c).

Moreover, we observed a transient increase in the serum concentration of cfDNA in the acute phase after stroke and myocardial infarction, suggesting that release of cfDNA after stroke and myocardial infarction could be the mediator linking remote organ ischaemia to the exacerbation of plaque inflammation (Fig. 2h and Extended Data Fig. 6d,e). The increased cfDNA after stroke was mainly DNA of less than 400 bp in length and confirmed to be primarily extravesicularly located (Extended Data Fig. 6f–h). To test the causative

function of cfDNA, we injected DNA into mice with a CCA stenosis without stroke or sham surgery and observed increased plaque inflammasome activation similar to the condition after stroke (Fig. 2i,j and Extended Data Fig. 7a,b). Correspondingly, stimulation of wild-type (WT) bone marrow-derived macrophages (BMDMs), but not of AIM2-deficient BMDM, with DNA led to oligomerization of the inflammasome adaptor protein ASC as an indicator of physical inflammasome formation (Fig. 2k). Moreover, caspase-1 activation was completely blunted in ASC-deficient or AIM2-deficient, but not in cGAS-deficient, NLRP1-deficient or NLRP3-deficient, BMDMs in response to stimulation with serum from stroke mice or stimulation with DNA (Extended Data Fig. 7c,d). To further analyse in vivo, the role of the AIM2 inflammasome in myeloid cells, we performed bone marrow transfer using a genetic bone marrow depletion model (poly(I:C) administration to *Mx1*[cre]:*c-myb*[fl/fl] mice) on the genetic background of atherogenic LDL-receptor-deficient mice to avoid confounding effects by irradiation or chemotherapy (Fig. 2l). *Aim2*[WT] or *Aim2*[−/−] bone

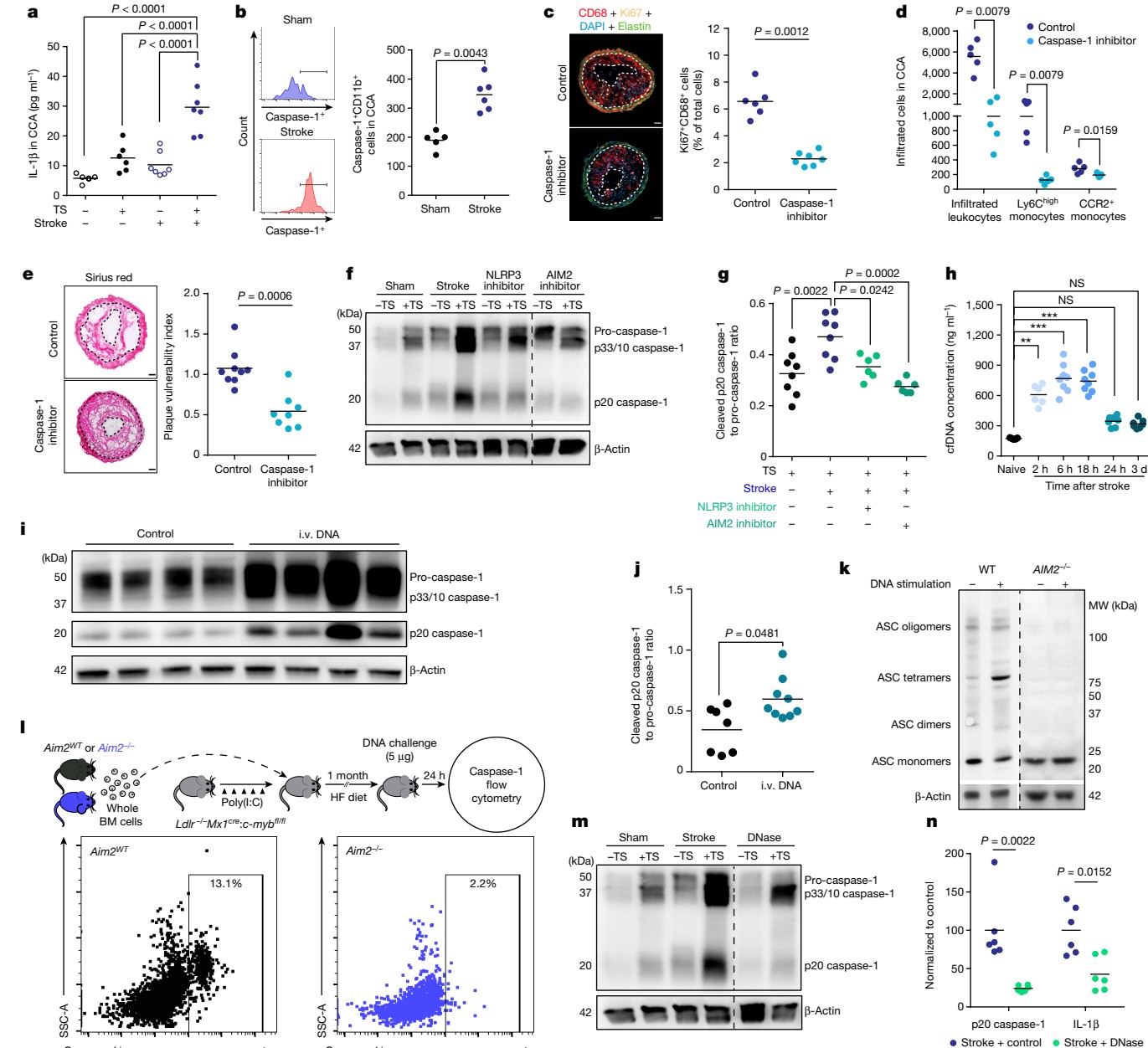

**Fig. 2 | Stroke induces double-stranded DNA-dependent inflammasome activation in atherosclerotic plaques. a**,**b**, IL-1β concentrations (*K*-test, *n* = 5–7 per group; **a**) and caspase-1 activity (FAM660 flow cytometry; **b**) in CCA 24 h after sham or stroke surgery (*U*-test, *n* = 5–6 per group). **c**,**d**, Analysis of macrophage proliferation normalized to total cell counts (right; *U*-test, *n* = 6–7 per group; **c**) and analysis of infiltrated leukocytes (**d**) in CCA sections of control-treated or caspase-1 inhibitor (VX765)-treated mice 7 days after stroke (*K*-test; *n* = 5 per group). Scale bars in **c**, 50 μm. **e**, Picro Sirius red stainings for stroke (left) and quantification of plaque vulnerability (right; *U*-test; *n* = 8–9 per group). Scale bars, 50 μm. **f**,**g**, Immunoblot for caspase-1 cleavage in CCA lysates 24 h after sham, stroke, stroke + NLRP3 inhibitor (MCC950) or AIM2 inhibitor (4-sulfonic calix[6]arene) administration (**f**) and corresponding immunoblot quantification (ANOVA; *n* = 8 per group; **g**). **h**, Total cfDNA serum concentrations at indicated timepoints after stroke and naive mice (*K*-test;

*n* = 5–8 per group). NS, not significant. **i**,**j**, Immunoblot (**i**) and quantification (**j**) of cleaved p20 caspase-1 in CCA lysates from HCD-fed *ApoE*−/− mice with tandem stenosis surgery alone (control) or 24 h after i.v. DNA injection (*U*-test; *n* = 7–9 per group). **k**, Immunoblot for ASC oligomerization of WT or *AIM2*−/− macrophages after stimulation with DNA. MW, molecular weight. **l**, Experimental design of *Aim2*WT or *Aim2*−/− bone marrow (BM) transplantation to *Ldlr*−/−:*Mx1*cre:*c-myb*fl/fl mice receiving i.v. DNA (top) and flow cytometry histograms (bottom) of caspase-1 activity in bone marrow recipients 24 h after i.v. DNA. HF, high fat; SSC-A, side scatter area. **m**,**n**, Immunoblot (**m**) and quantification (**n**) of cleaved p20 caspase-1 in CCA lysates of HCD-fed *ApoE*−/− with tandem stenosis surgery receiving 1,000 U i.v. DNase immediately before stroke (*U*-test; *n* = 6 per group, shown as a ratio to the mean of the control group). Raw membrane images of all immunoblots are in Supplementary Fig. 1. **a–e**,**g**,**h**,**j**,**n**, Bars indicate the mean.

marrow-reconstituted mice received a high-fat diet (HFD) for 4 weeks and were then stimulated by intravenous DNA injection to test the in vivo relevance of the specific DNA–AIM2 interaction, confirming an abrogated caspase-1 activation in AIM2-deficient monocytes/ macrophages. By contrast, neutralization of cfDNA after stroke by therapeutic administration of recombinant DNase following stroke

induction again significantly reduced the plaque inflammasome activation, as measured by caspase-1 cleavage and IL-1β secretion (Fig. 2m,n and Extended Data Fig. 7e). In summary, we identified the activation of the AIM2 inflammasome in vulnerable atherosclerotic plaques by stroke-induced DNA release and subsequent plaque rupture as the cause of recurrent ischaemic events.

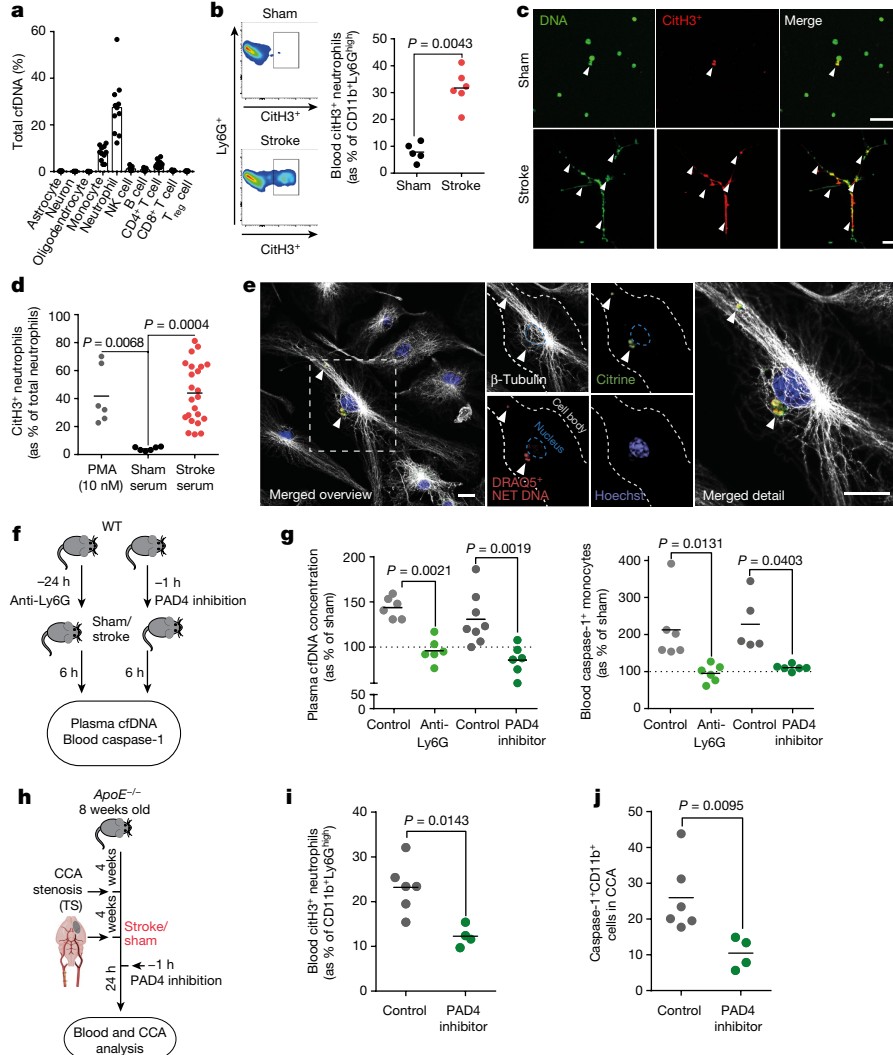

**Fig. 3 | Neutrophil NETosis is the main source of post-stroke cfDNA. a**, Tissue-of-origin analysis using cell-type-specific methylation markers for cfDNA isolated from $n = 10$ patients with stroke within 24 h after symptom onset. NK, natural killer; $T_{reg}$ cell, regulatory T cell. **b**, Flow cytometry histograms (left) and quantification (right) of citrullinated histone 3 (citH3[+]) Ly6G[+] neutrophils 4 h after sham or stroke ($U$-test; $n = 5$–6 per group). **c**, Microphotographs of mouse neutrophils stimulated with 25% sham or stroke serum for 4 h. The arrowheads indicate DNA[+]citH3[+] NET formations. Scale bars, 50 μm. **d**, Quantification of phorbol 12-myristate 13-acetate (PMA)-stimulated, sham serum-stimulated or stroke serum-stimulated neutrophils, shown as citH3[+] neutrophils of total cultured neutrophils ($K$-test; $n = 6$–23 replicates per group from 3 independent experiments). **e**, BMDMs from ASC–citrine reporter mice were incubated with 250 ng exogenous DRAQ5-labelled NET–DNA. Immunohistochemistry was used to visualize cytoplasmatic citrine[+]DRAQ5[+] ASC–DNA specks. The cytoplasm and nucleus of the BMDMs were visualized using β-tubulin and Hoechst. Arrowheads indicate colocalization of ASC specks and DRAQ5[+] NET DNA. Scale bars, 5 μm. **f**, Experimental design for anti-Ly6G neutrophil depletion and PAD4 inhibition before stroke. **g**, cfDNA concentration (shown as percentage normalized to sham) of mice with control, anti-Ly6G or PAD4 inhibitor treatment (left; $K$-test; $n = 6$–8 per group) and caspase-1 activity in blood monocytes (percentage of sham, indicated by dotted line) by flow cytometry (right; $K$-test; $n = 5$–8 per group). **h**, NETosis was inhibited using a PAD4 inhibitor before stroke in HCD-fed $ApoE^{-/-}$ with tandem stenosis. **i**, Quantification of citH3[+]Ly6G[+] neutrophils 24 h after stroke surgery in blood ($U$-test; $n = 4$–6 per group). **j**, Flow cytometry analysis of caspase-1 activity in CD11b[+] CCA monocytes 24 h after stroke ($U$-test; $n = 4$–6 per group). **b,d,g,i,j**, Bars indicate the mean.

## NETosis, the main source of post-stroke DNA

To identify the cellular DNA source, we performed a methylation pattern analysis of isolated blood cfDNA from patients with stroke within the first 24 h after stroke onset, based on a previously established DNA methylation atlas of human cell types[19]. The main cerebral cell populations—astrocytes, neurons and oligodendrocytes—did not contribute to the increased blood cfDNA levels after stroke, which was mainly attributed to a neutrophil-specific methylome pattern and to a lesser extent to other immune cells (Fig. 3a and Extended Data Fig. 7f). To further validate this observation, we performed flow cytometric analysis of NET formation after experimental stroke in mice and identified a substantial increase in neutrophil NETosis indicated by citrullinated histone H3-positive neutrophils (Fig. 3b); correspondingly, serum from mice after experimental stroke was sufficient to induce neutrophil NETosis of naive primary neutrophils ex vivo (Fig. 3c,d). By administration of fluorescently labelled NET–DNA to primed BMDMs from ASC–citrine reporter mice[20], we confirmed the effective uptake of NET–DNA and colocalization of exogenous NET–DNA with ASC specks resembling perinuclear inflammasomes (Fig. 3e). To test the contribution of neutrophil NETosis to post-stroke cfDNA in WT animals, we either depleted neutrophils by Ly6G-specific antibodies or inhibited NETosis by blocking the PAD4 molecule, which is required for NETosis (Fig. 3f and Extended Data Fig. 7g,h). Both depletion of the neutrophil population and inhibition of PAD4 efficiently prevented the post-stroke increase in blood cfDNA levels and reduced caspase-1 activation to levels in Sham-operated

control mice (Fig. 3g). Finally, we tested the efficacy of PAD4 inhibition in our animal model of rupture prone CCA plaques in APOE-deficient animals with CCA tandem stenosis and observed a similar reduction in post-stroke NETosis as in WT animals, and PAD4 inhibition efficiently reduced inflammasome activation in CCA plaques (Fig. 3h–j). Here we identified an unexpected prominent role of NETosis after stroke to the rapid increase in cfDNA levels, which represents a druggable process to reduce post-stroke systemic inflammation.

## MMPs lead to plaque erosion

Conceivably, enhanced plaque inflammation—as observed here after stroke induction—could contribute to plaque rupture and secondary arterioarterial embolism resulting in secondary infarctions. However, the exact mechanisms linking plaque inflammation to plaque destabilization and atherothrombosis are poorly studied. A more detailed study of the plaque fibrous cap composition by analysing its collagen fibre orientation indicated collagen fibre disorganization after stroke, suggesting extracellular matrix (ECM) remodelling (Fig. 4a). Indeed, we found that stroke resulted in increased activity of MMPs—the main enzymes involved in ECM remodelling—which was detected by in situ zymography of CCA plaque sections and validated by gel zymography for MMP2 and MMP9 (Fig. 4b and Extended Data Fig. 8a,b). To test whether soluble blood mediators mediate this effect on MMP activity after stroke, we treated BMDM with serum obtained from mice 4 h after stroke or sham surgery. Conditioning of BMDMs with serum from stroke mice resulted in massively increased active MMP2 and MMP9 secretion compared with sham serum treatment, both in total protein content (western blot) and enzymatic activity (zymography), suggesting a causative role of circulating factors after stroke on MMP expression and activation in the CCA plaque (Fig. 4c and quantification in Extended Data Fig. 8c–e). Given our finding of a critical role of AIM2 inflammasome activation, we tested the influence of IL-1β derived from inflammasome activation on MMP activity by treating BMDMs with increasing doses of recombinant IL-1β. Indeed, we detected a dose-dependent effect of IL-1β on MMP expression (Extended Data Fig. 8f,g). Moreover, the supernatant of NET–DNA-stimulated WT but not $Aim2^{-/-}$ BMDMs was sufficient to induce MMP2 expression in naive BMDMs (Extended Data Fig. 8h–j). Next, we tested the function of IL-1β in vivo by its neutralization using IL-1β-specific antibodies in a dose that we previously established to efficiently block IL-1β-mediated systemic effects[18,21]. Neutralization of IL-1β after experimental stroke significantly reduced MMP2/9 activity in CCA plaques 7 days after stroke and increased fibrous cap thickness compared with control (isotype IgG) treatment, indicating a causative role of IL-1β in mediating inflammatory plaque degradation (Fig. 4d,e). Although MMP-mediated degradation of the extracellular matrix causes plaque destabilization[22], formation of the actual thrombus on ruptured plaques depends on activation of the coagulation cascade, particularly by the contact activation pathway initiated by factor XII exposition to damaged tissue surfaces[23]. We therefore performed en face staining of the whole CCA and compared activated factor XII deposition on the luminal surface at the area of the CCA plaque between stroke-operated or sham-operated mice. We found a significant increase in factor XIIa deposition on the plaque surface after stroke (Fig. 4f), which was confirmed by western blot analysis of whole CCA plaques (Fig. 4g and Extended Data Fig. 9a). An intravenous NET–DNA challenge without stroke mimicked the effect of stroke with increased factor XIIa deposition at the CCA plaque (Fig. 4h and Extended Data Fig. 9b). These findings suggest that the AIM2 inflammasome activation and resulting IL-1β release after stroke could lead to plaque destabilization and atherothrombosis via activation of plaque-degrading MMPs and subsequent factor XIIa deposition. To further test this hypothesis, we performed in vivo MRI using time of flight (TOF)-MR angiography in combination with intravascular thrombus imaging using a recently established magnetic matrix particle-based imaging modality[24]. Here we were able to identify CCA occlusion 6 h after stroke induction, which coincided with intravascular thrombi formation in the affected distal MCA segments (Fig. 4i). Correspondingly, in vivo inflammasome inhibition using the caspase-1 inhibitor VX765 significantly reduced plaque MMP activity and luminal factor XIIa deposition (Fig. 4j,k and Extended Data Fig. 9c). Similarly, intravenous DNase treatment significantly reduced MMP activity and factor XIIa deposition (Fig. 4l and Extended Data Fig. 9d).

## DNase reduces stroke recurrence

Finally, we analysed whether blocking DNA-mediated inflammasome activation after stroke could be used therapeutically to prevent recurrent ischaemic events. For this, mice with CCA tandem stenosis were treated with the caspase-1 inhibitor VX765 or recombinant DNase. The occurrence of secondary ischaemic lesions was analysed 7 days after induction of the primary stroke and in the hemisphere contralateral to the primary stroke, that is, within the territory supplied by the stenotic CCA. Indeed, both caspase-1 inhibition and DNase treatment greatly and significantly reduced the recurrence rate (Fig. 4m). Of note, this therapeutic effect was achieved in a large sample size of animals (total sample size of 117 mice) and translates to a relative risk reduction of 82% and 75% for VX765 and DNase, respectively.

## Validation in patients with stroke

To further validate the translational relevance of the identified mechanisms, we obtained carotid endarterectomy samples from highly stenotic carotid artery plaques of asymptomatic patients and patients during the acute phase of ischaemic stroke (Fig. 5a). Flow cytometry analysis of the plaque material revealed a significant increase in monocyte counts in plaques from symptomatic compared with asymptomatic patients (Fig. 5b), whereas blood monocyte and lymphocyte counts did not differ (Extended Data Fig. 10a–c). Correspondingly, we found a significant increase in circulating cfDNA in plasma of patients with stroke, as well as a substantial increase in markers of inflammasome priming (pro-caspase-1 expression) and inflammasome activation (cleaved p20 isoform of caspase-1) in plaque material (Fig. 5c–f). We observed a similar increase in cfDNA in patients after myocardial infarction, emphasizing again the generalizability of our findings across ischaemic organ damage (Fig. 5g). Of note, we also detected a more than tenfold increase in MMP9 activity by gel zymography of plaques from symptomatic compared with asymptomatic patients (Fig. 5h and Extended Data Fig. 10d,e). Finally, the amount of plaque-associated factor XIIa was significantly increased in atherosclerotic plaques after stroke (Fig. 5i). Hence, the analyses of human endarterectomy samples confirmed all key steps of DNA-mediated inflammasome activation, vascular inflammation, MMP activation and initiation of thrombus formation in patients with stroke as identified in our animal model.

## Discussion

Early recurrent events after ischaemic stroke present a pressing sociomedical problem with an unmet need. Current secondary prevention therapies target mainly blood lipid levels (statins) and platelet aggregation (aspirin), which are effective for long-term prevention of vascular events. However, these therapies have overall an only minor effect recurrence after large-artery stroke[2], most likely because this is driven by so far largely unknown immunological mechanisms[11]. Here we identified inflammasome activation by cfDNA as an initiator of an inflammatory cascade leading to atherosclerotic plaque degradation and thrombosis (Extended Data Fig. 10f). Furthermore, we detected neutrophil NETosis as the unexpected major source of post-stroke cfDNA. Our results propose the inhibition of NETosis, the

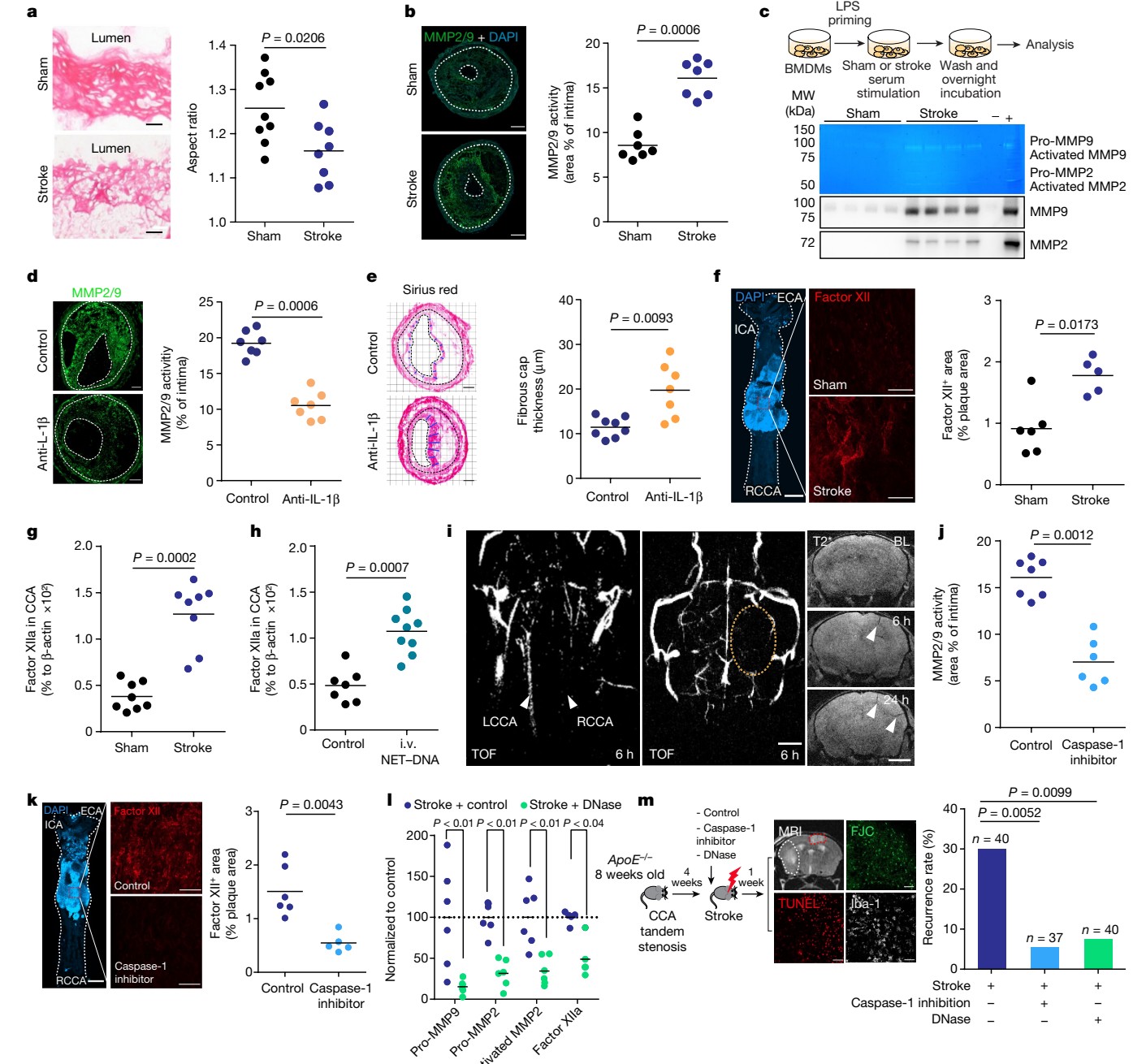

**Fig. 4 | Inhibition of post-stroke inflammasome activation prevents plaque destabilization and recurrent stroke events. a**, Collagen I orientation in fibrous caps using aspect ratio (AR > 1 is predominant orientation and AR ~ 1 is random orientation; *U*-test; *n* = 8–9). Scale bars, 50 μm. **b**, Quantification of MMP2/9 activity by in situ zymography on CCA sections 7 days after sham or stroke (*U*-test; *n* = 7). Scale bars, 50 μm. **c**, Experimental design (top), representative images (middle) for gelatin zymography and immunoblot (bottom) of MMP2/9 in culture medium (quantification in Extended Data Fig. 8c,d). **d**, In situ zymography of CCA sections 7 days after stroke, receiving control or IL-1β-specific antibodies (*U*-test; *n* = 7). Scale bars, 50 μm. **e**, Analysis of fibrous cap thickness between treatment groups (*U*-test; *n* = 7–8). Scale bars, 50 μm. **f**, Whole CCA en face immunohistology for factor XII 24 h after sham or stroke (*U*-test; *n* = 5–6). ECA, external carotid artery; ICA, internal carotid artery; RCCA, right CCA. Scale bars, 500 μm. **g**, Activated factor XII in CCA lysates 7 days after sham or stroke (*U*-test; *n* = 8). **h**, Factor XIIa in CCA lysates 24 h after

i.v. DNA injection (*U*-test; *n* = 7–9). **i**, HCD-fed *ApoE⁻/⁻* with tandem stenosis surgery received i.v. DNA. Cerebral vascularization (TOF-MR angiography) and intravascular thrombus formation by in vivo MRI in HCD-fed *ApoE⁻/⁻*. Left CCA (LCCA) and (patent) right CCA (RCCA) are indicated by arrowheads (left); the orange dotted circle (middle panel) denotes the hypoperfused right middle cerebral artery territory. The arrowheads (right) indicate intravascular thrombi. Scale bars, 2 mm. BL, baseline. **j**, In situ zymography for MMP2/9 activity in CCA sections 7 days after stroke (*U*-test; *n* = 6–7). **k**, Factor XII⁺ area on CCA en face images between treatment groups (*U*-test; *n* = 5–6). Scale bars, 500 μm. **l**, Pro-MMP9, pro-MMP2, activated MMP2 and factor XIIa quantification in mice receiving 1,000 U DNase after stroke (*U*-test; *n* = 6, normalized to control, indicated by horizontal dotted line). **m**, Experimental design (left) and quantification of the 7-day recurrence rate between treatment groups (right; chi-squared test). Scale bars, 50 μm. **a**,**b**,**d**–**h**,**j**–**l**, Bars indicate the mean.

neutralization of cfDNA or inhibition of downstream inflammasome activation as efficient therapeutic approaches to prevent recurrent vascular events.

The CANTOS trial—testing the use of IL-1β-specific antibody treatment in patients with previous myocardial infarction—has clearly highlighted the relevance of residual inflammatory risk and demonstrated

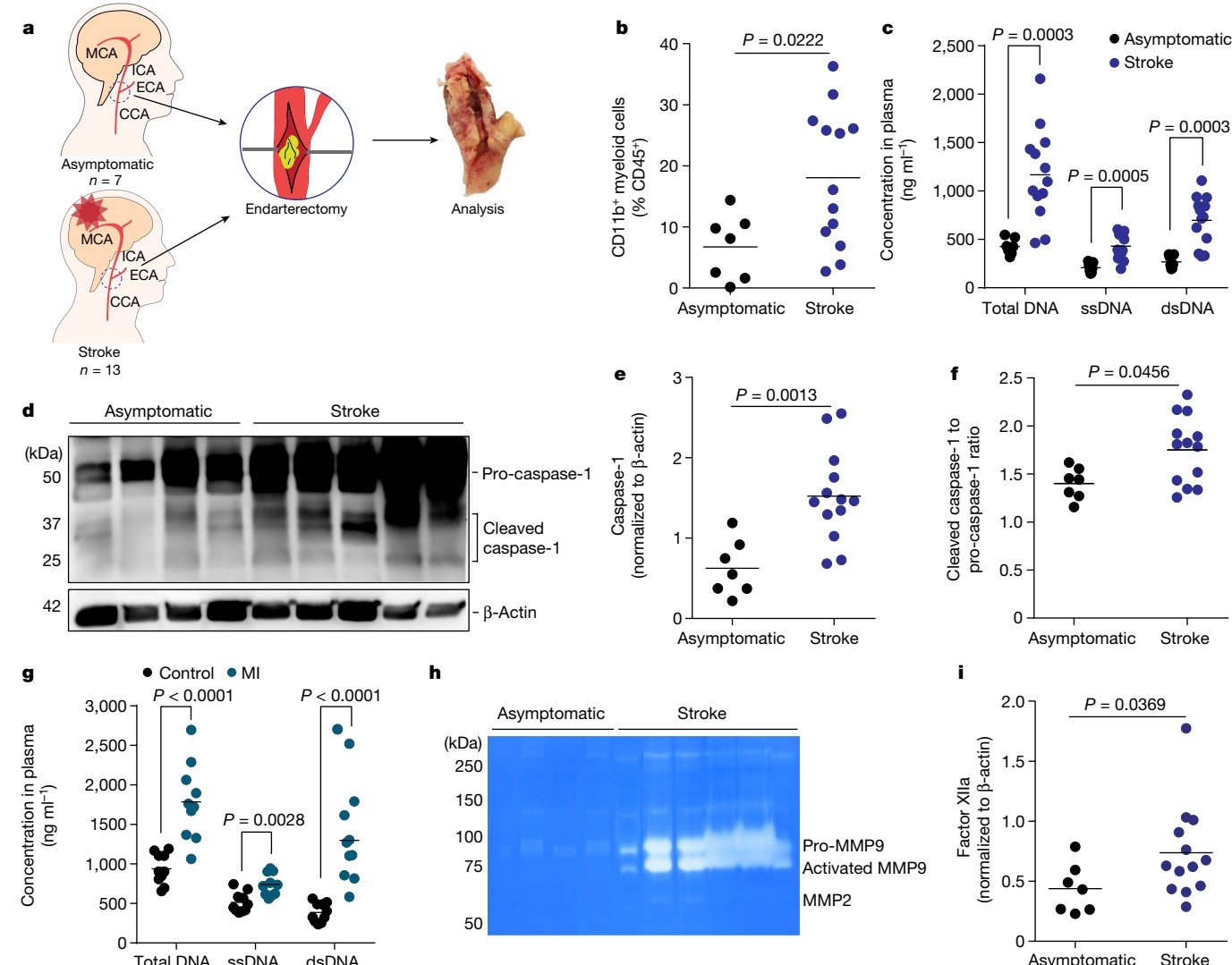

**Fig. 5 | Stroke increases atherosclerotic plaque inflammation and MMP activity in patients. a**, Study layout illustrating the collection of endarterectomy samples from 7 asymptomatic patients with ICA stenosis and 13 patients with stroke undergoing endarterectomy of the symptomatic carotid artery in the acute phase after stroke. **b**, Flow cytometry analysis of plaques showing the percentage of CD11b[+] myeloid cells out of total leukocytes (CD45[+]; *U*-test; $n = 7$–13 per group). **c**, Quantification of total cfDNA (total DNA), single-stranded DNA (ssDNA) and double-stranded DNA (dsDNA) in plasma (*U*-test; $n = 7$–13 per group). **d**–**f**, Immunoblot for caspase-1 cleavage in atherosclerotic plaque lysates of asymptomatic or symptomatic patients (**d**) and corresponding quantification of total caspase-1 (**e**, normalized to β-actin) and cleaved p20 caspase-1 (**f**, normalized to caspase-1; *U*-test; $n = 7$–13 per group). **g**, Plasma cfDNA concentrations in patients with myocardial infarction (*U*-test; $n = 10$ per group). **h**, Gelatine zymography of plaque lysates from asymptomatic or symptomatic patients (quantification in Extended Data Fig. 10e). **i**, Factor XIIa in plaque lysates from asymptomatic or symptomatic patients (*U*-test; $n = 7$–13 per group). **b**,**c**,**e**–**g**,**i**, Bars indicate the mean.

the potential of anti-inflammatory therapies to prevent recurrent ischaemic events[25]. However, targeting this central cytokine of innate immune defence also significantly increased the risk of fatal infections, which might preclude further clinical development of such unspecific anti-inflammatory strategies[26]. A critical limitation of previous clinical trials targeting inflammatory mechanisms in high-risk patients with atherosclerosis was the insufficient mechanistic knowledge on the exact immunological events that resulted in recurrent vascular events. This knowledge gap has so far prevented the development of more specific, and thereby safer, therapeutic strategies targeting molecular mechanisms of stroke recurrence.

Here we identified increased circulating NET–DNA concentrations and activation of the AIM2 inflammasome as the key mechanisms leading to exacerbated plaque inflammasome activation, whereas other inflammasomes or redundant inflammatory mechanisms could additionally contribute to this phenomenon. We demonstrated that the

increase in circulating cfDNA was sufficient and causally involved in plaque destabilization using an in vivo cfDNA-challenge experiment and validated in human patient samples. In addition, cfDNA was sufficient to induce CCA occlusion and intravascular thrombi in downstream vascular territories as observed by in vivo MRI and histological analysis. Correspondingly, use of recombinant DNase efficiently prevented plaque inflammation, destabilization and recurrent ischaemic events. No direct immunosuppressive function is known for the experimental or clinical use of DNase in various disease conditions including cystic fibrosis and pleural infections[27–29]. By contrast, we have previously demonstrated that prevention of AIM2 inflammasome activation by DNase treatment might even paradoxically improve immunocompetence during secondary bacterial infections, because reduction of early inflammasome activation in response to tissue injury prevents subacute immunosuppression[18,30]. Of note, lower endogenous DNase activity has recently been described as an independent risk factor

for stroke-associated infections following severe ischaemic stroke[31]. Therefore, the use of DNase is a promising candidate for further clinical development as a therapeutic approach in early secondary prevention that is highly efficient but also safe due to its specific and non-immunosuppressive function. Hence, we have initiated a clinical proof-of-concept trial that will test the efficacy of DNase treatment for the prevention of systemic inflammation in patients with stroke (ReScInD trial; ClinicalTrials.gov ID: NCT05880524). Of note, although DNase I proved to be highly efficient to prevent cfDNA-mediated vascular inflammation, we cannot exclude (synergistic) efficacy of other nucleases to prevent AIM2 inflammasome activation.

Not only the immunogenic mediators leading to inflammatory plaque rupture but also the exact pathways contributing to local plaque destabilization and recurrent atherothrombosis after stroke were so far elusive. Here we focused on the role of MMP-mediated plaque degradation and subsequent initiation of the contact-dependent coagulation cascade. In correspondence with our observations, increased plaque MMP activity has been previously associated with markers of increased plaque vulnerability[22]. In addition, increased blood MMP9 activity has been previously associated with worse stroke outcome[32]. We observed that inflammasome activation and specifically the release of local IL-1β can dose-dependently increase MMP activity. Moreover, we demonstrated that neutralization of systemic IL-1β prevents MMP activation and plaque degradation after stroke. It is likely that MMP-mediated degradation of the plaque ECM leads to destabilization and exposition of ECM components to the blood circulation[33]. Correspondingly, we found inflammasome-dependent luminal accumulation of factor XIIa at the CCA plaque—a process that we found to be associated with the occurrence of secondary infarctions. Factor XII activation at the site of vascular damage has been previously proposed as an initiator mechanism of thrombus formation on ruptured plaques, and its pharmacological targeting has recently been demonstrated to stabilize atherosclerotic plaques[34]. Therefore, on the basis of our findings, factor XIIa might represent another potential therapeutic target to prevent recurrent ischaemic events and warrants further exploration.

In summary, in this study, we present a mechanistic explanation for the high rate of early recurrent events after stroke and myocardial infarction in patients with atherosclerosis. Using a novel animal model of post-stroke plaque rupture and secondary infarctions, we identified the immunological mechanisms and validated these in human carotid artery plaque samples. We confirmed the efficient therapeutic targeting of this signalling cascade with DNase administration as a promising therapeutic candidate for further clinical translation.

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

**DEMDAS Study Group**

**Martin Dichgans[13], Matthias Endres[18], Marios K. Georgakis[1], Thomas G. Liman[18], Gabor Petzold[19], Annika Spottke[19], Silke Wunderlich[20] & Inga Zerr[21]**

[18]German Center for Neurodegenerative Diseases (DZNE), Berlin, Germany. [19]German Center for Neurodegenerative Diseases (DZNE), Bonn, Germany. [20]Department of Neurology, Klinikum rechts der Isar, School of Medicine, Technical University of Munich, Munich, Germany. [21]German Center for Neurodegenerative Diseases (DZNE), Göttingen, Germany.

## Methods

All consumables, such as antibodies, chemicals and so on can, be found in the resource list in Supplementary Table 6.

### Animal experiments

All animal experiments were performed in accordance with the guidelines for the use of experimental animals and were approved by the government committee of Upper Bavaria (Regierungspraesidium Oberbayern; #02-2018-12). All mice used in this study were between 6 and 16 weeks of age independent of genotype and experimental setup. $ApoE^{-/-}$ (B6.129P2-Apoe$^{tm1Unc}$/J; JAX strain: 002052), WT (C57BL/6J; JAX strain: 000664), $Aim2^{-/-}$ (B6.129P2-Aim2$^{Gt(CSG445)Byg}$/J; JAX strain: 013144), $Pycard^{-/-}$ ($Asc^{-/-}$; B6.129S5-Pycard$^{tm1Vmd}$) and R26-CAG-ASC-citrine (B6.Cg-Gt(ROSA)26or$^{tm1.1(CAG-Pycard/mCitrine*,-CD2*)}$; JAX strain: 030744) mice were bred and housed at the animal core facility of the Centre for Stroke and Dementia Research (Munich, Germany). $Ldlr^{-/-}$:$Mx1^{cre}$:$c$-$Myb^{fl/fl}$ mice were bred and housed at the animal facility of Walter Brendel Centre (Munich, Germany). $ApoE^{-/-}$ mice were fed an HFD (#88137, ssniff) from 8 weeks onwards. $cGas^{-/-}$ (B6(C)-Cgas$^{tm1d(EUCOMM)Hmgu}$/J), $Nlrp1^{-/-}$ (Del(11Nlrp1a-Nlrp1c-ps)$^{1Smas}$) and $Nlrp3^{-/-}$ (C57BL/6J-NLRP3$^{tm1Tsc}$) mice were bred and housed at the Gene Centre of the LMU University Munich (Germany).

For this exploratory study, animal numbers were estimated based on previous results from the transient ischaemia–reperfusion stroke model on extent and variability of atheroprogression after stroke. Data were excluded from all mice that died during surgery. Detailed exclusion criteria are described below. Animals were randomly assigned to treatment groups, and all analyses were performed by investigators blinded to group allocation. All animal experiments were performed and reported in accordance with the Animal Research: Reporting of In Vivo Experiments (ARRIVE) guidelines[35].

### Drug administrations

**Oral gavage with aspirin and rosuvasatin.** Mice received a daily bolus of aspirin (20 mg kg$^{-1}$, Sigma Aldrich) and rosuvastatin (5 mg kg$^{-1}$, Sigma-Aldrich) via oral gavage. Aspirin and rosuvastatin were solved in water (sterile ddH$_2$O) and mixed with a powdered chow diet (ssniff). A single daily bolus was 500 µl.

**Recombinant human DNase I.** $ApoE^{-/-}$ mice received DNase injections as we previously described[18]. In brief, 1,000 U recombinant DNase I (Roche) dissolved in incubation 1× buffer (40 mM Tris-HCl, 10 mM NaCl, 6 mM MgCl$_2$ and 1 mM CaCl$_2$, pH 7.9, diluted in PBS; Roche) was injected intravenously via tail vein right before surgery in a final volume of 100 µl. The control group was administered saline injections at the same volume, routine and timing as the experimental group.

**Caspase-1 inhibitor (VX765).** The caspase-1 inhibitor VX765 (stock in DMSO) was dissolved in PBS (Belnacasan, Invivogen) and injected intraperitoneally 1 h before surgery at a dose of 100 mg kg$^{-1}$ body weight at a final volume of 300 µl (ref. 18). The control group was administrated saline injections at the same volume, routine and timing as the experimental group.

**NLRP3 inflammasome inhibitor (MCC950).** Mice received two injections (1 h before and 1 h after surgery) of the NLRP3 inflammasome inhibitor MCC950 dissolved in sterile saline at a dose of 50 mg kg$^{-1}$ body weight (Invivogen). MCC950 or the control (sterile saline) was injected intraperitoneally in a final volume of 200 µl.

**AIM2 inhibitor (4-sulfonic calixarene).** The AIM2 inhibitor 4-sulfonic calixarene has recently been characterized by Green et al.[36]. The stock solutions (in DMSO) were dissolved in PBS and injected intraperitoneally 1 h before surgery at a dose of 5 mg kg$^{-1}$ body weight at a final

volume of 200 µl. The control group was administered control injections (sterile saline) at the same volume, routine and timing as for the intervention group.

**DNA challenge.** DNA derived from stimulated neutrophils (see below for neutrophil stimulation and isolation of NET–DNA) was injected intravenously at a dose of 5 µg at a final volume of 200 µl (sterile saline). The control group was administrated control injections (sterile saline) at the same volume, routine and timing as for the intervention group.

**Neutrophil depletion.** Neutrophils were depleted using a specific antibody to Ly6G (clone: 1A8, BioXCell). Mice were administered at a concentration of 14 mg kg$^{-1}$ body weight at a final volume of 250 µl intraperitoneally. The control group was administrated control IgG injections at the same volume, routine and timing as the experimental group.

**PAD4 inhibitor (GSK484).** The PAD4 inhibitor GSK484 (stock in 100% EtOH) was diluted in PBS to a final concentration of 4 mg kg$^{-1}$ body weight at a final volume of 100 µl. GSK484 was administered 1 h before surgery. For more than 6 h of survival time, a second bolus (4 mg kg$^{-1}$ body weight) was injected intraperitoneally 4 h post-surgery. The control group was administrated control injections (sterile saline) at the same volume, routine and timing.

**IL-1β neutralization.** Mice received two injections of antagonizing anti-IL-1β in sterile saline (clone: B122, BioXCell), 1 h before and 1 h after surgery. Anti-IL-1β or the corresponding IgG control was injected intraperitoneally at a dose of 4 mg kg$^{-1}$ body weight in a final volume of 200 µl.

### Patient cohorts for epidemiological analysis

The analysis presented in Fig. 1a was performed using patient data from two multicentre prospective observational hospital-based cohort studies of patients with acute ischaemic stroke in Germany (PROSCIS and DEMDAS/DEDEMAS). All patients provided written informed consent. Patient characteristics are provided in Supplementary Table 1. The study protocols and the detailed baseline patient characteristics have been previously described[13,14]. Basic demographic and stroke-related characteristics are summarized below. In brief, for both the PROSCIS and the DEMDAS/DEDEMAS cohorts, patients 18 years of age or older with an acute ischaemic stroke confirmed with neuroimaging and with symptom onset in the past 7 and 5 days, respectively, were recruited through the local stroke units of seven tertiary stroke centres in Germany. Patients in PROSCIS underwent follow-up telephone interviews at 3 and 12 months after stroke, whereas patients in DEMDAS/DEDEMAS underwent a telephone interview at 3 months after stroke as well as face-to-face interviews and inspection of their medical records by a physician at 6 and 12 months after stroke. The outcome of interest for the current analysis included the occurrence of a recurrent ischaemic stroke or transient ischaemic attack within the first 12 months after stroke, as self-reported by the patient and documented in their medical records.

### Patient cohort for carotid endarterectomy sample analysis

Patients scheduled for carotid endarterectomy due to symptomatic or asymptomatic carotid stenosis were prospectively recruited at the Department of Neurology and Cardiothoracic Transplantation and Vascular Surgery at Hannover Medical School between June 2018 and December 2020. Patients characteristics are provided in Supplementary Table 2. Carotid stenosis was defined as symptomatic if cerebral ischaemia occurred in the territory of the affected artery, and concurrent stroke aetiologies were excluded following standardized stroke diagnostics including cranial computed tomography

and/or MRI, computed tomography or MR-angiography, transthoracic or transoesophageal echocardiography, cardiac rhythm monitoring and Doppler/duplex ultrasound. Peripheral venous blood was drawn immediately before surgery, and EDTA whole-blood samples were used for flow cytometry analysis. Carotid plaque samples were obtained during carotid endarterectomy and immediately preserved in PBS. Both blood and tissue samples were sent for further analysis on the same day of collection. All patients provided written informed consent and the ethics committee at Hannover Medical School approved the study.

Thirteen patients with symptomatic and seven patients with asymptomatic, high-grade carotid stenosis were recruited. The median age was 73 years (25th–75th percentile: 62–80 years of age). See Supplementary Table 2 for clinical and demographic patient details.

### Patient cohort for myocardial infarction sample analysis
Patients with ST-elevation myocardial infarction (STEMI) were prospectively recruited between September 2016 and February 2018 at the German Heart Centre Munich and the Klinikum rechts der Isar (both at the Technical University of Munich). Patients characteristics are provided in Supplementary Table 3.

The diagnosis of STEMI was based on chest pain within the past 12 h, persistent ST-segment elevation of 1 mm or more in at least two extremities or 2 mm or more in at least two chest leads and diagnosis of type 1 myocardial infarction according to cardiac catheterization. Exclusion criteria were: cardiogenic shock, left ventricle ejection fraction of 35 or less, coexisting chronic or inflammatory diseases, anti-inflammatory drug therapy (for example, cortisol) and myocardial infarction type 2–5. Blood samples for plasma analysis were collected in EDTA tubes immediately after admission to the hospital or latest 6 h after coronary intervention.

Age-matched and sex-matched patients with known stable coronary artery disease served as controls. They were prospectively recruited between February 2017 and February 2018 during consultation in the outpatient department of the German Heart Centre Munich for routine examination. Exclusion criteria were: history of myocardial infarction, reduced left ventricle ejection fraction, chronic or inflammatory diseases, and anti-inflammatory drug therapy. Blood samples for plasma analysis were collected in EDTA tubes on the day of consultation in the outpatient department. All patients provided written informed consent and the institutional ethics committee at Technical University Munich approved the study (235/16S). EDTA tubes of both STEMI and control patients were centrifuged at 4 °C and 1,600g for 30 min directly after collection. Plasma aliquots were stored at −80 °C for further analysis.

### Carotid tandem stenosis model
Tandem stenosis surgery was performed as previously described[37]. In brief, at 12 weeks of age, 4 weeks after commencement of an HFD, $ApoE^{-/-}$ mice (C57BL/6J background) were anaesthetized with 2% isoflurane delivered in a mixture of 30% and 70% $N_2O$. An incision was made in the neck, and the right common carotid artery was dissected from circumferential connective tissues. To control the stenosis diameter, a 150-μm or 450-μm dummy was placed on top of the exposed right common carotid artery, with the distal point 1 mm away from the bifurcation and proximal point 3 mm from the distal stenosis; subsequently, a 6-0-braided polyester fibre suture was tied around both the artery and the needle, and then the pin was carefully removed. Animals were fed with an HFD for an additional 4 weeks after tandem stenosis surgery.

### Bone marrow transplantation
Donor animals (B6.129P2-$Aim2^{Gt(CSG445)Byg}$/J or C57BL/6J) were euthanized and the femur, tibia and humerus were collected in cold PBS. Bone marrows were isolated from bones and filtered through 40-μm cell strainers to obtain single-cell suspensions. After washing, cell number and viability were assessed using an automated cell counter (Countess 3, Thermo Fisher Scientific) and Trypan Blue solution (Merck). Cells were injected intravenously into $Mx1^{cre}$:$c$-$Myb^{fl/fl}$ recipient mice ($Ldlr^{-/-}$ background; 8–15 × 10⁶ cells per mouse) in a total volume of 100 μl saline. At the time of transplantation, recipient mice had previously been treated with poly(I:C) solution at a dose of 10 μg g$^{-1}$ body weight every other day five times to induce bone marrow depletion[38]. Mice were fed with a HFD and maintained for 4 weeks after transplantation to establish efficient bone marrow repopulation.

### Ischaemia–reperfusion stroke model
Four weeks after tandem stenosis surgery, $ApoE^{-/-}$ mice were anaesthetized with 2% isoflurane delivered in a mixture of 30% $O_2$ and 70% $N_2O$. An incision was made between the ear and the eye to expose the temporal bone. Mice were placed in supine position, and a laser Doppler probe was attached to the skull above the middle cerebral artery (MCA) territory. The common carotid artery and left external carotid artery were exposed via middle incision and further isolated and ligated. A 2-mm silicon-coated filament (Doccol) was inserted into the internal carotid artery, advanced gently to the MCA until resistance was felt, and occlusion was confirmed by a corresponding decrease in blood flow (that is, a decrease in the laser Doppler flow signal by 80% or more). After 60 min of occlusion, the animals were re-anaesthetized and the filament was removed. After recovery, the mice were kept in their home cage with ad libitum access to water and food. Sham-operated mice received the same surgical procedure, but the filament was removed in lieu of being advanced to the MCA. Body temperature was maintained at 37 °C throughout surgery in all mice via feedback-controlled heating pad. Exclusion criteria included: (1) insufficient MCA occlusion (a reduction in blood flow to more than 20% of the baseline value), (2) death during the surgery, and (3) lack of brain ischaemia as quantified post-mortem by histological analysis.

### Myocardial infarction
Myocardial infarction surgery was performed as previously described[39]. In brief, mice were intubated under MMF anaesthesia (midazolam 5.0 mg kg$^{-1}$ body weight, medetomidine hydrochloride 1.0 mg kg$^{-1}$ body weight and fentanyl citrate 0.05 mg kg$^{-1}$ body weight; intraperitoneally) and thoracotomy was performed in the left intercostal space. The left anterior descending coronary artery was identified and myocardial infarction was induced by permanent ligation with an 8-0 prolene suture. Atipamezole hydrochloride (5 mg kg$^{-1}$ body weight) and flumazenil (0.1 mg kg$^{-1}$ body weight) injected subcutaneously were used to antagonize MMF anaesthesia. Mice received subcutaneous buprenorphine (0.3 mg kg$^{-1}$ body weight) as an analgesic every 8 h for 3 days starting at the end of the surgical procedure.

### Ultrasound imaging (mouse carotid artery Doppler analysis)
Carotid artery blood flow in $ApoE^{-/-}$ mice was measured with a high-frequency ultrasound imaging system (Vevo 3100LT, Visual Sonics) with a 40-MHz linear array transducer (MX550D, Visual Sonics) before and right after tandem stenosis surgery, and weekly for 4 weeks afterwards. Mice were anaesthetized with isoflurane delivered in a mixture of 30% $O_2$ and 70% $N_2O$. B-mode, colour-Doppler mode and pulsed Doppler velocity spectrum were recorded from both sides of the CCA. For the right CCA (RCCA), five locations were examined: before proximal ligation, near proximal ligation, between two ligations, near distal ligation and above the distal ligations. For the left CCA (LCCA), as it was not ligated, only three locations were measured: proximal, middle and distal part of the LCCA. Pulsed Doppler velocity was determined with the sample volume calibrated to cover the entire vascular lumen and the smallest possible angle of interception (less than 60°) between the flow direction and the ultrasound beam. The peak systolic velocity (PSV) and end diastolic velocity (EDV) were recorded from CCAs of both sides.

VevoLab v3.2.0 software was used for ultrasound imaging analysis. The mean velocity was calculated as: mean velocity = (PSV + 2 × EDV)/3.

## MRI for secondary lesion detection

MRI was performed in a small-animal scanner (3T nanoScan PET/MR, Mediso, with 25-mm internal diameter quadrature mouse head coil) at 2 and 7 days after sham or stroke surgery. For scanning, mice were anaesthetized with 1.2% isoflurane in 30% $O_2$//70% $N_2O$ applied via face mask. Respiratory rate and body temperature (37 ± 0.5 °C) were continuously monitored via an abdominal pressure-sensitive pad and rectal probe, and anaesthesia was adjusted to keep them in a physiological range. The following sequences were obtained: coronal T2-weighted imaging (2D fast-spin echo (FSE), repetition time/echo time (TR/TE) = 3,000/57.1 ms, averages 14, resolution 167 × 100 × 500 µm$^3$), coronal T1-weighted imaging (2D FSE, TR/TE = 610/28.6 ms, averages 14, resolution 167 × 100 × 500 µm$^3$) and diffusion-weighted imaging (2D spin echo, TR/TE = 1,439/50 ms, averages 4, resolution 167 × 100 × 700 µm$^3$). MRI images were then post-processed using ImageJ.

## Microparticle of iron oxide MRI for thrombus detection

MRI was performed as previously described[24]. In brief, mice were anaesthetized using isoflurane in a mixture of $O_2$/$N_2O$ (30/70) and kept under anaesthesia during all the procedure, while maintaining a body temperature of 37 °C. Before MRI, mice were subjected to caudal vein catheterization for DNA and microsized matrix-based magnetic particle (M3P) administration. MRI was performed with a BioSpec 7T TEP-MRI system and a surface coil (Brukery). Imaging data were obtained using a TOF sequence to visualize vascular structures, a T2*-weighted sequence for iron-sensitive imaging and a T2-weighted sequence for tissue contrast. The MRI parameters were set at TR/TE = 12/4.2 ms for the TOF sequence, TR/TE = 50/8.6 ms for the T2*-weighted sequence and TR/TE = 3,500/40 ms for the T2-weighted sequence. T2*-weighted images are presented as a stack of four slices (minimum intensity) after bias fields correction using ImageJ software.

## Organ and tissue processing

Mice were deeply anaesthetized with ketamine (120 mg kg$^{-1}$) and xylazine (16 mg kg$^{-1}$) and venous blood was drawn via cardiac puncture of the right ventricle in 50 mM EDTA (Sigma-Aldrich); the plasma was isolated by centrifugation at 3,000$g$ for 15 min and stored at −80 °C until further use. Immediately following cardiac puncture, mice were transcardially perfused with ice-cold saline. Subsequently, the CCAs from both sides as well as the aortic arches and hearts were carefully isolated and embedded in compound (OCT, Tissue-tek), frozen over dry ice and stored at −80 °C until sectioning.

Heads were cut just above the shoulders. Skin was removed from the head and the muscle was stripped from the bone. After removal of the mandibles and the skull rostral to maxillae, the whole brain with skull was post-fixed by 4% paraformaldehyde (PFA) overnight at 4 °C. Subsequently, samples were transferred to a decalcification solution of 0.3 M EDTA (C. Roth, 292.94 g mol$^{-1}$) at pH 7.4 and stored at 4 °C. EDTA solution was changed after 3 days. Samples were immersed with 30% sucrose in PBS and then frozen down in −20 °C isopentane (Sigma-Aldrich). Coronal sections (15 µm thick) were obtained at the level of the anterior commissure for immunohistochemical analysis. Sections were mounted on SuperfrostPlus Slides (Thermo Scientific) and stored in −80 °C.

## Preparation of plasma samples for free nucleotide quantification

Mouse venous blood from cardiac puncture s drawn in 50 mM EDTA tubes. Afterwards samples were centrifuged at 1,500$g$ for 10 min at room temperature. Plasma was isolated, transferred to a new tube and spined again at 3,000$g$ for 10 min. Plasma was then carefully collected and immediately frozen down at −80 °C until further processing.

## Histology and immunofluorescence

Carotid (5 µm) cryosections were histologically stained with haematoxylin and eosin (H&E) in 100-µm intervals. Total collagen content was assessed by Picro Sirius red staining (Abcam) in consecutive sections. For immunofluorescence staining, cryosections were fixed with 4% PFA followed by antigen blockade using 2% goat serum-blocking buffer containing 1% BSA (Sigma), 0.1% cold fish skin gelatin (Sigma-Aldrich), 0.1% Triton X-100 (Sigma) and 0.05% Tween-20 (Sigma). Next, sections were incubated overnight at 4 °C with the following primary antibodies: rat anti-mouse CD68 (1:200; ab53444, Abcam), mouse anti-mouse smooth muscle actin (1:200; ab7817, Abcam), rabbit anti-mouse Ki67 (1:200; 9129S, Cell Signaling), mouse anti-mouse caspase-1 (1:200; AG-20B-0042-C100, Adipogen), sheep anti-von Willebrand factor (1:100; ab11713, Abcam), rat anti-CD31 (1:200; BM4086, OriGene) and anti-collagen I (1:250; ab279711, Abcam). After washing, sections were incubated with secondary antibodies as following: AF647 goat anti-rat (1:200; Invitrogen), Cy3 goat anti-mouse IgG H&L (1:200; Abcam), AF594 goat anti-rabbit (1:200; Invitrogen), AF488 goat anti-mouse (1:200; Invitrogen) and AF594 donkey anti-sheep (1:200; Invitrogen). Counterstain to visualize nuclei was performed by incubating with DAPI (1:5,000; Invitrogen). Finally, sections were mounted with fluoromount medium (Sigma-Aldrich). Microphotographs of immunofluorescent samples were taken with a confocal microscope (LSM 880 and LSM 980; Carl Zeiss) using ZEN2 software (blue edition). Histological sections were imaged with an epifluorescent microscope (Axio Imager M2, Carl Zeiss) and quantified by using ImageJ software (US National Institutes of Health).

For the visualization of suspected secondary infarct lesions in the contralateral hemisphere, brain sections (15 µm) were first stained for Fluoro Jade C (FJC) to identify degenerating neurons. FJC staining was performed using the Fluoro-Jade C Ready-to-Dilute staining kit (TR-100-FJ, Biosensis) according to the manufacturer's instructions. To confirm the secondary lesion, double staining of the microglia marker Iba-1 (1:200; FUJIFILM Wako Pure Chemical Corporation) and terminal deoxynucleotidyl transferase dUTP nick end labelling (TUNEL) staining was performed using the Click-iT Plus TUNEL Assay for In Situ Apoptosis Detection (Alexa Fluor 647 dye, C10619, Thermo Scientific) according to the manufacturer's instructions. Brain samples were photographed on an epifluorescence microscope (Zeiss Axiovert 200M, Carl Zeiss) and a confocal microscope (LSM880, Carl Zeiss).

## Mouse CCA plaque analysis

Plaque vulnerability was assessed as previously described[15]. In brief, intima, media and necrotic core area were analysed in H&E-stained sections. The necrotic core was defined as the area devoid of nuclei underneath a formed fibrous cap. Collagen content was quantified on Picro Sirius red-stained sections. The vulnerability plaque index (VPI) was calculated as VPI = (% necrotic core area + % CD68 area)/(% smooth muscle actin area + % collagen area).

## Flow cytometry analysis

Isolated CCA samples were mixed with digestion buffer, consisting of collagenase type XI (125 U ml$^{-1}$, C7657), hyaluronidase type 1-s (60 U ml$^{-1}$, H3506), DNase I (60 U ml$^{-1}$, D5319), collagenase type I (450 U ml$^{-1}$, C0130; all enzymes from Sigma-Aldrich) in 1× PBS[40], and were digested at 750 rpm for 30 min at 37 °C. After digestion, CCA materials were homogenized through a 40-µm cell strainer, washed at 500$g$ for 7 min at 4 °C and resuspended in flow cytometry staining buffer (00-4222-26, Thermo Scientific) to generate single-cell suspensions. Cell suspensions were incubated with flow cytometry antibodies and analysed using a spectral flow cytometer (Northern Light, Cytek). Alternatively, cell suspension was incubated with the fluorescent inhibitor probe 660-YVAD-FMK (#9122, Immunochemistry Technology) to label active caspase-1 in living cells. A detailed

antibody list for flow cytometry analysis is available in the resource table in Supplementary Table 6.

For neutrophil flow cytometry, CCA single-cell suspensions (see above) or full EDTA blood was incubated with CD45-specific, CD11b-specific, Ly6C-specific, Ly6G-specific and citrullinated histone3 (citH3)-specific antibodies and analysed using spectral flow cytometry. Neutrophils were defined as $CD45^+CD11b^+Ly6G^{high}$ cells; neutrophil activation was defined via citH3 detection.

Representative gating strategies for individual flow cytometry experiments are provided in Supplementary Fig. 2.

### Analysis of plaque-infiltrating leukocytes

Circulating leukocytes were discriminated by intravenous administration of an anti-CD45 antibody[41] (eFluor450, clone: 30-F11, eBioscience) immediately before stroke surgery. Twenty-four hours after stroke, to exclude the blood contamination in the CCA, an anti-CD45 antibody (APC–Cy7, clone: 30-F11, BioLegend) was injected intravenously 3 min before euthanasia. Then, CD45 eFluor450-positive but APC–Cy7-negative population were considered as the 'infiltrating leukocytes' population.

### Immunoblot analysis

Ipsilateral and contralateral CCA materials were carefully isolated and snap frozen on dry ice. Whole frozen CCA was lysed with RIPA lysis/extraction buffer with added protease/phosphatase inhibitor (Thermo Fisher Scientific). Total protein was quantified using the Pierce BCA protein assay kit (Thermo Fisher Scientific). Whole-tissue lysates were fractionated by SDS–PAGE and transferred onto a polyvinylidene difluoride membrane (Bio-Rad). After blocking for 1 h in TBS-T (TBS with 0.1% Tween-20, pH 8.0) containing 4% skin mile powder (Sigma), the membrane was washed with TBS-T and incubated with the primary antibodies to the following antibodies: mouse anti-caspase-1 (1:1,000; Adipogen), rabbit anti-actin (1:1,000; Sigma) and rabbit anti-factor XII (1:1,000, Invitrogen). Membranes were washed three times with TBS-T and incubated for 1 h with horseradish peroxidase-conjugated anti-rabbit or anti-mouse secondary antibodies (1:5,000, Dako) at room temperature. Membranes were washed three times with TBS-T, developed using ECL substrate (Millipore) and acquired via the Vilber Fusion Fx7 imaging system.

### ASC oligomerization assay

ASC oligomerization was performed as previously published[42]. In detail, bone marrow was isolated from mice and cultured for 7 days with L929 cell-conditioned medium. After differentiation, cells were washed in PBS and primed with 100 ng ml$^{-1}$ of lipopolysaccharide (LPS) for 4 h. Cells were either left primed or additionally activated by stimulation with 250 µg ml$^{-1}$ NET–DNA for 2 h. Cells were washed in PBS and detached by scraping in PBS containing 2 mM EDTA. After centrifugation, cell pellets were resuspended in 0.5 ml of ice-cold buffer A and lysed through sonication. After centrifugation to remove bulk nuclei, 20 µl of lysate was stored as input before oligomerization. Buffer A was added to the remaining lysate; following centrifugation, supernatants were diluted with CHAPS buffer and pelleted through further centrifugation. Proteins were crosslinked for 30 min at room temperature with 50 µl of CHAPS buffer containing 4 mM disuccinimidyl suberate. After centrifugation pellets were resuspended in Laemmli buffer and boiled at 70 °C for 10 min. Samples were loaded onto SDS–PAGE and separated at 150 V. Transfer was performed at 4 °C and 100 V for 1 h and membranes were blocked with 4% BSA in TBS-T. Membranes were incubated overnight at 4 °C with anti-ASC antibody (AL177, Adipogen), washed three times in TBS-T and incubated at room temperature for 1 h with goat anti-Rb horseradish peroxidase antibody. After washing and detecting the images, membranes were re-incubated with actin–antibody for 1 h at room temperature, proceeded by washing, secondary antibody and imaging.

### ELISA

CCA tissue samples were carefully isolated and snap-frozen on dry ice. Whole frozen CCA was lysed with cell lysis buffer (#895347, R&D System). Then, the concentration of IL-1β in total CCA lysates was measured by ELISA according to the manufacturer's instructions (MLB00C, R&D system). Absorbance at 450 nm was measured by an iMark Microplate reader (Bio-Rad).

### Gelatin zymography of mouse CCA extracts, BMDMs culture medium and patient plaque lysates

CCA tissue extracts were analysed using gelatin zymography (Novex TM 10% zymography plus protein; ZY00100BOX, Thermo Scientific) according to the manufacturer's instructions. CCA tissue was lysed with cell lysis buffer (#895347, R&D System). Total protein was quantified using a Pierce BCA protein assay kit (Thermo Fisher Scientific). Aliquots of appropriately diluted tissue extracts were loaded on gels at a total volume of 20 µl. After electrophoresis, gels were incubated in 1× Zymogram renaturing buffer (LC2670, Invitrogen) for 30 min at room temperature with gentle agitation. Afterwards, Zymogram renaturing buffer was decanted and 1× Zymogram developing buffer (LC2671, Invitrogen) was added to the gel. The gel was then equilibrated for 30 min at room temperature with gentle agitation. After an additional wash with 1× Zymogram developing buffer, the gels were incubated at 37 °C overnight. Gels were stained with a colloidal blue staining kit (LC6025, Invitrogen) and acquired on a gel scanner. BMDM culture medium was collected and loaded on gelatin zymography gels at a total volume of 25 µl. MMP activity in BMDM culture medium was analysed using the same protocol as for the tissue samples.

### MMP2 and MMP9 in situ zymography

MMP2 and MMP9 in situ zymography on CCA sections was performed as previously described with slight modifications[8]. DQ-gelatin (D12054, Invitrogen) was dissolved in reaction buffer (50 mM Tris-HCl, 150 mM NaCl, 5 mM CaCl$_2$ and 200 mM sodium azide, pH 7.6). Cryosections (5 µm) were incubated for 2 h at 37 °C with the gelatin-containing reaction buffer. Negative control sections were pre-incubated for 1 h with the MMP inhibitor 1,10-phenatheroline (Sigma). Nuclei were stained with DAPI. MMP activity was detected with an Axio Observer Z1 microscope with ×20 magnification (Carl Zeiss). Data are shown as normalized MMP intensity (normalized MMP area = MMP$^+$ area/intima area).

### Neutrophil isolation and stimulation

Neutrophils were generated from the tibia and femur of transcardially perfused WT mice. After isolation and dissection of the tibia and femur, bone marrow was flushed out of the bones through a 40-µm strainer using a plunger and 1-ml syringe filled with sterile 1× PBS. Strained bone marrows cells were washed with PBS and resuspended in 1× sterile PBS with 5% BSA. Afterwards, neutrophils were isolated by using a neutrophil isolation kit (130-097-658, Miltenyi) according to the manufacturer's instructions. Of cells, $1 \times 10^7$ were plated onto 150-mm culture dishes in RIPA 1640 (Gibco), supplemented with 10% FBS and 1% penicillin–streptomycin.

For the generation of NET–DNA, $2 \times 10^7$ cells (for isolation, see above) were cultured in a 150-mm tissue culture-treated dish. Cells were then stimulated with 100 nM phorbol 12-myristate 13-acetate (PMA) overnight. The next day, supernatant was removed from the culture dish and processed in a centrifugation protocol for isolating NET–DNA. Cell culture supernatants were centrifuged at 500$g$ for 10 min, then the supernatant was kept and centrifuged again at 15,000$g$ for 10 min. Supernatant was decanted and the pellet (NET–DNA) was resolved in 50 µl of nuclease-free water. NET–DNA was then labelled with the fluorescent DNA probe DRAQ5 (Alexa 647; Thermo Fisher) following the manufacturer's instructions.

For ex vivo serum stimulation of neutrophils, $3 \times 10^5$ cells were cultured in 12-well plates. Cells were then treated with either 25% serum from stroke-operated or sham-operated mice for 4 h. As a positive control, 10 nM PMA was used. Cells were then washed and immediately fixed with 3.7% PFA/sucrose. Fixed neutrophils were then stained with citH3-specific antibody and NET–DNA was counterstained with 1 μM SYTOX green.

## cfDNA isolation from human and mouse plasma

Mouse venous blood from cardiac puncture was drawn in 50 mM EDTA 2-ml collection tubes. Samples rested at a maximum of 15 min at room temperature on the bench. Afterwards, samples were centrifuged at 3,000g for 10 min at 4 °C. Plasma was isolated, transferred to a new tube and spined again at 3,000g for 10 min. Plasma was then carefully collected and immediately frozen down at −80 °C until further processing. We used 500 μl plasma to isolate cfDNA with a column-based kit (Plasma/Serum Cell-Free Circulating DNA Purification Kit; 55100; Norgen Biotek). In the last step, DNA was eluted with 30 μl buffer from the column. Afterwards, total circulating DNA and single-stranded DNA were measured with a Nanodrop Spectrophotometer (1000ND, Peqlab). Double-stranded DNA (dsDNA) concentrations were acquired with a Qubit 2.0 fluorophotometer (Invitrogen) using a specific fluorescent dye-binding dsDNA (HS dsDNA Assay kit, Thermo Fisher Scientific). Dilutions and standards were generated following the manufacturer's instructions.

Length distribution of circulating cfDNA fragments after DNA isolation was acquired using the automated gel electrophoresis platform Bioanalyzer (Agilent) and the High Sensitivity DNA kit (Agilent). Data were analysed using the Bioanalyzer 2100 Expert software (Bioanalyzer, Agilent). Data in Fig. 2h (mouse, 2–72 h after stroke), Fig. 3g (mouse, 6 h after stroke), Extended Data Fig. 7d (mouse, 24 h after stroke), Extended Data Fig. 7e (mouse, 12 or 24 h after myocardial infarction) and Extended Data Fig. 8a (intravenous DNA measured 24 h after injection) were generated following the above protocol.

Human samples from patients with asymptomatic/symptomatic CCA (stroke (human; CCA sample)) were acquired at Hannover Medical School (see 'Patient cohort for carotid endarterectomy sample analysis'). Full-blood samples (blood withdrawal in EDTA collection tubes) were transported to Munich (180 min or longer transportation time on 4 °C). Once arrived, 3 ml of each sample was then centrifuged at 3,000g for 15 min, and plasma was collected and stored at −80 °C. We used 500 μl plasma to isolate cfDNA with a column-based kit (Plasma/Serum Cell-Free Circulating DNA Purification Kit; 55100, Norgen Biotek). In the last step, DNA was eluted with 30 μl buffer from the column. Details of the cfDNA isolation protocols are in Supplementary Table 4.

Human samples acquired at the medical centre of LMU (Munich, Germany; 'stroke (human; cfDNA methylation)') and at the medical centre of Technical university (Munich, Germany; 'MI (human)') were collected in EDTA containing tubes and centrifuged after 15–30 min at room temperature. Human myocardial infarction samples were centrifuged at 1,600g for 30 min, and plasma was collected and stored at −80 °C until cfDNA was isolated. Human stroke samples were centrifuged at 1,500g for 10 min, and plasma was collected and stored at −80 °C until further processing.

## Extracellular vesicle spin down

Mouse plasma samples were diluted 1:1 with nuclease-free water. Dilution was centrifuged at 16,500g for 20 min, and then supernatant was filtered through 0.22-μm filter. The filtered supernatant was then transferred to an ultracentrifuge and spined at 110,000g for 60 min. Supernatant was kept as 'extravesicular' DNA. The pellet was resuspended and 'vesicular' DNA concentration was measured.

## Human cfDNA methylation pattern analysis

We collected platelet-poor plasma from 17 patients with ischaemic stroke (median of 73.9 years of age, interquartile range of 65.8–87.2 years of age; 10 women (58.8%)) upon hospital admission in the emergency department before any acute treatment or intervention (time from symptom onset to sampling: median of 6.5 h, interquartile range of 2.2–8.3 h). Patients had a median infarct volume of 57.4 ml (interquartile range of 40.5–124.0 ml).

cfDNA extracted from plasma was treated with bisulfite to convert unmethylated cytosines to uracils, and amplified using a two-step PCR protocol as previously described[43], followed by next-generation sequencing. Sequencing results were analysed to determine the proportion of molecules from each locus that carries the methylation pattern of the cell type of interest (typically complete demethylation). We used a cocktail of brain markers (methylation markers of neurons, oligodendrocytes and astrocytes) as previously described[44] and another cocktail that amplifies the markers of immune and inflammatory cells: neutrophils, monocytes, eosinophils, and T cells and B cells[45]. The raw values (GE ml$^{-1}$ and percentage) of DNA methylation analysis are in Supplementary Table 5.

## BMDM isolation and cell culture

BMDMs were isolated and cultured as previously described[18]. In brief, BMDMs were generated from the tibia and femur of transcardially perfused WT mice. After careful isolation and dissection of the tibia and femur, bone marrow was flushed out of the bones through a 40-μm strainer using a plunger and 1-ml syringe filled with sterile 1× PBS. Strained bone marrow cells were washed with PBS and resuspended in DMEM + GlutaMAX-1 (Gibco), supplemented with 10% FBS and 1% gentamycin (Thermo Scientific) and counted. Of cells, $5 \times 10^7$ were plated onto 150-mm culture dishes. Cells were differentiated into BMDMs over the course of 8–10 days. For the first days after isolation, cells were supplemented with 20% L929 cell-conditioned medium, as a source of M-CSF. Cultures were then maintained at 37 °C with 5% CO$_2$ until they reached 90% or more confluence.

## BMDM stimulation with sham or stroke serum

BMDMs were cultured for 8–10 days for full differentiation. Cells were then harvested, washed, counted and seeded in flat-bottom tissue culture-treated 24-well plates at a density of $2 \times 10^5$ cells per well in a total volume of 500 μl, and then cultured overnight for at least 16 h. BMDMs were then stimulated with LPS (100 ng ml$^{-1}$) for 4 h. Afterwards, the cells were incubated with serum from either stroke-operated or sham-operated WT mice (4 h post-surgery) at a concentration of 25% total volume for 1 h. Control-treated BMDMs received only FBS-containing culture medium. Afterwards, cell lysates were collected in RIPA buffer and stored at −80 °C until further processing.

For MMP secretion, the supernatant was discarded after stimulation and the cells were washed with sterile PBS to ensure no leftover serum on the cells. Afterwards, 500 μl serum-free DMEM was added to the BMDMs, which were then incubated overnight for 16 h at 37 °C and 5% CO$_2$. The culture medium was then collected for further analysis.

## BMDMs stimulation with NET–DNA

ASC–citrine (B6.Cg-Gt(ROSA)26Sor$^{tm1.1(CAG-Pycard/mCitrine*,-CD2*)Dtg}$/J) BMDMs were cultured as described above. Cells were then harvested and seeded in a 12-well flat-bottom well plates equipped with 15-mm coverslips. A total of $3 \times 10^5$ cells were seeded per well in a total volume of 1 ml and then cultured overnight for at least 16 h (37 °C and 5% CO$_2$). BMDMs were then primed with LPS (100 ng ml$^{-1}$) for 4 h. Afterwards, BMDMs were stimulated with 250 ng of DRAQ5-labelled NET–DNA (see the section 'Neutrophil isolation and stimulation') for 1 h. BMDMs were then washed with PBS and fixed with 3.7% PFA/sucrose. Coverslips containing BMDMs were then removed from the well plates, cytoskeleton was

stained for β-tubulin (Thermo Fisher) and nuclei were counterstained with Hoechst 33342 (Immunohistochemistry.com).

### En face immunofluorescence staining
Both ipsilateral and contralateral CCAs were carefully dissected, and adventitial fat and ligation nodes were thoroughly trimmed away. CCAs were then cut open, unfolded and pinned out on a silicon-elastomer for fixation in 4% PFA at room temperature for 2 h. The CCAs were then washed for 1 h at room temperature in 5% BSA with 0.3% Triton X-100 (Sigma-Aldrich). Afterwards, CCAs were incubated with rabbit anti-factor XII (1:100; PA5-116703, Invitrogen) at 4 °C overnight. After washing in 5% BSA with 0.3% Triton X-100 for 1 h at room temperature, CCAs were incubated with AF647 goat anti-rabbit (1:100; Invitrogen) and DAPI for 2 h at room temperature. Finally, CCAs were mounted with fluoromount medium (Sigma-Aldrich). Microphotographs were taken with a confocal microscope (LSM 980, Carl Zeiss).

### Aspect ratio of collagen I
For assessment of the collagen structural organization at the fibrous cap, 20X PSR images from 3 to 4 sequential segments were taken. Quantitative analysis using ImageJ software was done as previously described[46,47]. In brief, fast Fourier transformation was performed on the approximately 40-μm subendothelial fibrous cap area. Thresholded fast Fourier transformation images then underwent an elliptic fit and the aspect ratio value was calculated, as a measure of collagen fibre distribution anisotropy at the fibrous cap region.

### Statistical analysis and reproducibility
Data were analysed using GraphPad Prism version 6.0. All summary data were expressed as the mean ± standard deviation unless indicated otherwise. All datasets were tested for normality using the Shapiro–Wilk normality test. The groups containing normally distributed independent data were analysed using a two-way Student's $t$-test (for two groups) or ANOVA (for more than two groups). Normally distributed dependent data were analysed using a two-way ANOVA. The remaining data were analysed using the Mann–Whitney $U$-test (for two groups) or Kruskal–Wallis test ($H$-test; for more than two groups). $P$ values were adjusted for comparison of multiple comparisons using Bonferroni correction or Dunn's multiple comparison tests. $P < 0.05$ was considered to be statistically significant.

$t$-Distributed stochastic neighbour embedding for anti-Ly6G-depleted bone marrow (Extended Data Fig. 8f) was performed using the integrated tSNE platform of FlowJo (Treestar). One thousand CD45[+] cells were used per sample. The plugin was set to 250 iterations and a perplexity value of 30.

Principal component analysis (Extended Data Fig. 3f) was performed using Rstudio version 1.1.477. Absolute values from the necrotic core, smooth muscle actin, collagen, fibrous cap thickness and CD68[+] macrophage area were $Z$-scored and then principal components were calculated using the 'prcomp()' command (built-in R stats package). The arrows represent the variable correlation showing the relationship between all variables. Principal components were picked by their percentage of explained variance (62.73% (PC1) and 21.05% (PC2)) and visualized using the 'ggplot2' package (version 3.4.3; https://ggplot2.tidyverse.org). Relative contribution and quality of representation were calculated and visualized using the 'corrplot' package (version 0.92; https://github.com/taiyun/corrplot).

All in vivo experiments were performed in 3–5 independent experiments. Within one independent experiment, all groups were represented. All in vitro experiments were performed in 2–4 independent experiments. All conditions of the in vitro assays were represented in each independent experiment. Experiment data shown for Fig. 1d,i represent three independent experiments. Experiment data shown in Fig. 2c,e represent three independent experiments. Representative images in Fig. 2f,i,k,m show the outcome of one of three independent experiments. Representative microphotographs in Fig. 3c,e show representative images from experiments that were independently repeated at least five times.

### Reporting summary
Further information on research design is available in the Nature Portfolio Reporting Summary linked to this article.

## Data availability
Further information, data and requests for resources or reagents should be directed to and will be fulfilled by the corresponding authors.

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

**Acknowledgements** We thank K. Thuß-Silczak for excellent technical support; Q. Zhou for support with ultrasound analysis; U. Schillinger and R. Fang for support in establishing the small-animal MRI analysis; O. Söhnlein for critical input in study design and establishing the tandem stenosis model; and R. D. Stauss, C. Schrimpf and M. Wilhelmi for their support in collecting human carotid plaque samples. This work was funded by the Vascular Dementia Research Foundation, the European Research Council (ERC-StGs 802305 and 759272), the 'Else-Kröner-Fresenius-Stiftung' (2020_EKSE.07), the China Scholarship Council, the Medical Research Council grant MR/T016515/1, and the German Research Foundation (DFG) under Germany's Excellence Strategy (EXC 2145 SyNergy ID 390857198 and EXC-2049 ID 390688087), through FOR 2879 (ID 405358801), CRC 1123 (ID 238187445), TRR 295 (ID 424778381), TRR 332 (ID 449437943), TRR 355 (ID 490846870) and by grant ME 3696/3-1.

**Author contributions** J.C. and S.R. performed most of the experiments, analysed the data and contributed to the writing of the manuscript. G.M.G. provided the human carotid endarterectomy samples. M.G., A.K., T.L. and S.W. acquired and analysed the clinical epidemiological data. A.H., R.S. and Y.D. performed the DNA methylome analysis on human samples, which were collected and analysed by S.T. D.M. and C.S. performed the bone marrow transfer experiments. C.J., A.P. and D.V. performed the in vivo MRI. V.H. provided AIM2-deficient, NLRP1-deficient and NLRP3-deficient animals. S.Z., C.F., X.L., O.C., J.Z., S.M., C.S. and Y.A. performed the experiments and analysed the data. D.B., J.P.G., H.S. and M.D. contributed critical material and techniques for this study. H.B.S. and M.D. contributed critical input to study design and manuscript writing. A.L. initiated and coordinated the study, analysed the data and wrote the manuscript.

**Competing interests** The authors declare no competing interests.

**Additional information**
**Correspondence and requests for materials** should be addressed to Stefan Roth or Arthur Liesz.

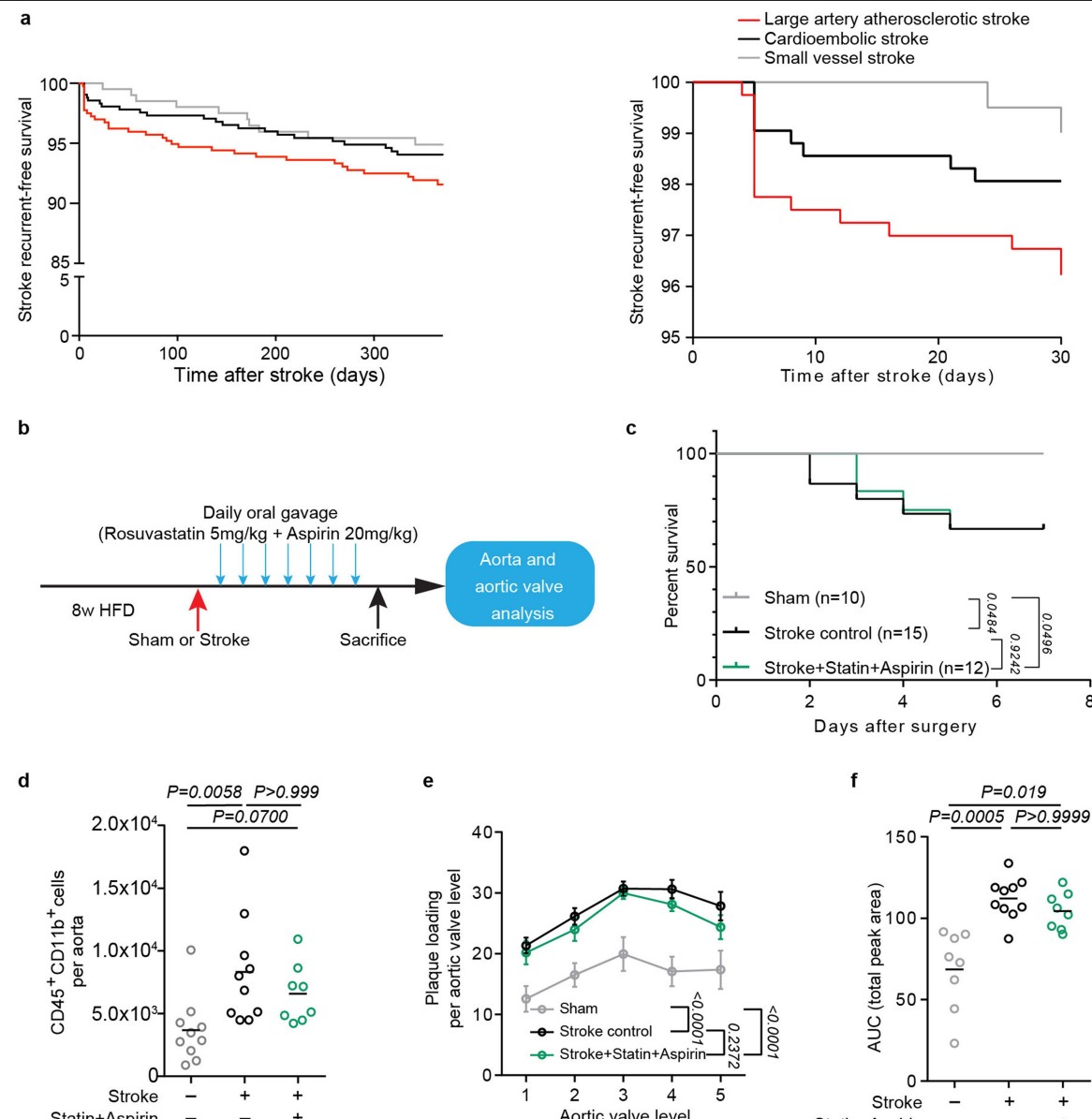

**Extended Data Fig. 1 | Established secondary prevention fails to attenuate early post-stroke vascular inflammation and atheroprogression. a.** Kaplan-Meier curves for recurrence-free survival of stroke patients from the same cohorts (PROSCIS and DEMDAS/DEDEMAS) as shown in Fig. 1a. Recurrence-free survival is illustrated by etiology of the incident stroke for the full-time range of 1 year after the incident event (left) and magnified for the first post-stroke month (right). Recurrence risk is highest after large artery atherosclerosis stroke in the early (first week) after the incident event. **b.** Experimental design: 8 w-old HFD fed *ApoE*−/− mice underwent sham or stroke surgery. Mice were treated orally with either control or a combination of Rosuvastatin (5 mg/kg)

and Aspirin (20 mg/kg) for 7 consecutive days after stroke. **c.** Kaplan-Meier survival curves of stroke control, statin and aspirin treated, or sham operated mice. Mantel-Cox test; n = 10 (sham), n = 15 (control), n = 12 (statin + aspirin treatment). **d.** Flow cytometry analysis of whole aorta cell suspensions for total monocyte (CD45+CD11b+) cell counts between control or treated mice after stroke compared to sham-operated mice (ANOVA, n = 8–10 per group). **e, f.** Quantification of aortic valve plaque load displayed no differences between stroke control and statin + aspirin-treated mice. Data is shown as **e.** percentage of plaque area per aortic valve level and **f.** area under the curve (ANOVA, n = 8–10 per group).

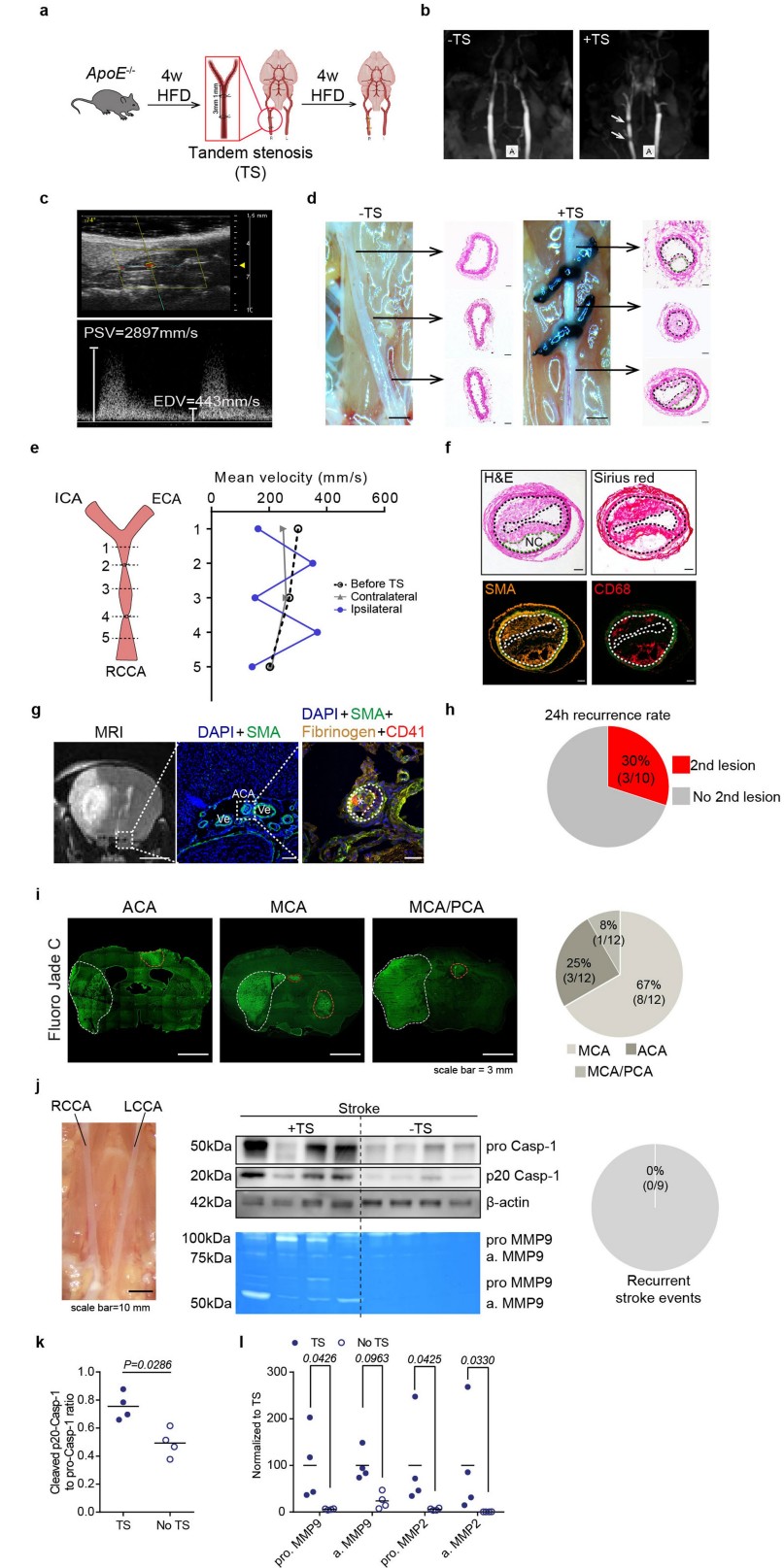

**Extended Data Fig. 2 |** See next page for caption.

**Extended Data Fig. 2 | Animal model of rupture-prone CCA plaques.**
**a**. Schematic illustration of the tandem stenosis (TS) model for induction of vulnerable atherosclerotic plaques: 8 w-old HFD fed $ApoE^{-/-}$ mice received tandem stenosis (TS) surgery on the right common carotid artery (RCCA). Mice were then fed with high fat diet for an additional 4 w. **b**. Representative images of CCA MRI TOF sequence 4w after mice received control or TS surgery. White arrows highlight the two ligations on the RCCA. **c**. Representative pulse-wave mode ultrasound image of the RCCA 4 w after TS, imaged at the location of proximal ligation at 40 MHz (upper panel). Corresponding CCA velocity waveform measured at the location of proximal ligation location 4 w after TS surgery (lower panel). PSV: peak systolic velocity, EDV: end diastolic velocity. **d**. Representative photograph of the CCA anatomy and corresponding H&E staining for each vessel segment 4 w after TS surgery of both CCAs (-TS represents contralateral (left) CCA without TS ligation; +TS represents ipsilateral (right) CCA with TS ligation, scale bar =100 μm; H&E staining, scale bar = 50 μm). **e**. Schematic description of locations for blood flow measurement on the right CCA (ICA: internal carotid artery; ECA: external carotid artery; RCCA: right common carotid artery). Corresponding quantification of mean velocity in the CCA measured at both CCAs before and 4 w after TS surgery (right). **f**. Representative images of the unstable plaque in the right CCA 4 w after TS surgery (area between two dotted lines indicates intima area, green dotted line

indicates necrotic core, scale bar = 50 μm). **g**. Representative T2 MRI image (left) and immunohistochemistry of a thrombus formation stained for smooth muscle actin (SMA), Fibrinogen and thrombocytes (CD41) in the ACA territory. **h**. 8-week-old $ApoE^{-/-}$ mice fed a high cholesterol diet (HCD-fed $ApoE^{-/-}$) underwent tandem stenosis (TS) surgery, and stroke surgery 4w later. The recurrence of secondary ischemia in the contralateral hemisphere was examined by histological analysis 24 h after stroke surgery (n = 10 per group). **i**. Analysis of the vessel territory of secondary ischemic events in all mice from Fig. 1c. Vessel territories were defined as MCA, ACA or MCA/PCA territory (n = 12 per group). **j**. HCD-fed $ApoE^{-/-}$ mice did not undergo tandem stenosis (TS) surgery, but stroke surgery 4w later. Left: Representative microphotograph of the RCCA and LCCA without TS. Middle: Western blot analysis of caspase-1 cleavage and MMP2/9 zymography in the CCA without TS. Right: Detection of recurrent ischemic events in mice without TS, but stroke (n = 9 per group). Raw membrane images of immunoblots and zymography images with cropping indication can be found in Supplementary information 1. **k**. Quantification of cleaved caspase-1 p20 intensity normalized to b-actin in CCA lysates with or without TS surgery (U test, n = 4 per group). **l**. Quantification of MMP2 and MMP9 (pro- and active form) normalized to TS surgery in CCA lysates from mice with or without TS surgery (n = 4 per group; K test).

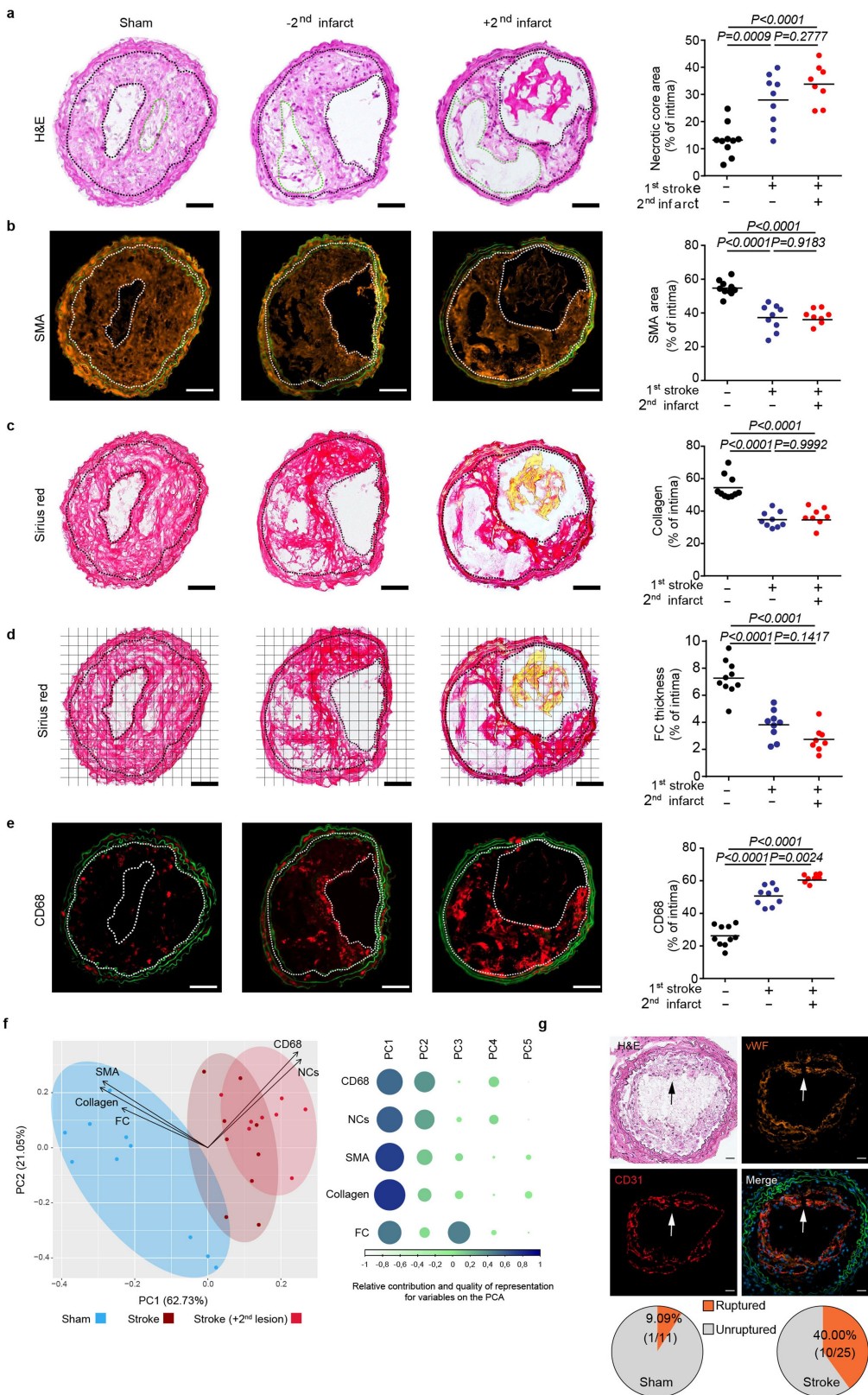

**Extended Data Fig. 3** | See next page for caption.

**Extended Data Fig. 3 | Stroke accelerates plaque destabilization and causes plaque rupture. a**. Representative microphotographs of CCA H&E staining. Area between two black dotted lines represents intima. Green dotted lines represent necrotic cores. Necrotic core area was quantified as the percentage of total intima area (ANOVA, n = 8–10 per group). **b**. Representative images of smooth muscle actin (SMA) immunofluorescence staining. SMA area was quantified as the percentage of total intima area (ANOVA, n = 8–10 per group). **c**. Representative microphotographs of Picro-Sirius red staining. Collagen area was quantified as the percentage of total intima area (ANOVA, n = 8–10 per group). **d**. Representative images of Picro-Sirus red stained CCA sections with the according grid for fibrous cap (FC) thickness quantification. FC thickness was quantified at the locations were the FC crossed the applied grid (ANOVA, n = 8–10 per group). **e**. Representative microphotographs of CD68 immunofluorescence staining. Images were segmented by thresholding to convert fluorescence signal into a binary image. CD68 area was quantified as the percentage of total intima area (ANOVA, n = 8–10 per group). **f**. Left: Principal component analysis (PCA) using CCA plaque vulnerability readouts from sham, stroke and stroke mice with detected secondary lesion found in **a**. to **e**. (n = 8–10 per group) Right: Contribution of the Plaque vulnerability readouts in principal component 1 to 5 (PC1 to PC5), weighted for their relative quality of representation in the PCA. **g**. Arrows indicate a disrupted fibrous cap in lesion. The pie charts illustrate the proportion of mice with ruptured CCA plaques 1 w after sham or stroke surgery (n = 11 or 25 mice per group)(all scale bars = 50 µm).

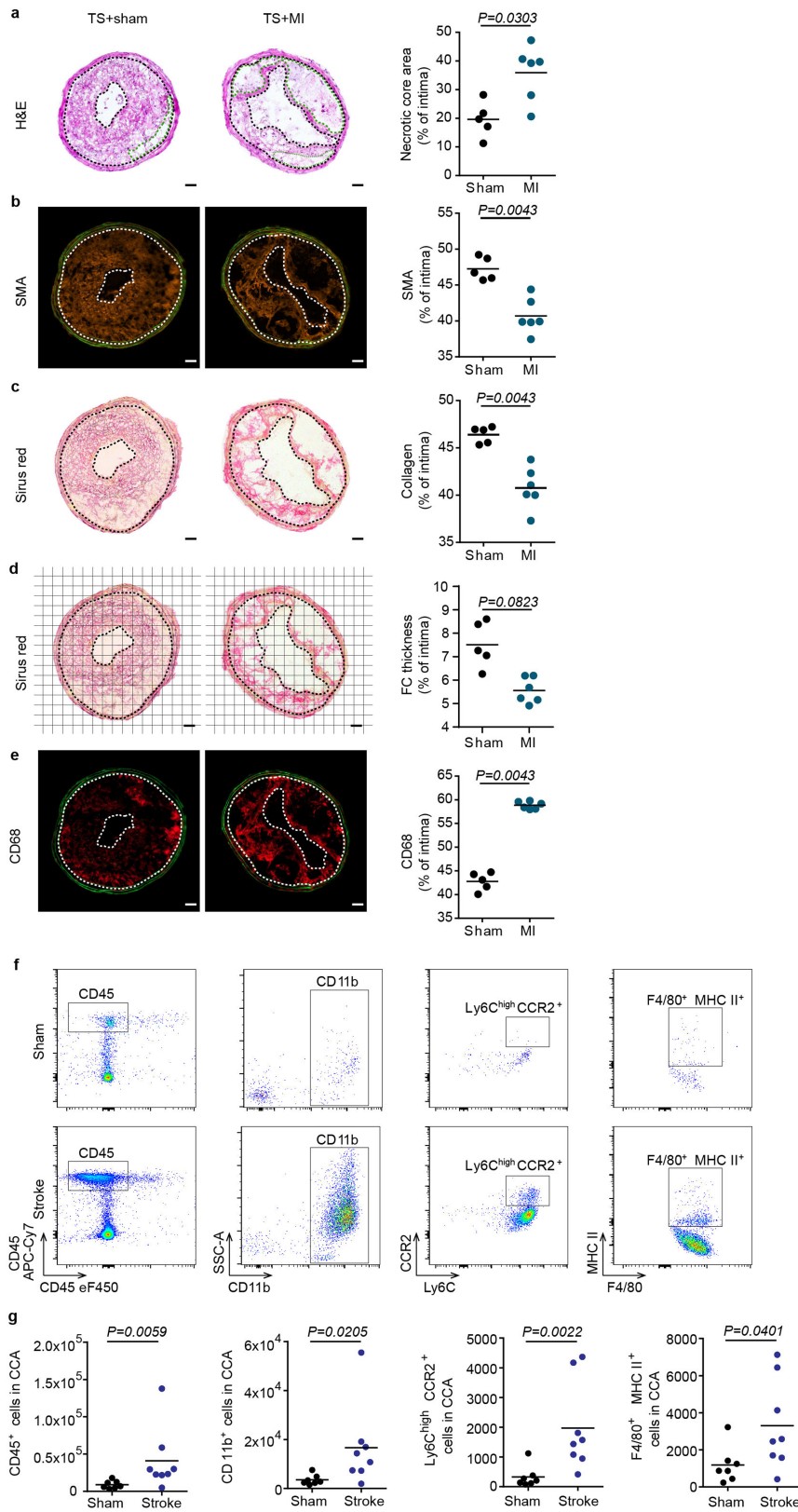

**Extended Data Fig. 4 | MI exacerbates plaque vulnerability and stroke leads to more cellular inflammation.** Representative microphotographs of H&E **a**., SMA **b**., Picro-Sirius red **c**., fibrous cap thickness analysis **d**. and CD68 **e**. staining in CCA sections 1w after sham or myocardial infarction (MI) surgery (scale bar = 50 μm). Area between two dotted lines indicate intima area. Green dotted line represents necrotic core area. Corresponding quantification of necrotic core area, SMA, collagen and CD68 area 1w after sham or MI operated mice (quantification were performed as described in Extended Data Fig. 3, U test, n = 5–6 per group). **f.** Representative gating strategy for flow cytometry analysis of whole CCA cell suspensions 24 h after sham or stroke surgery. **g.** Flow cytometry analysis of CCA cell suspensions showing total leukocytes (CD45+), monocytes (CD11b+), proinflammatory subset (Ly6Chigh CCR2+) and macrophages (F4/80+ MHCII+) cell counts after experimental stroke compared to sham (U test, n = 7–8 per group).

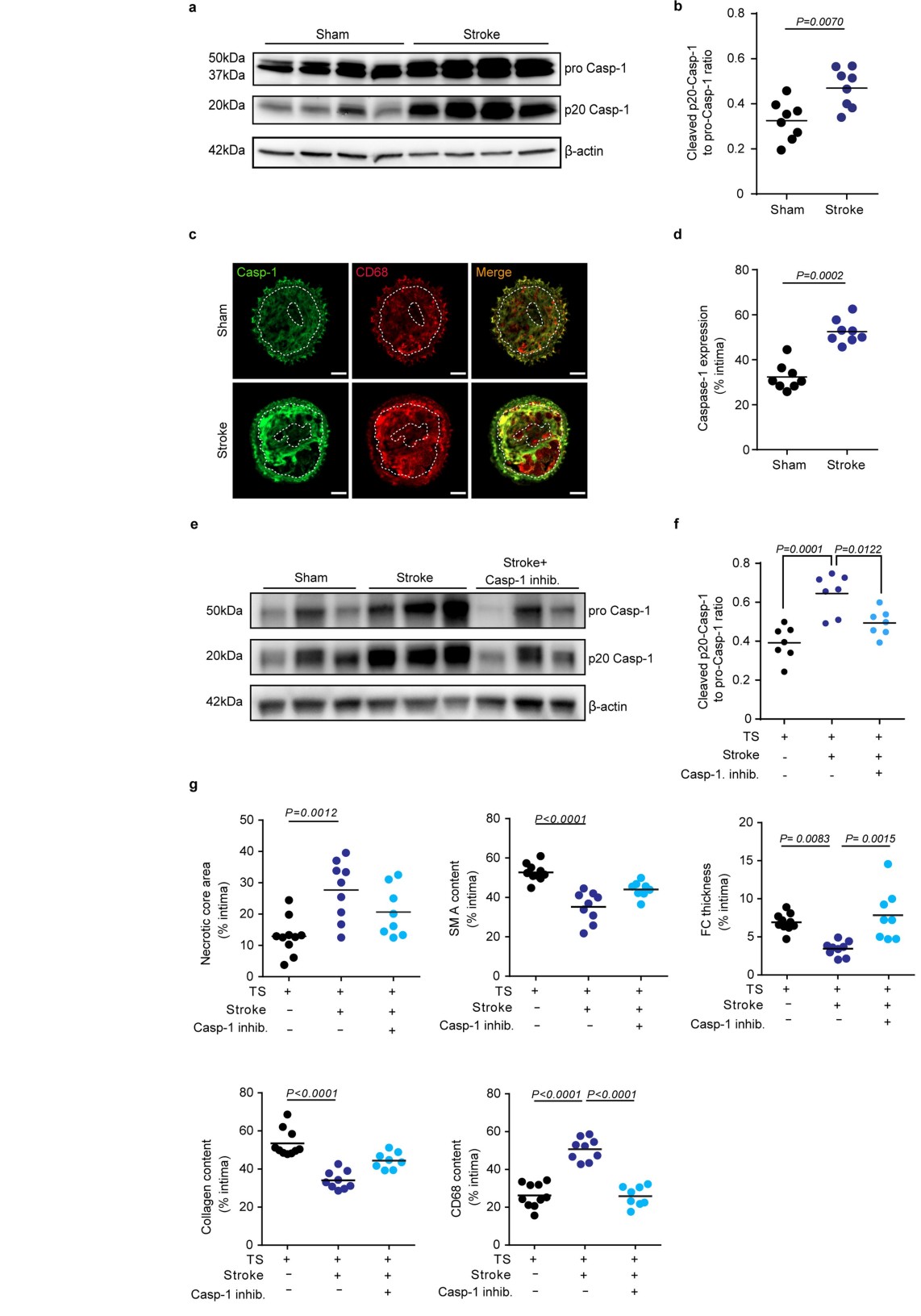

**Extended Data Fig. 5** | See next page for caption.

**Extended Data Fig. 5 | Stroke induces inhibitable inflammasome activation in atherosclerotic plaques. a**. Representative immunoblot of different cleavage forms of caspase-1 in CCA lysates with TS 1w after sham or stroke surgery. **b**. Quantification of cleaved caspase-1 (p20 Casp-1) as ratio to pro caspase-1 (ProCasp-1; U test, n = 8 per group). **c**. Representative immunofluorescence staining of caspase-1 (Casp-1) in CCA sections 1 w after sham or stroke surgery (scale bar = 50 μm). Images were segmented by thresholding to convert fluorescence signal into a binary image. Area between two white dotted lines represent intima. **d**. Caspase-1 expression was quantified as the percentage of total intima area (U test, n = 8 per group). **e**. Representative immunoblot image of the different cleavage forms of caspase-1 (Casp-1) in CCA lysates 1 w after sham, stroke control and stroke + caspase-1 inhibitor (VX 765) administration. **f**. Quantification of cleaved p20 Casp-1 intensity normalized to β-actin in CCA lysates (+TS) 1 w in the three treatment groups (black: sham; blue: stroke; light blue: stroke + VX765, ANOVA, n = 7 per group). **g**. Quantification of necrotic core area, SMA, Fibrous cap thickness, collagen and CD68 area 1w after sham or stroke in the respective treatment groups (performed as shown in Extended Data Fig. 3, ANOVA, n = 8–10 per group). Raw membrane images of all immunoblots can be found in Supplementary Fig. 1.

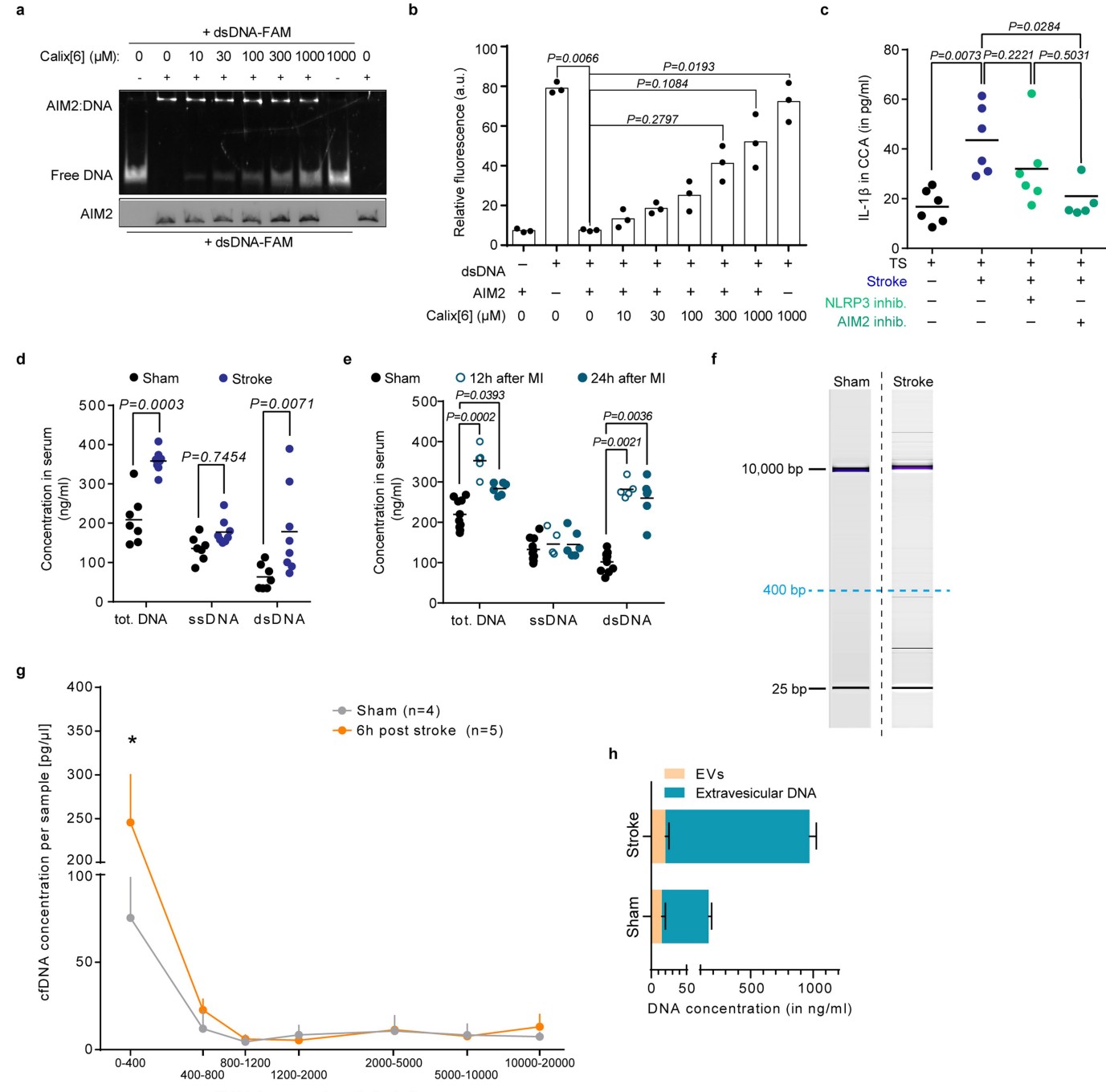

**Extended Data Fig. 6 | Post-stroke plaque inflammasome activation is mediated by cell-free DNA. a**. Representative EMSA gel microphotograph of different Calixarene concentrations (0–1000 µM) interfering with the AIM2-dsDNA complex resulting in increased free DNA. **b**. Quantification of AIM2-free DNA based on its relative fluorescence in the EMSA assay (K test; n = 3 per group; 3 independent experiments). **c**. ELISA analysis of IL-1β in CCA lysates from mice with tandem stenosis (TS), 24 h after stroke in control-, NLRP3 inhibitor- (MCC950) or AIM2 inhibitor- (4-sulfonic calixarene) treated mice, and in sham operated mice (K test; n = 5–6 per group). **d**. Total cell-free DNA (cfDNA), single-stranded DNA (ssDNA) and double-stranded DNA (dsDNA) in mouse serum 24 h after sham or stroke surgery (U test, n = 7–8 per group).

**e**. Total DNA, ssDNA and dsDNA in mice serum after sham or 12 h, 24 h after myocardial infarction (MI) surgery (black: sham; blue circle: 12 h after MI; blue dot: 24 h after MI, multiple t test, n = 5–10 per group). **f**. Representative gel electrophoresis photographs of cfDNA isolated from sham and stroke-operated mouse plasma. **g**. Quantification of electrophoresis-based cfDNA fragment length analysis of sham or stroke-operated mice (K test; n = 4–5 per group; 0–400 bp fragments: Sham vs Stroke: P < 0.0001). **h**. Quantification of extra-vesicular and intra-vesicular DNA after sham or stroke surgery (U test; EV DNA: Sham vs Stroke P = 0.1746; vesicle-free DNA: Sham vs Stroke P = 0.0079; n = 5 per group).

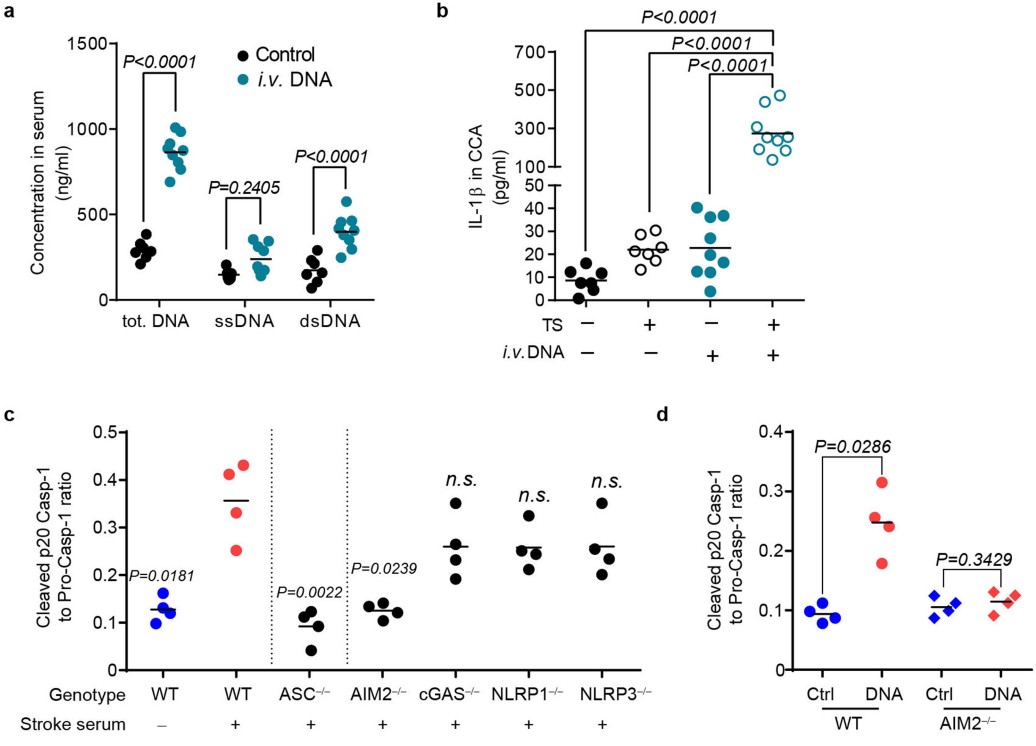

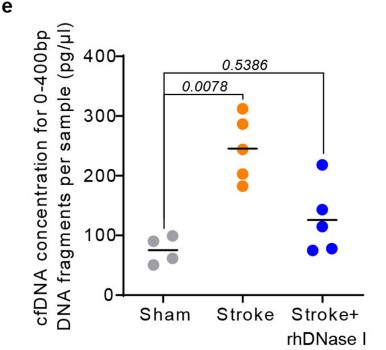

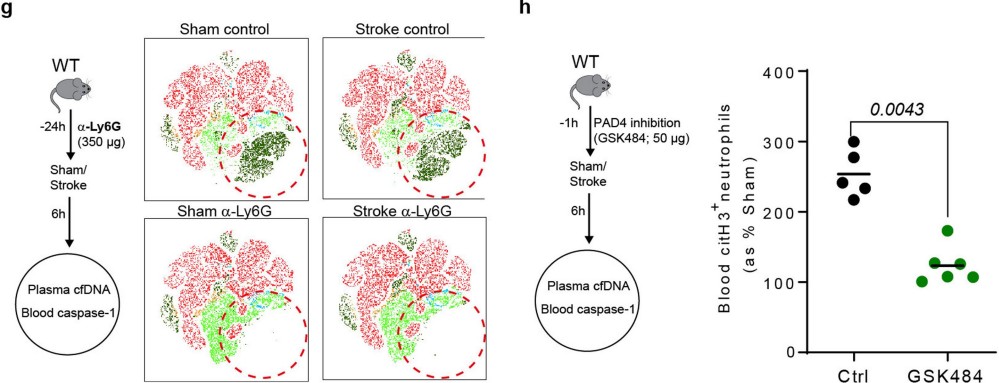

**Extended Data Fig. 7** | See next page for caption.

**Extended Data Fig. 7 | Early neutrophil activation after stroke drives systemic inflammasome activation. a**. total DNA, ssDNA and dsDNA in mouse serum were measured 24 h after control or *i.v.* DNA intravenous injection (multiple t test, control, n = 7; DNA challenge, n = 9). **b**. ELISA analysis for IL-1β in CCA lysates 24 h after *i.v.* DNA challenge (black, control; blue, DNA challenge, ANOVA, n = 7–9 per group). **c**. Caspase-1 cleavage was analyzed via western blotting in BMDMs primed (100 ng/ml LPS for 4 h) and stimulated for 2 h with 25% serum of stroke-operated WT mice. WT BMDMs were compared with BMDMs deficient for ASC, AIM2, cGAS, NLRP1 and NLRP3 (K test; n = 4 per group; 2–3 independent experiments). **d**. Caspase-1 cleavage in WT and AIM2-deficient BMDMs was analyzed by priming (100 ng/ml LPS for 4 h) and stimulating with 250 ng cell-free NET DNA for 2 h (K test; n = 4 per group; 2 independent experiments). **e**. Quantification of short-fragmented (0-400 bp) cfDNA from sham or stroke-operated mice which received rhDNase I treatment (1000U) immediately after stroke surgery (K test; n = 4–5 per group). **f**. Exact percentages (mean and range) of total cfDNA per cell population presented in Fig. 3a. **g**. Representative t-distributed stochastic neighbor embedding (tSNE) plot of antibody-based (α-Ly6G; 1A8) neutrophil depletion efficacy 24 h after antibody administration. **h**. Quantification of citrullinated Histone3[+] (citH3[+]) neutrophils in control or PAD4 inhibitor treated mice. Data is presented as percentage of respective sham group (U test; n = 5–6 per group).

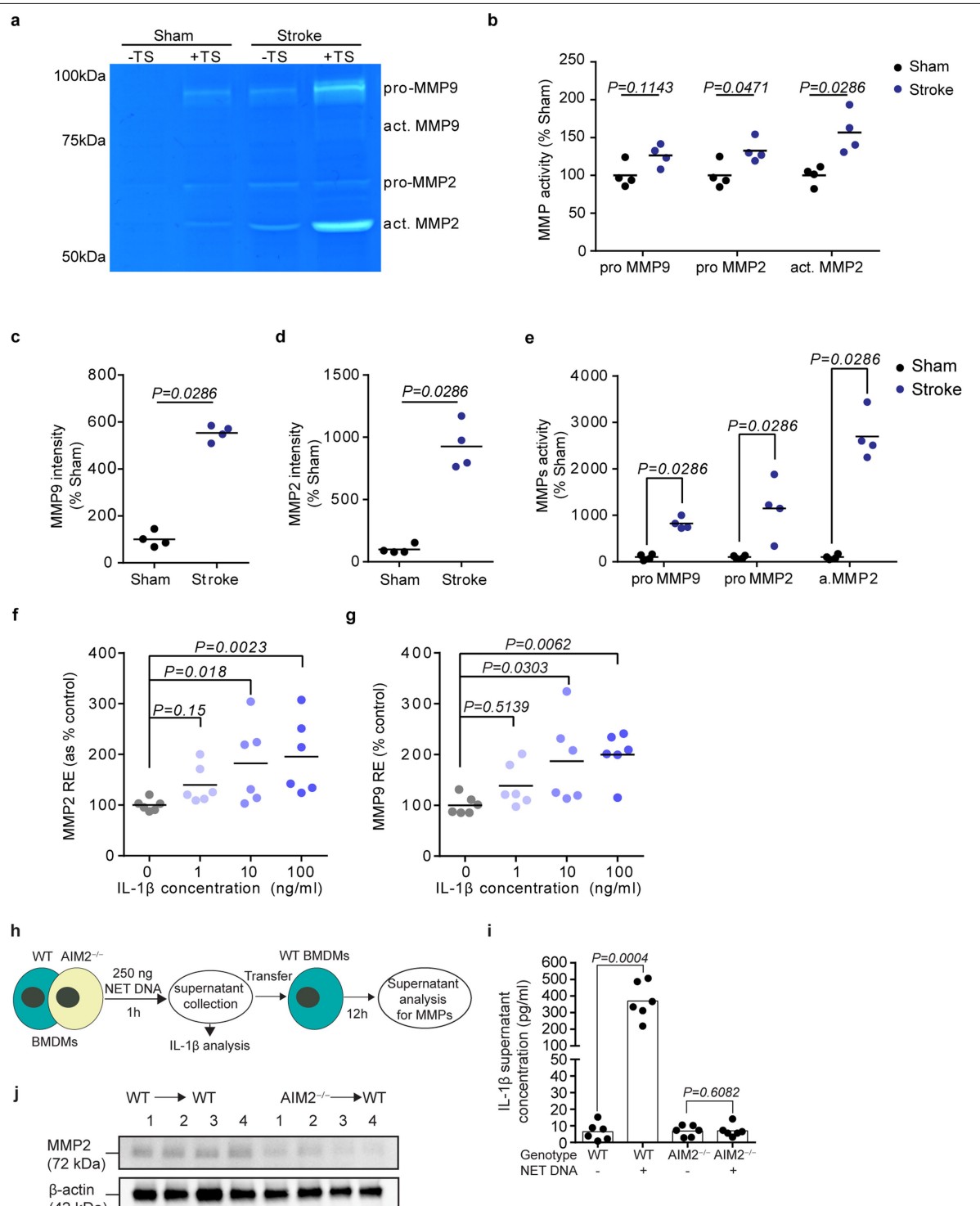

**Extended Data Fig. 8 | Stroke increases matrix metalloproteinase activity in atherosclerotic plaques. a**. Representative images of gelatin zymography of CCA lysates for MMP activity in mice 1w after sham or stroke surgery. The region of MMP activity appears as a clear band against dark blue background where the MMP has digested the gelatin substrate on the zymogram gel. **b**. MMP activity shown in **a**. was quantified as the gelatin digestion area 1w after stroke surgery normalized to sham operated mice (multiple t test, n = 4 per group). **c., d**. Quantification of MMP9 and MMP2 intensity from immunoblot micrograph shown in Fig. 3b (normalized to sham stimulated, U test, n = 4 per group). **e**. MMP activity shown in Fig. 4c was quantified as the gelatin digestion area in the stroke serum-stimulated medium normalized to sham

serum-stimulated group (t test, n = 4 per group). **f**. Relative expression (RE) of MMP2 expression in WT BMDMs after IL-1β stimulation was quantified as the percentage of the control group (H test, n = 6 per group). **g**. Relative expression (RE) for MMP9 mRNA in BMDMs after IL-1ß stimulation (H test, n = 6 per group). **h**. Schematic for NET DNA challenge on WT or AIM2$^{-/-}$ BMDMs and subsequent supernatant transfer to WT BMDMs for MMP expression. **i**. IL-1β supernatant concentration from WT and AIM2$^{-/-}$ BMDMs stimulated with NET DNA (n = 6 per group). **j**. Representative Immunoblot image of Supernatant MMP2 analysis from BMDM supernatants (n = 3 per group). Raw images of zymography with cropping indication can be found in Supplementary Fig. 1h.

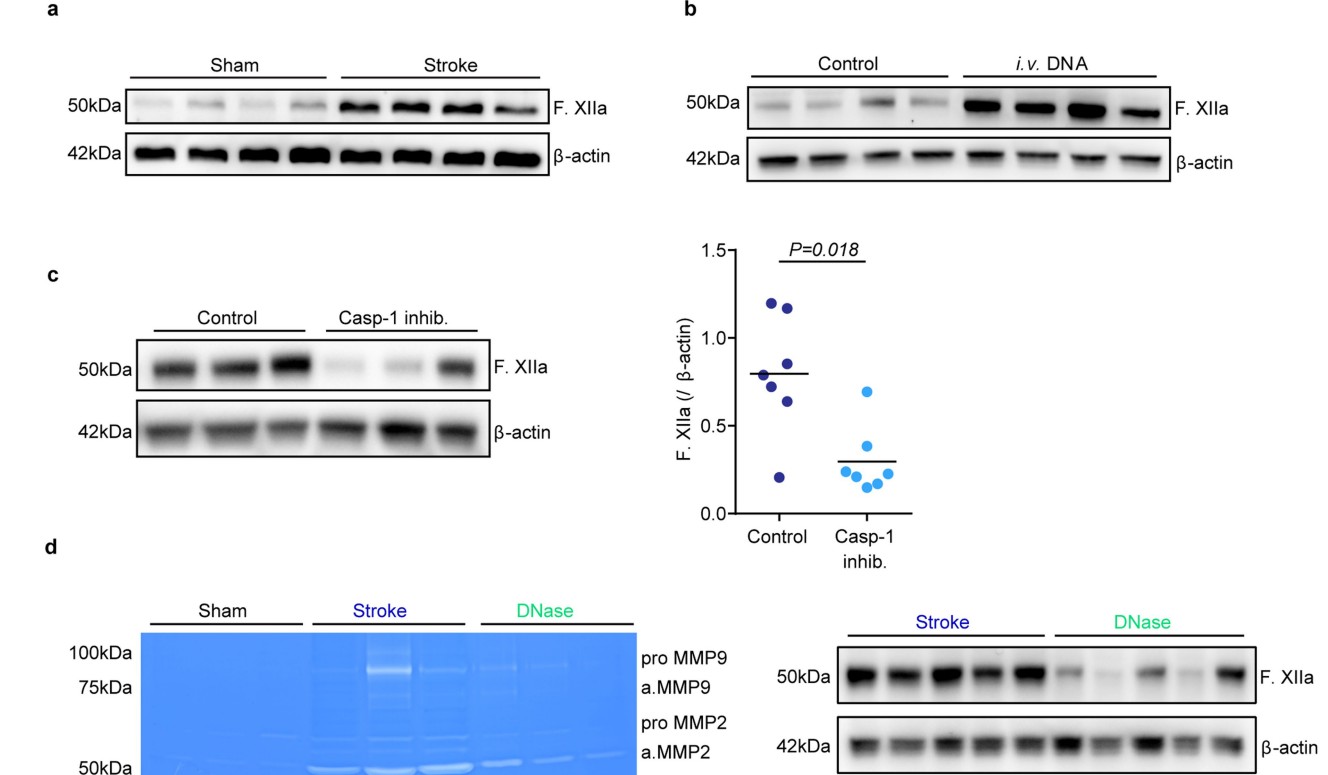

**Extended Data Fig. 9 | Stroke initiates the intrinsic coagulation cascade at atherosclerotic plaques.** All analyses were performed on CCA lysates in mice with stenotic CCA plaques after TS surgery in HFD fed *ApoE⁻/⁻*. **a**. Representative immunoblot of activated Factor XII (F. XIIa) 1 w after stroke or sham surgery. **b**. Representative immunoblot micrograph of F. XIIa in CCA lysates 24 h after *i.v.* DNA challenge. **c**. Representative immunoblot of the F. XIIa in CCA lysates 24 h after stroke in mice treated with control treatment or caspase-1 inhibition (VX765). Corresponding quantification of F. XIIa intensity normalized to β-actin in CCA lysates 1 w after stroke in control- or caspase-1 inhibitor- treated mice (U test, n = 7 per group). **d**. Representative gelatin zymography of CCA lysates for MMP activity in mice 24 h after sham, stroke or stroke + rhDNase I treatment (left). Representative immunoblot of F. XIIa in +TS CCA lysates 24 h after surgery (right). Raw membrane images of immunoblots with cropping indication can be found in Supplementary Fig. 1j–n.

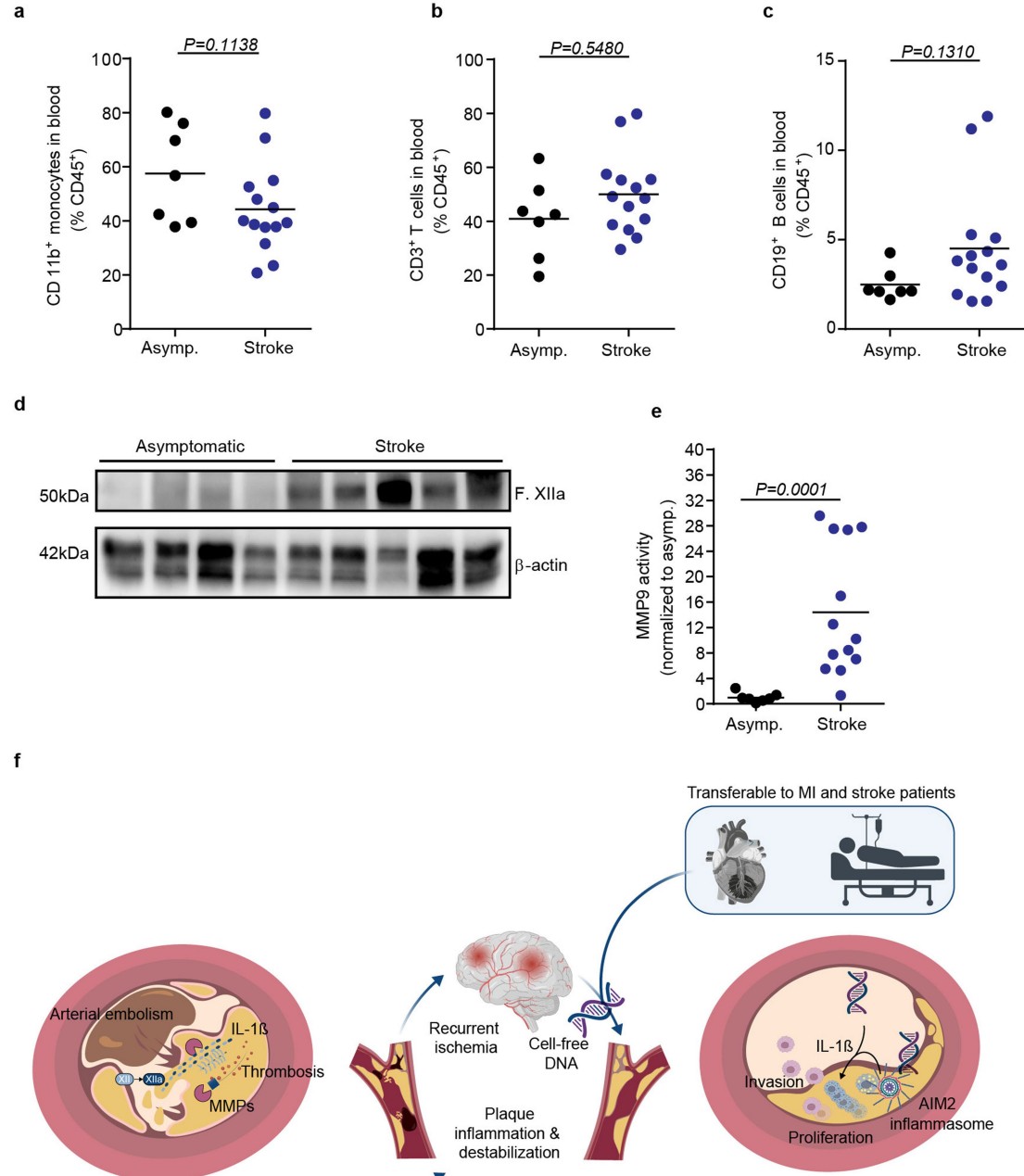

**Extended Data Fig. 10 | Blood leukocyte counts do not differ between stroke and asymptomatic patients with high-grade atherosclerosis.**
**a**–**c**. Flow cytometry analysis of blood from asymptomatic patients or stroke patients showing the percentage of monocytes (CD11b⁺), T cells (CD3⁺) and B cells (CD19⁺) out of total leukocytes (CD45⁺) (U test, asymptomatic patients, n = 7; symptomatic patients, n = 13). **d**. Representative immunoblot from asymptomatic and stroke patients for F. XIIa and β-actin (Quantification can be found in Fig. 4i). **e**. Quantification of MMP9 activity normalized to the activity in asymptomatic patients (Representative image shown in Fig. 4h).

Raw membrane images of immunoblots with cropping indication can be found in Supplementary Fig. 1q. **f**. Overview schematic: Stroke leads to the release of NETosis-derived cell-free DNA activating the AIM2 inflammasome and subsequent secretion of IL-1β. The release of IL-1β drives MMP expression in atherosclerotic plaque, leading to fibrous cap destabilization. The fibrous cap rupture initiates the activation of the intrinsic coagulation cascade resulting in atherothrombosis and subsequent arterio-arterial embolism with secondary brain infarctions.

# Reporting Summary

## Statistics

For all statistical analyses, confirm that the following items are present in the figure legend, table legend, main text, or Methods section.

| n/a | Confirmed | |
|---|---|---|
| ☐ | ☒ | The exact sample size (*n*) for each experimental group/condition, given as a discrete number and unit of measurement |
| ☐ | ☒ | A statement on whether measurements were taken from distinct samples or whether the same sample was measured repeatedly |
| ☐ | ☒ | The statistical test(s) used AND whether they are one- or two-sided *Only common tests should be described solely by name; describe more complex techniques in the Methods section.* |
| ☒ | ☐ | A description of all covariates tested |
| ☐ | ☒ | A description of any assumptions or corrections, such as tests of normality and adjustment for multiple comparisons |
| ☐ | ☒ | A full description of the statistical parameters including central tendency (e.g. means) or other basic estimates (e.g. regression coefficient) AND variation (e.g. standard deviation) or associated estimates of uncertainty (e.g. confidence intervals) |
| ☐ | ☒ | For null hypothesis testing, the test statistic (e.g. $F$, $t$, $r$) with confidence intervals, effect sizes, degrees of freedom and $P$ value noted *Give P values as exact values whenever suitable.* |
| ☒ | ☐ | For Bayesian analysis, information on the choice of priors and Markov chain Monte Carlo settings |
| ☒ | ☐ | For hierarchical and complex designs, identification of the appropriate level for tests and full reporting of outcomes |
| ☒ | ☐ | Estimates of effect sizes (e.g. Cohen's *d*, Pearson's *r*), indicating how they were calculated |

*Our web collection on statistics for biologists contains articles on many of the points above.*

## Software and code

Policy information about availability of computer code

| Data collection | Agilent 2100 Bioanalyzer were used to detect cfDNA fragment lengths, Carl Zeiss LSM880 confocal microscope, Carl Zeiss LSM980 confocal microscope, Carl Zeiss Axio Imager M2 epifluorescent microscope, Carl Zeiss Axiovert 200M, C epifluorescent microscope, Carl Zeiss Axio Observer Z1 microscope were used to collect microscopy data. Northern Light (Cytek Biosciences, USA) flow cytometer was used to collect flow cytometry data. IL-1 beta ELISA (MLB00C, R&D system) kit and iMark Microplate reader (BIO-RAD, Germany) were used for IL-1 beta ELISA. Vevo3100LT (VisualSonics, Fujifilm) was used for ultrasonographic analyses. 3T nanoScan PET/MR (Mediso, Budapest) small animal scanner was used for MRI imaging. Fluoro-Jade C Ready-to-Dilute Staining Kit (TR-100-FJ, Biosensis) was used to detect degenerative neurons. Click-iT Plus TUNEL Assay for In Situ Apoptosis Detection, Alexa Fluor 647 dye kit (C10619, Thermofisher) was used to detect apoptotic cells. Picro-Sirius Red Stain Kit (ab245887, Abcam) was used for collagen analysis. Pierce BCA protein assay kit (23227, Thermofisher), Mini-PROTEAN Tetra Vertical Electrophoresis cell (BIO-RAD, Germany) and Mini Trans-Blot cell (BIO-RAD, Germany) were used for immunoblot analysis. Fusion Fx7 imaging system (Vilber, Germany) was used to acquire immunoblot images. Neutrophil isolation kit (130-097-658, Miltenyi Biotec) was used to isolate neutrophils. Plasma/serum Cell-free circulating DNA purification kit (#55100, NORGEN Biotek, Canada), Nanodrop Spectrophotometer (1000ND, Peqlab, USA), HS dsDNA Assay kit (Q32851, Thermofisher) were used to enrich and measure the cell-free DNA in bloodstream. Novex TM 10% Zymogram Plus (Gelatin) Protein Gels (ZY00100BOX, Thermofisher), colloidal blue staining kit (LC6025, Invitrogen) were used for MMP2, MMP9 gelatin zymography analysis. |
|---|---|
| Data analysis | Fiji, Image J v2 - Open source; VevoLab v3.2.0 software was used for ultrasound imaging analysis; ZEN 2 (blue edition) was used for image analysis; Cytek Northern Lights spectraflo software v3.0 (Cytek Biosciences, USA), FlowJo v. 10.6 (Treestar Inc.) software were used to analyse flow cytometry data. Nucline nanoScan 3.04.014.0000 softeware was used to process MRI images. Statistical analysis were performed using the GraphPad Prism (GraphPad Inc.). Rstudio Vers 1.1.477 was used for principal component analysis with ggplot2 package (Vers. 3.4.3). |

For manuscripts utilizing custom algorithms or software that are central to the research but not yet described in published literature, software must be made available to editors and reviewers. We strongly encourage code deposition in a community repository (e.g. GitHub). See the Nature Portfolio guidelines for submitting code & software for further information.

# Data

Policy information about availability of data

All manuscripts must include a data availability statement. This statement should provide the following information, where applicable:
- Accession codes, unique identifiers, or web links for publicly available datasets
- A description of any restrictions on data availability
- For clinical datasets or third party data, please ensure that the statement adheres to our policy

Any requests for raw data or reagents should be directed to and will be fulfilled by the corresponding authors, Arthur Liesz (Arthur.Liesz@med.uni-muenchen.de) and Stefan Roth (Stefan.Roth@med.uni-muenchen.de).

# Research involving human participants, their data, or biological material

Policy information about studies with human participants or human data. See also policy information about sex, gender (identity/presentation), and sexual orientation and race, ethnicity and racism.

| | |
|---|---|
| Reporting on sex and gender | Sex distribution was equally assigned in this study. Detailed information please see 'Patient corhots for epidemiological analysis', 'Patient cohort for carotid endarterectomy sample analysis' and 'Patient cohort for myocardial infarction sample analysis' in the Methods section. |
| Reporting on race, ethnicity, or other socially relevant groupings | Information on race/ethnicity was not recorded from study participants and was not used as analysis readout or proxy. |
| Population characteristics | Carotid endarterectomy samples of symptomatic or asymptomatic patients were collected at the Department and Cardiothoracic-Transplantation- and Vascular Surgery at Hannover Medical School between June 2018 and December 2020. . Carotid stenosis was defined as symptomatic if cerebral ischemia occurred in the territory of the affected artery and concurrent stroke etiologies were excluded following standardized stroke diagnostics including cranial computed tomography (CT) and/or magnetic resonance (MR) imaging. CT or MR-angiography, transthoracic or transesophageal echocardiography, cardiac rhythm monitoring and Doppler/duplex ultrasound. Peripheral venous blood was drawn immediately prior to surgery and EDTA whole blood samples were used for flow cytometry analysis. Carotid plaque samples were obtained during carotid endarterectomy and immediately preserved in phosphate-buffered saline. Both blood and tissue samples were sent for further analysis on the same day of collection. All patients provided written informed consent and the ethics committee at Hannover Medical School approved the study.Thirteen patients with symptomatic and seven patients with asymptomatic, high-grade carotid stenosis were recruited. Median age was 73 years (25th-75th percentile: 62-80 years). STEMI patients were prospectively recruited between September 2016 and February 2018 at the German Heart Centre Munich and the Klinikum rechts der Isar (both at the Technical University of Munich). The diagnosis of STEMI was based on chest pain within the last 12 hours, persistent ST-segment elevation ≥ 1mm in at least 2 extremities or ≥ 2mm in at least 2 chest leads and diagnosis of type 1 myocardial infarction according to cardiac catheterization. Exclusion criteria were: cardiogenic shock, LV-EF ≤ 35, co-existing chronic or inflammatory diseases, anti-inflammatory drug therapy (e.g. cortisol), myocardial infarction type 2 – 5. Blood samples for plasma analysis were collected in EDTA tubes immediately after admission to the hospital or latest 6 hours after coronary intervention. Age- and sex-matched patients with known stable coronary artery disease served as controls. They were prospectively recruited between February 2017 and February 2018 during consultation in the outpatient department of the German Heart Centre Munich for routine examination. Exclusion criteria were: history of myocardial infarction, reduced LV-EF, chronic or inflammatory diseases, anti-inflammatory drug therapy. Blood samples for plasma analysis were collected in EDTA tubes on the day of consultation in the outpatient department. All patients provided written informed consent and the institutional ethics committee at Technical University Munich approved the study (235/16 S). EDTA tubes of both STEMI and control patients were centrifuged at 4°C and 2000xg for 15 minutes directly after collection. Plasma aliquots were stored at -80°C for further analysis. |
| Recruitment | Hannover: Patients were prospectively recruited at the Department of Neurology and Cardiothoracic-Transplantation- and Vascular Surgery at the Hannover Medical School between June 2018 and December 2020. All patients provided written informed consent to participate in the study. Stroke blood samples/DEMDAS/PROCIS: Patients over 18y with an acute stroke (< 72h) were recruited through local certified stroke units. All patients provided written informed consent to participate in the study. Exclusion criteria were stroke onset >72h and hemorrhagic stroke. STEMI samples: STEMI patients were prospectively recruited between September 2016 and February 2018 at the German Heart Centre Munich and the Klinikum rechts der Isar (both at the Technical University of Munich). The diagnosis of STEMI was based on chest pain within the last 12 hours, persistent ST-segment elevation ≥ 1mm in at least 2 extremities or ≥ 2mm in at least 2 chest leads and diagnosis of type 1 myocardial infarction according to cardiac catheterization. Exclusion criteria were: cardiogenic shock, LV-EF ≤ 35, co-existing chronic or inflammatory diseases, anti-inflammatory drug therapy (e.g. cortisol), myocardial infarction type 2 – 5. Blood samples for plasma analysis were collected in EDTA tubes immediately after admission to the hospital or latest 6 hours after coronary intervention. |
| Ethics oversight | Stroke blood samples and PROCIS/DEDEMAS: The local ethics committee of the University Hospital of Munich (Number 17-005, 20-0935, 20-0935 and 121-09) approved the study and written informed consent for permission was given by all patients. STEMI samples: All patients provided written informed consent and the institutional ethics committee at Technical University |

Munich approved the study (235/16 S).
Endarterectomy samples: This study was approved by the ethics committee at Hannover Medical School (Ethics vote No. 7484-2017) and conducted in accordance with the ethical principles outlined in the Declaration of Helsinki.

Note that full information on the approval of the study protocol must also be provided in the manuscript.

# Field-specific reporting

Please select the one below that is the best fit for your research. If you are not sure, read the appropriate sections before making your selection.

☒ Life sciences ☐ Behavioural & social sciences ☐ Ecological, evolutionary & environmental sciences

For a reference copy of the document with all sections, see nature.com/documents/nr-reporting-summary-flat.pdf

# Life sciences study design

All studies must disclose on these points even when the disclosure is negative.

| | |
|---|---|
| Sample size | 1. In vivo: For this exploratory study, animal numbers were estimated based on previous results from the transient ischemia-reperfusion stroke model on extent and variability of atheroprogression after stroke.<br>2. In vitro: For the in vitro experiments, sample size was estimated based on previous in vitro studies. 3 biological independent experiments were considered with at least duplicates for technical replicates.<br>3. Hannover: Exploratory study with completely novel approach, no pre-emptive, dedicated sample size calculation was possible given the lack of prior data.<br>4. DEMDAS/PROCIS: Both studies are observational hospital-based cohort studies in patients suffering from acute stroke with no interventional group. No a priori calculation was made.<br>5. Myocard Infarct: This samples were used in exploratory study to unravel differences in cfDNA after myocardial infarction. No historical data was available to perform an a priori calculation.<br>6. DNA methylation: This experiment was as well planned as an exploratory study since no methylation analysis for post-stroke cfDNA was made before. |
| Data exclusions | For ischemia-reperfusion stroke model, data were excluded: 1. Insufficient MCA occlusion (a reduction in blood flow > 20% of the baseline value). 2. Death during the surgery. 3. Lack of brain ischemia as quantified post-mortem by histological analysis. For carotid tandem stenosis model, mice were excluded: 1. Death during the surgery. 2. No sign of atherosclerotic plaque formation as analyzed by histological analysis. |
| Replication | All attempts for replication were successful. At least 4 biological replicates were used to confirm reprodicibility. |
| Randomization | 1. In vivo: Animals were randomly assigned to different groups.<br>2. In vitro: Grouping for in vitro (murine neurophils and macrophages) was not randomized.<br>3. Hannover: The clinical part of the work is observational in nature, and therefore no randomisation was carried out. Groups were defined according to whether patients were symptomatic or asymptomatic. Both groups are patient collectives with severe atherosclerotic disease and a correspondingly high level of vascular risk factors. Since we were explicitly interested in the possible effect of stroke, which is fulfilled as per definition by this group allocation, an adjustment for confounders is not only not necessary, but would even introduce additional bias.<br>4. DEMDAS/PROCIS: Both studies are observational hospital-based cohort studies in patients suffering from acute stroke with no interventional group. No randomization was used.<br>5. Myocard Infarct: Prospectively collected samples from STEMI or control patients were not randomized.<br>6. DNA methylation: Prospectively collected samples from acute ischemic stroke or control patients were not randomized. |
| Blinding | All analyses were performed by investigators blinded to group allocation during data collection. |

# Reporting for specific materials, systems and methods

We require information from authors about some types of materials, experimental systems and methods used in many studies. Here, indicate whether each material, system or method listed is relevant to your study. If you are not sure if a list item applies to your research, read the appropriate section before selecting a response.

## Materials & experimental systems

| n/a | Involved in the study |
|---|---|
| ☐ | ☒ Antibodies |
| ☒ | ☐ Eukaryotic cell lines |
| ☒ | ☐ Palaeontology and archaeology |
| ☐ | ☒ Animals and other organisms |
| ☒ | ☐ Clinical data |
| ☒ | ☐ Dual use research of concern |
| ☒ | ☐ Plants |

## Methods

| n/a | Involved in the study |
|---|---|
| ☒ | ☐ ChIP-seq |
| ☐ | ☒ Flow cytometry |
| ☐ | ☒ MRI-based neuroimaging |

# Antibodies

| | |
|---|---|
| Antibodies used | **Immunostaining:**<br>anti-CD68 (rat, ab53444, abcam, 1:200), anti-alpha smooth muscle actin (mouse, ab7817, abcam, 1:200), anti-Iba-1 (rabbit, 019-19741, Wako, 1:200), anti-Ki 67 (rabbit, 9129S, Cell Signaling, 1:200), anti-mouse caspase-1 (p20; mouse, AG-20B-0042-C100, Adipogen, 1:1000), recombinant anti-MMP2 (rabbit, ab92536, abcam, 1:1000), anti-MMP9 (rabbit, ab38898, abcam, 1:1000), anti-mouse actin (rabbit, A2066-.2ml, Sigma, 1:1000), anti-human caspase-1 (p20; mouse, AG-20B-0042B-C100, Adipogen, 1:1000), anti-Factor XII (rabbit, PA5-116703, Invitrogen, 1:100), anti-CD31 (rat, BM4086, OriGene, 1:200), anti-Von Willebrand Factor (sheep, ab11713, abcam, 1:50), anti-CD41 (rat, ab33661, abcam, 1:200), anti-Fibrinogen (rabbit, ab34269, abcam, 1:100), goat anti-rabbit IgG (H+L) Crossed-Absorbed Secondary Antibody, Alexa Fluor 594 (A-11012, Invitrogen, 1:200), goat anti-Mouse IgG (H+L) Highly Crossed-Absorbed Secondary Antibody, Alexa Fluor 488 (A-32723, Invitrogen, 1:200), goat anti-rat IgG (H+L) Crossed-Absorbed Secondary Antibody, Alexa Fluor 647 (A-21247, Invitrogen, 1:200), donkey anti-sheep IgG (H+L) Crossed-Absorbed Secondary Antibody, Alexa Fluor 594 (A-11016, Invitrogen, 1:200), anti-mouse IgG (goat, HRP-conjugated, P0447, Dako, 1:5000), anti-rabbit IgG (goat, HRP-conjugated, PI-1000, Vector, 1:5000). anti-mouse ASC (rabbit, AL177, 1:100), anti-mouse beta tubulin (mouse, T4027; 1:100). anti-mouse collagen I (rabbit, EPR24331-53, 1:250).<br>**Flow cytometry:**<br>anti-mouse CD45-APC-Cy7 (clone: 30-F11, 103116, Biolegend, 3 μg per mouse), anti-mouse CD45-eFluor450 (clone: 30-F11, 48-0451-82, Invitrogen, 3μg per mouse; 1:200), anti-mouse CD11b-PerCP-Cy5.5 (clone: M/70, 45-0112-82, Invitrogen, 1:200), anti-mouse Ly6G-PE-Fluor610 (clone: 1A8-ly6g, 61-9668-82, Invitrogen, 1:200), anti-mouse Ly6C-BV570 (clone: HK1.4, 128030, Biolegend, 1:200), anti-mouse CD192-APC (clone: SA203G11, 150628, Biolegend, 1:200), anti-mouse MHC II-PE (clone: NIMR-4, 12-5322-81, Invitrogen, 1:200), anti-mouse F4/80-PE-Cyanine7 (clone: BM8, 25-4801-82, Invitrogen, 1:200), anti-human CD3-FITC (clone: HIT3a, 11-0039-42, Invitrogen, 1:200), anti-human CD8a-PE (clone: SK1, 12-0087-42, Invitrogen, 1:200), anti-human CD19-APC (clone: HIB19, 17-0199-42, Invitrogen, 1:200), anti-human CD45-eFluor 450 (clone: 2D1, 48-9459-42, Invitrogen, 1:200), anti-human CD14-PerCP-Cy5.5 (clone: 61D3, 45-0149-42, Invitrogen, 1:200), anti-human CD16-FITC (clone: CB16, 11-0168-42, Invitrogen, 1:200), anti-human CD11b-PE (clone: ICRF44, 12-0118-42, Invitrogen, 1:200)<br>**Antibody-based depletion:**<br>anti-mouse Ly6G InVivoMab (clone: 1A8, BE0075-1, BioXCell, 14 mg kg-1 body weight); anti-mouse IgG non-reactive isotype control (BE0083, BioXCell, 14 mg kg-1 body weight), anti-mouse IL-1beta InvivoMab (clone: B122, BE0246, BioXCell, 4 mg kg-1 body weight), Polyclonal armenian hamster IgG InVivoMab (BE0091, BioXCell, 4 mg kg-1 body weight) |
| Validation | **Immunostaining:**<br>CD68 (rat, ab53444, abcam): mouse, immunofluorescence, https://www.abcam.com/cd68-antibody-fa-11-ab53444.html;<br>alpha smooth muscle actin (mouse, ab7817, abcam): mouse/human, immunofluorescence, https://www.abcam.com/alpha-smooth-muscle-actin-antibody-1a4-ab7817.html;<br>Iba-1 (rabbit, 019-19741, Wako): mouse/human, immunofluorescence, https://www.fujifilmcdi.com/anti-iba1-polyclonal-antibody-019-19741?gclid=CjwKCAiA_vKeBhAdEiwAFb_nrUda60XIg-w6G9L4Fm-qgsxqOmIH59dikDjcZDr7TgocUXwRKrWSoBoCSA8QAvD_BwE;<br>Ki 67 (rabbit, 9129S, Cell Signaling): mouse/human, immunofluorescence, https://www.cellsignal.com/products/primary-antibodies/ki-67-d3b5-rabbit-mab/9129?site-search-type=Products&N=4294956287&Ntt=9129s&fromPage=plp&_requestid=144354;<br>CD31 (rat, BM4086, OriGene): mouse, immunofluorescence, https://www.origene.com/catalog/antibodies/primary-antibodies/bm4086/pecam1-rat-monoclonal-antibody-clone-id-er-mp12;<br>Von Willebrand Factor (sheep, ab11713, abcam): mouse/human, immunofluorescence, https://www.abcam.com/von-willebrand-factor-antibody-ab11713.html;<br>CD41 (rat, ab33661, abcam): mouse, immunofluorescence, https://www.abcam.com/cd41-antibody-mwreg30-ab33661.html;<br>Fibrinogen (rabbit, ab34269, abcam): mouse/human, immunofluorescence, https://www.abcam.com/fibrinogen-antibody-ab34269.html;<br>Factor XII (rabbit, PA5-116703, Invitrogen): mouse/human, immunoblot, immunofluorescence, https://www.thermofisher.com/antibody/product/Factor-XII-Antibody-Polyclonal/PA5-116703;<br>Mouse caspase-1 (p20; mouse, AG-20B-0042-C100, Adipogen): mouse, immunoblot, immunofluorescence, https://adipogen.com/ag-20b-0042-anti-caspase-1-p20-mouse-mab-casper-1.html/;<br>MMP2 (rabbit, ab92536, abcam): mouse/human, immunoblot, https://www.abcam.com/mmp2-antibody-epr1184-ab92536.html;<br>MMP9 (rabbit, ab38898, abcam): mouse, immunoblot, https://www.abcam.com/mmp9-antibody-ab38898.html;<br>Actin (rabbit, A2066-.2ml, Sigma): mouse/human, immunoblot, https://www.sigmaaldrich.com/DE/en/product/sigma/a2066;<br>Human caspase-1 (p20; mouse, AG-20B-0042B-C100, Adipogen): human, immunoblot, https://adipogen.com/ag-20b-0042b-anti-caspase-1-p20-mouse-mab-casper-1-biotin.html;<br>Goat anti-rabbit IgG (H+L) Crossed-Absorbed Secondary Antibody, Alexa Fluor 594 (A-11012, Invitrogen): mouse/human, immunofluorescence, https://www.thermofisher.com/antibody/product/Goat-anti-Rabbit-IgG-H-L-Cross-Adsorbed-Secondary-Antibody-Polyclonal/A-11012;<br>Goat anti-Mouse IgG (H+L) Highly Crossed-Absorbed Secondary Antibody, Alexa Fluor 488 (A-32723, Invitrogen): mouse/human, immunofluorescence, https://www.thermofisher.com/antibody/product/Goat-anti-Mouse-IgG-H-L-Highly-Cross-Adsorbed-Secondary-Antibody-Polyclonal/A32723;<br>Goat anti-rat IgG (H+L) Crossed-Absorbed Secondary Antibody, Alexa Fluor 647 (A-21247, Invitrogen): mouse/human, immunofluorescence, https://www.thermofisher.com/antibody/product/Goat-anti-Rat-IgG-H-L-Cross-Adsorbed-Secondary-Antibody-Polyclonal/A-21247;<br>Mouse IgG (goat, HRP-conjugated, P0447, Dako): mouse, immunoblot, https://www.agilent.com/en/product/specific-proteins/elisa-kits-accessories/goat-anti-mouse-immunoglobulins-hrp-affinity-isolated-2717109;<br>Rabbit IgG (goat, HRP-conjugated, PI-1000, Vector): rabbit, immunoblot, https://vectorlabs.com/products/antibodies/peroxidase-goat-anti-rabbit-igg.<br>**Flow cytometry:**<br>CD45 (APC-Cy7, clone: 30-F11, 103116, Biolegend): mouse, https://www.biolegend.com/en-us/products/apc-cyanine7-anti-mouse-cd45-antibody-2530?GroupID=BLG1932;<br>CD45 (eFluor450, clone: 30-F11, 48-0451-82, Invitrogen): mouse, https://www.thermofisher.com/antibody/product/CD45-Antibody-clone-30-F11-Monoclonal/48-0451-82; |

CD11b (PerCP-Cy5.5, clone: M/70, 45-0112-82, Invitrogen): mouse, https://www.thermofisher.com/antibody/product/CD11b-Antibody-clone-M1-70-Monoclonal/45-0112-82;
Ly6G (PE-Fluor610, clone: 1A8-ly6g, 61-9668-82, Invitrogen): mouse, https://www.thermofisher.com/antibody/product/Ly-6G-Antibody-clone-1A8-Ly6g-Monoclonal/61-9668-82;
Ly6C (BV570, clone: HK1.4, 128030, Biolegend): mouse, https://www.biolegend.com/en-us/products/brilliant-violet-570-anti-mouse-ly-6c-antibody-7392;
CD192 (APC,clone: SA203G11, 150628, Biolegend): mouse, https://www.biolegend.com/en-us/products/apc-anti-mouse-cd192-ccr2-antibody-17676;
MHC II (PE, clone: NIMR-4, 12-5322-81, Invitrogen): mouse, https://www.thermofisher.com/antibody/product/MHC-Class-II-I-A-Antibody-clone-NIMR-4-Monoclonal/12-5322-81;
F4/80 (PE-Cyanine7, clone: BM8, 25-4801-82, Invitrogen): mouse, https://www.thermofisher.com/antibody/product/F4-80-Antibody-clone-BM8-Monoclonal/25-4801-82;
CD3 (FITC, clone: HIT3a, 11-0039-42, Invitrogen): human, https://www.thermofisher.com/antibody/product/CD3-Antibody-clone-HIT3a-Monoclonal/11-0039-42;
CD8a (PE, clone: SK1, 12-0087-42, Invitrogen): human, https://www.thermofisher.com/antibody/product/CD8a-Antibody-clone-SK1-Monoclonal/12-0087-42;
CD19 (APC, clone: HIB19, 17-0199-42, Invitrogen): human, https://www.thermofisher.com/antibody/product/CD19-Antibody-clone-HIB19-Monoclonal/17-0199-42;
CD45 (eFluor 450, clone: 2D1, 48-9459-42, Invitrogen): human, https://www.thermofisher.com/antibody/product/CD45-Antibody-clone-2D1-Monoclonal/48-9459-42;
CD14 (PerCP-Cy5.5, clone: 61D3, 45-0149-42, Invitrogen): human, https://www.thermofisher.com/antibody/product/CD14-Antibody-clone-61D3-Monoclonal/45-0149-42;
CD16 (FITC, clone: CB16, 11-0168-42, Invitrogen): human, https://www.thermofisher.com/antibody/product/CD16-Antibody-clone-eBioCB16-CB16-Monoclonal/11-0168-42;
CD11b (PE, clone: ICRF44, 12-0118-42, Invitrogen): mouse/human, https://www.thermofisher.com/antibody/product/CD11b-Antibody-clone-ICRF44-Monoclonal/12-0118-42.
ASC (clone: AL177, Rabbit, AG-25B-0006-C100, Adipogen): https://adipogen.com/ag-25b-0006-anti-asc-pab-al177.html/
Beta tubulin (clone: T4027; mouse, Sigma): https://www.sigmaaldrich.com/deepweb/assets/sigmaaldrich/product/documents/295/024/t4026dat.pdf
Collagen I (clone: EPR24331-53, rabbit, Abcam):  https://www.abcam.com/products/primary-antibodies/collagen-i-antibody-epr24331-53-bsa-and-azide-free-ab279711.html
anti-mouse Ly6G InVivoMab (clone: 1A8, BioXCell); https://bioxcell.com/invivomab-anti-mouse-ly6g-be0075-1
anti-mouse IgG non-reactive isotype control (BioXCell), https://bioxcell.com/invivomab-mouse-igg1-isotype-control-unknown-specificity-be0083
anti-mouse IL-1beta InvivoMab (clone: B122, BioXCell), https://bioxcell.com/invivomab-anti-mouse-rat-il-1b
Polyclonal armenian hamster IgG InVivoMab (BioXCell), https://bioxcell.com/invivomab-polyclonal-armenian-hamster-igg-be0091

# Animals and other research organisms

Policy information about studies involving animals; ARRIVE guidelines recommended for reporting animal research, and Sex and Gender in Research

| | |
|---|---|
| Laboratory animals | All mice used in this study were between 6 to 20 weeks of age. C57BL/6J WT mice were purchased from Charles River and housed at the animal core facility of the Centre for Stroke and Dementia Research (Munich, Germany). Apoe-/- mice on C57BL/6J background were bred and housed at the animal core facility of the Centre for Stroke and Dementia Research (Munich, Germany). ApoE-/- (B6.129P2-Apoetm1Unc/J; JAX strain: 002052), wildtype (C57BL6/J; JAX strain: 000664), AIM2-/- (B6.129P2-Aim2Gt(CSG445)Byg/J; JAX strain: 013144), Pycard-/- (ASC-/-B6.129S5-Pycardtm1Vmd) and R26-CAG-ASC-citrine mice (B6.Cg-Gt(ROSA)26ortm1.1(CAG-Pycard/mCitrine*,-CD2*); JAX strain: 030744)  were bred and housed at the animal core facility of the Centre for Stroke and Dementia Research (Munich, Germany). LDLr-/-Mx1Cre:c-Mybfl/fl mice were bred and housed at the animal facility of Walter Brendel Centre (Munich, Germany). ApoE-/-mice were fed an HFD (#88137, ssniff) from 8 weeks on. cGAS-/- (B6(C)-Cgastm1d(EUCOMM)Hmgu/J), NLRP1-/- (Del(11Nlrp1a-Nlrp1c-ps)1Smas) and NLRP3-/-C57BL6/J-NLRP3tm1Tsc) mice were bred and housed at the Gene Centre of the LMU University Munich (Germany).<br>All mice (besides ApoE-/-) were maintained on a standard rodent chow diet until used for experiments. Apoe-/- mice were maintained on a rodent chow diet for 8 weeks, and were fed a HFD (#88137, ssniff) containing 42% kcal fat from lard, 15% kcal protein, and 43% kcal carbohydrates from 8 weeks on. All mice were housed under specific pathogen free conditions in 12/12 h light/dark cycles, at 21 °C and 50% humidity with food and water. |
| Wild animals | This study did not involve wild animals. |
| Reporting on sex | Apoe-/- mice on C57BL/6J background were bred and housed at the animal core facility of the Centre for Stroke and Dementia Research (Munich, Germany). Both male and female mice were randomly assigned to different groups in this study. |
| Field-collected samples | This study did not involve samples collected from the field. |
| Ethics oversight | All animal experiments were performed in accordance with the guidelines for the use of experimental animals and were approved by the government committee of Upper Bavaria (Regierungspraesidium Oberbayern). Animal experiments were performed according to the guidelines of the Animal Research: Reporting of In Vivo Experiment (ARRIVE). |

Note that full information on the approval of the study protocol must also be provided in the manuscript.

# Plants

| | |
|---|---|
| Seed stocks | *Report on the source of all seed stocks or other plant material used. If applicable, state the seed stock centre and catalogue number. If plant specimens were collected from the field, describe the collection location, date and sampling procedures.* |
| Novel plant genotypes | *Describe the methods by which all novel plant genotypes were produced. This includes those generated by transgenic approaches, gene editing, chemical/radiation-based mutagenesis and hybridization. For transgenic lines, describe the transformation method, the number of independent lines analyzed and the generation upon which experiments were performed. For gene-edited lines, describe the editor used, the endogenous sequence targeted for editing, the targeting guide RNA sequence (if applicable) and how the editor was applied.* |
| Authentication | *Describe any authentication procedures for each seed stock used or novel genotype generated. Describe any experiments used to assess the effect of a mutation and, where applicable, how potential secondary effects (e.g. second site T-DNA insertions, mosiacism, off-target gene editing) were examined.* |

# Flow Cytometry

## Plots

Confirm that:

☒ The axis labels state the marker and fluorochrome used (e.g. CD4-FITC).

☒ The axis scales are clearly visible. Include numbers along axes only for bottom left plot of group (a 'group' is an analysis of identical markers).

☒ All plots are contour plots with outliers or pseudocolor plots.

☒ A numerical value for number of cells or percentage (with statistics) is provided.

## Methodology

| | |
|---|---|
| Sample preparation | Isolated CCA samples were mixed with digestion buffer, consisting of collagenase type XI (125 U/ml, C7657), hyaluronidase type 1-s (60 U/ml, H3506), DNase I (60 U/ml, D5319), collagenase type I (450 U/ml, C0130; all enzymes from Sigma Aldrich, Germany) in 1x PBS, and were digested at 750 rpm for 30 min at 37 °C. After digestion, CCA materials were homogenized through a 40 μm cell strainer, washed at 500 g for 7 min at 4 °C and resuspended in flow cytometry staining buffer (00-4222-26, ThermoFisher) to generate single cell suspensions. Cell suspension were incubated with according flow cytometry antibodies, and were washed and resuspended in FACS buffer before analysis. For each experiment, a compensation was developed using signle staining controls.<br>EDTA Blood was prepared using Histopaque Gradient (1500 xg for 30min). The buffy coat was isolated, washed at 500 g for 7 min at 4 °C and resuspended in flow cytometry staining buffer (00-4222-26, ThermoFisher) to generate single cell suspensions. Cell suspension were incubated with according flow cytometry antibodies, and were washed and resuspended in FACS buffer before analysis. For each experiment, a compensation was developed using signle staining controls. |
| Instrument | Data were acquired by a spectral flow cytometer (Nortnern Light, CYTEK, USA). |
| Software | Data were analyzed with FlowJo (v. 10.6). |
| Cell population abundance | Does not apply. No FACS sorting used. |
| Gating strategy | For all cell types, initial forward scatter versus side-scatter were adjusted to include all leukocyte. Forward scatter-A versus forward scatter-H were used to gate singlets. Dead cells were then excluded using Zombie NIR Fixable Viability Kit (423106, Biolegend). Cell populations were gated on live cells and defined as inflammatory monocytes: CD45+ CD11b+ Ly6Chigh CCR2 +; activated macrophages: CD45+ CD11b+ F4/80+ MCH II+; infiltrating leukocytes: CD45 eFluor450+ APC-Cy7-. Data were presented as percentage of specific cell populations or calculated as cell numbers from total live cells. |

☒ Tick this box to confirm that a figure exemplifying the gating strategy is provided in the Supplementary Information.

# Magnetic resonance imaging

## Experimental design

| | |
|---|---|
| Design type | Fixed interval imaging after stroke. |
| Design specifications | MRI was performed in a small animal scanner (3T nanoScan® PET/MR, Mediso, with 25 mm internal diameter quadrature mouse head coil) at 2 and 7 days after sham or stroke surgery. For scanning, mice were anesthetized with 1.2% isoflurane in 30 % oxygen/70 % air applied via face mask. Respiratory rate and body temperature (37 +/- 0,5 °C) were continuously monitored via an abdominal pressure sensitive pad and rectal probe and anaesthesia adjusted to keep them in a physiological range.<br>For Thrombus detection MRI was performed in a small animal scanner (BioSpec 7-T TEP-MRI system, and a surface coil (Bruker, Germany)) at 6 and 24h after NET DNA challenge. Mice were anesthetized using Isoflurane in a mixture of O2/ |

N2O (30/70) and kept under anesthesia during all the procedure, while maintaining a body temperature of 37°C. Prior to MRI, mice were subjected to caudal vein catheterization for DNA and M3P administration. Imaging data were obtained using a TOF sequence to visualize vascular structures, a T2*-weighted sequence for iron-sensitive imaging, and a T2-weighted sequence for tissue contrast. The MRI parameters were set at TR/TE = 12 ms/4.2 ms for the TOF sequence, TR/TE = 50 ms/8.6 ms for the T2*-weighted sequence, and TR/TE = 3500 ms/40 ms for the T2-weighted sequence. T2*-weighted images are presented as stack of 4 slices (minimum intensity), after bias fields correction using ImageJ software.

| Behavioral performance measures | This study did not include behavioral performance measures. |

## Acquisition

| Imaging type(s) | Structural, diffusion. |

| Field strength | Small animal scanner (3T nanoScan® PET/MR, Mediso, with 25 mm internal diameter quadrature mouse head coil). BioSpec 7-T TEP-MRI System, and a surface coil (Bruker, Germany) |

| Sequence & imaging parameters | The following sequences were obtained: coronal T2-weighted imaging (2D fast-spin echo (FSE), TR/TE = 3000/57.1 ms, averages 14, resolution 167 x 100 x 500 µm3), coronal T1-weighted imaging (2D fast-spin echo (FSE), TR/TE = 610/28,6 ms, averages 14, resolution 167 x 100 x 500 µm3)

For thrombus detection, MRI parameters were set at TR/TE = 12 ms/4.2 ms for the TOF sequence, TR/TE = 50 ms/8.6 ms for the T2*-weighted sequence, and TR/TE = 3500 ms/40 ms for the T2-weighted sequence |

| Area of acquisition | Mouse brain and carotid arteries |

Diffusion MRI ☒ Used ☐ Not used

Parameters | DWI (2D spin echo (SE), TR/TE = 1439/50 ms , averages 4, resolution 167 x 100 x 700 µm3). |

## Preprocessing

| Preprocessing software | Nucline nano Scan 3.04.014.0000; ImageJ software |

| Normalization | *If data were normalized/standardized, describe the approach(es): specify linear or non-linear and define image types used for transformation OR indicate that data were not normalized and explain rationale for lack of normalization.* |

| Normalization template | *Describe the template used for normalization/transformation, specifying subject space or group standardized space (e.g. original Talairach, MNI305, ICBM152) OR indicate that the data were not normalized.* |

| Noise and artifact removal | *Describe your procedure(s) for artifact and structured noise removal, specifying motion parameters, tissue signals and physiological signals (heart rate, respiration).* |

| Volume censoring | *Define your software and/or method and criteria for volume censoring, and state the extent of such censoring.* |

## Statistical modeling & inference

| Model type and settings | *Specify type (mass univariate, multivariate, RSA, predictive, etc.) and describe essential details of the model at the first and second levels (e.g. fixed, random or mixed effects; drift or auto-correlation).* |

| Effect(s) tested | *Define precise effect in terms of the task or stimulus conditions instead of psychological concepts and indicate whether ANOVA or factorial designs were used.* |

Specify type of analysis: ☒ Whole brain ☐ ROI-based ☐ Both

| Statistic type for inference

(See Eklund et al. 2016) | *Specify voxel-wise or cluster-wise and report all relevant parameters for cluster-wise methods.* |

| Correction | *Describe the type of correction and how it is obtained for multiple comparisons (e.g. FWE, FDR, permutation or Monte Carlo).* |

## Models & analysis

| n/a | Involved in the study |
| ☒ | ☐ Functional and/or effective connectivity |
| ☐ | ☒ Graph analysis |
| ☒ | ☐ Multivariate modeling or predictive analysis |

| Graph analysis | *Report the dependent variable and connectivity measure, specifying weighted graph or binarized graph, subject- or group-level, and the global and/or node summaries used (e.g. clustering coefficient, efficiency, etc.).* |

