## [Peer Review File · Nature]

Manuscript Title: DNA-sensing inflammasomes cause recurrent atherosclerotic stroke

Reviewer Comments & Author Rebuttals

Reviewer Reports on the Initial Version:

Referees' comments:

Referee #1 (Remarks to the Author):

The authors confirm using contemporary data the high rate of recurrent stroke in the very early period following an ischemic stroke associated with large artery atherosclerosis versus other forms of stroke. They develop a mouse model in which carotid artery atherosclerosis is increased by flow disruption in Apoe^{-/-} mice and then show that unilateral stroke induction promotes contralateral stroke formation. They provide evidence that the underlying mechanism is related to formation of unstable atherosclerotic plaques that undergo rupture and promote thrombus formation in the contralateral artery, which may be related to activation specifically of the AIM2 inflammasome induced by increased circulating DNA. MMP activity is increased in the plaques and may contribute to plaque disruption, and also plaque undergo pro-thrombotic changes related to circulating DNA. Importantly contralateral stroke is reduced by Caspase-1 inhibition or DNase injection consistent with a role of inflammasome in this process, suggesting a therapeutic approach to early recurrent stroke. Comparing asymptomatic carotid plaques to carotid plaques from stroke patients they confirm several features of the pre-clinical model such as increased monocyte/macrophages, increased circulating DNA and increased inflammasome priming in the plaques.

General Comments-

The mouse model of early stroke recurrence and its potential prevention by therapeutic interventions is novel and interesting and the studies are generally well performed. However, the evidence that early recurrence of stroke is caused by rupture of an atherosclerotic plaque in the contralateral artery is incomplete and not very convincing. The main finding in plaques seems to be an increased content of myeloid cells, rather than other features of plaque destabilization or rupture. Although in a different model, this is somewhat expected given previous studies showing that MI increases myelopoiesis and worsens atherosclerosis in part by infiltration of inflammatory myeloid cells (Ref 9, Dutta et al). The evidence that plaque destabilization is related to activation of the AIM2 inflammasome by circulating DNA is incomplete and needs support by a genetic model. Overall, the findings are interesting but the mechanism of early stroke recurrence has not been convincingly shown.

Specific Comments:

1) The images of plaque rupture are unconvincing as there is no thrombus contiguous to a plaque with breakdown of a fibrous cap. In Fig 1D what is the incidence of thrombus in the ACA? Is this thrombus specifically only seen in association with atherosclerotic plaque? Fig 1E. The vulnerability index does not predict plaque rupture in mice since this is generally a rarity. In this study the

increase in this index may simply reflect increased myeloid cells in the plaque so it provides little additional information. In Suppl Fig 3 the images of plaque indicate that the major change is an increase in the content of CD68+ macrophages, without alteration in necrotic core or collagen content. This would be consistent with earlier studies showing an increase in myelopoiesis post MI leading to increased plaque. The images purporting to show ruptured fibrous caps (Fig 3E) rather are showing irregularities of the endothelium without assessment of the fibrous cap. Overall, these images are not convincing for atherosclerotic plaque rupture.

Suppl Fig 2 is not mentioned in the text.

2) The evidence for inflammasome activation is weak. Increased inflammasome priming is shown but may simply reflect increased entry of myeloid cells into plaque and may not be increased relative to the increase in these cells. The AIM2 inhibitor data is interesting but needs to be backed up with a genetic model. Fig 2A, E and H show increased inflammasome priming (increased IL-1b and pro-caspase-1) in the carotid lesions, which may be explained simply by an increased content of inflammatory myeloid cells in the plaque. However, the active form of caspase-1 (p20) does not appear to be increased relative to the pro-form, therefore increased inflammasome activation has not been shown. P20 amount should not be normalized to actin but rather shown as a ratio of p20/pro-caspase-1. The direct evidence for involvement of AIM2 is based on use of an inhibitor that is described in a preprint. This needs to be supported by experiments in mice with genetic deficiency of AIM2 in myeloid cells.

3) Although the idea that increased circulating DNA can activate the macrophage AIM2 inflammasome has been proposed previously the evidence is lacking and the conjecture may not be completely credible. The activation of AIM2 inflammasome in cultured macrophages requires introduction of double stranded DNA (such as polydA:dT) in liposomes in order to access the cytosolic AIM2 sensor. Can the authors show that dsDNA such as they injected or isolated from patient plasma can activate AIM2 in cultured macrophages?

4) The data in Fig 3 suggesting that increased MMP activity contributed to plaque rupture would be more compelling if the authors had shown a breakdown of collagen in the surface of the plaque. IL-1b has a small effect on MMP expression – was IL-1b made in bacteria and perhaps contaminated with bacterial products. It would be useful to show the parameters in Fig 3 were reversed by AIM2 deficiency.

5) Again Fig 4D shows an increase in inflammasome priming but not activation. If dsDNA were activating the AIM2 inflammasome, p20 would be increased relative to pro-caspase-1.

Referee #2 (Remarks to the Author):

In the document entitled “Stroke induces early recurrent events by inflammasome-dependent atherosclerotic plaque rupture”, Cao, Roth and colleagues investigated the underlying mechanisms leading to increased vascular risk and recurrent stroke. This is an interesting manuscript with potential on the mechanistic understanding of stroke, but less convincing on the translational consequences of these observations.

I focused my review on the cell-free DNA results from the manuscript, and have the following comments:

1. The characterization of “cell-free DNA” is not technically convincing. First the quantification is performed by different methods (“total DNA” and “ssDNA” by nanodrop, “dsDNA” by Qubit). But none of these methods are completely specific to the targeted stranding. Also, viewing the jaggedness of cfDNA, it is unclear how accurate are the cfDNA measurements. Finally, because no structural or size analysis is performed, the authors cannot confirm that they are detecting “cfDNA” (this could be contaminating genomic DNA or EV DNA in the bloodstream).
2. Numerous studies have previously investigated cfDNA for the diagnostic and management of stroke (or myocardial infarction), and the potential impact of nuclease treatment. This is reducing the novelty of the work related to the observations. See for example: PMID: 34991335; PMID: 36928073; PMID: 31978408; PMID: 29691397; PMID: 31876653 (among many other).
3. The link between the mechanistic paragraphs (corresponding to Fig1-3) and translational paragraph (corresponding to Fig4) is unclear. The demonstration that stroke increases plaque inflammation and MMP activity in patients is unrelated to their potential response to a DNase treatment (which is potentially the major novelty of the work).
4. As a large fraction of DNA in bloodstream is contained in vesicles (e.g. exosomes, apoptotic bodies, etc thus protected from nucleases), how can a simple DNase treatment lead to a clinical effect? Could this suggest that the treatment is actually acting on a subpopulation of “extracellular DNA” population, more than the bulk “cell-free DNA population”? Could it be neutrophil extracellular traps? In general, this could be more detailed as it could lead to misinterpretation in the mechanism involved.
5. Because the cfDNA methods used to evaluate differences between asymptomatic and stroke patients are unspecific, they can be biased by the pre-analytical conditions or experimental conditions used. Based on the method section in the supplementary documents it is unclear if such conditions could alter the observations made in Figure Figure 2G and Figure 4. The authors should clarify this.
6. Most of the mechanistic work is demonstrated (logically) on animal models. This is a limitation of the work conclusions, which is not clearly stated in the text. The author should also discuss how a translation into clinic from their observation could be achieved (and if this is possible).

7. Because alterations in the nuclease activity leads to specific marks in the cell-free DNA structure, could this be used as an early marker for stroke detection (cf <https://doi.org/10.1016/j.tig.2021.04.005>)? In general the generation of cfDNA involve a group of nucleases, and the potential use of a group of nucleases could further increase the treatment efficiency. This could maybe be further discussed or investigated.

Referee #3 (Remarks to the Author):

To Authors: This is a potentially exciting study that may provide a mechanism underlying a major clinical problem, i.e. secondary cardiovascular events in stroke patients. The hypothesis is interesting, the data are relevant, and the analyses are mostly reasonable. However, there are a few major issues that require further consideration.

1. Some added controls may be required. Experiments compare CCA-TS only vs CCA-TS plus stroke mice. What about stroke only mice? Are there thrombi that also occur days or weeks later? Where? What happens to inflammasomes, caspases, MMPs, circulating free cfDNAs etc? What happens to the carotids in mice subjected to stroke-inducing surgery alone?
2. Seems like the secondary strokes occur in the ACA territory? Is this surprising since the major flow routes should come into MCA territories? What are the percentage distributions of secondary stroke locations?
3. The authors show secondary thrombi in CCA-TS plus stroke mice. Should one see these in the more-upstream ACA territories instead of the large vessels in the base of the brain? Importantly, these data should be quantified in terms of amount, location of thrombi in various vascular regions and how they might correspond to location of severity of the secondary strokes. It may also be helpful to compare these numbers between all three experimental groups (CCA-TS, stroke, CCA-TS plus stroke).
4. The authors try to make a distinction between NLRP3 vs AIM2, stating that only inhibition of AIM2 but not NLRP3 prevented inflammasome activation. However, in the western blot shown in Fig 2E, it seems that inhibition of NLRP3 also reduces caspase-1 activation.
5. There may be important questions about timing. cfDNA is very rapidly increased but this is transient, and levels essentially returns back normal by 24 h post-stroke. In contrast, the recurrent events were detected at 7 days post-stroke. How does a very early transient signal cause such a delayed response?
6. The authors try to prove cfDNA causality by injecting this into mice. More details are needed. How much, exactly how, and when are these injections performed? Importantly, how can one equate the in vivo DNA challenge treatment (50 ug) to the endogenous levels of stroke-induced upregulation of cfDNA (600-800 ng/ml)?
7. In Fig 3, besides using serum, does DNA challenge also induce MMP activation? Moreover, does DNAase eliminate stroke-induced upregulation of MMP and Factor XIIIa?
8. Finally, there is a critical question regarding the proposed cfDNA mechanism. Many other factors are known to induce plaque instability, including cytokines. Many of these cytokines (e.g. IFN, IL6 etc) are known to be upregulated after stroke. How can the authors prove that cfDNA is the trigger and not these other cytokines?

Author Rebuttals to Initial Comments

Stroke induces early recurrent vascular events by inflammasome-dependent atherosclerotic plaque rupture

We would like to thank the reviewers and editor for the overall very positive evaluation of our manuscript and the constructive comments. We have performed a multitude of additional experiments, included new and re-analyzed data and have substantially revised the manuscript.

Particularly, we performed several additional experiments to confirm AIM2 inflammasome activation as the key inflammatory pathomechanism using genetic models and bone-marrow transplantation experiments. We deeply phenotyped effects on atherosclerotic plaque morphology by additional histological analyses and in vivo imaging of atherothrombosis. Finally, we were able to identify the unexpected source of cfDNA after stroke – neutrophil NETosis – in stroke patients by methylome analysis and confirmed by several new in vivo experiments in animal models NETosis as a druggable mechanism after stroke initiating the cfDNA-mediated inflammasome activation cascade.

We are confident that with the help of the reviewers' comments the revised manuscript has been substantially improved and hope that it will be acceptable for publication. Please find below a point-by-point reply to the reviewers' individual comments:

Referee #1:

General Comments

The mouse model of early stroke recurrence and its potential prevention by therapeutic interventions is novel and interesting and the studies are generally well performed. However, the evidence that early recurrence of stroke is caused by rupture of an atherosclerotic plaque in the contralateral artery is incomplete and not very convincing. The main finding in plaques seems to be an increased content of myeloid cells, rather than other features of plaque destabilization or rupture. Although in a different model, this is somewhat expected given previous studies showing that MI increases myelopoiesis and worsens atherosclerosis in part by infiltration of inflammatory myeloid cells (Ref 9, Dutta et al). The evidence that plaque destabilization is related to activation of the AIM2 inflammasome by circulating DNA is incomplete and needs support by a genetic model. Overall, the findings are interesting but the mechanism of early stroke recurrence has not been convincingly shown.

Response: We thank the reviewer for the overall positive assessment and acknowledgement of the novelty of the used animal model and proposed mechanism. Prompted by the reviewer's suggestions, we performed several new experiments including use of genetic models, bone marrow transfer, in vivo imaging and therapeutic interventions to define the exact mechanisms in much greater detail than originally described. We believe that these additional findings confirm the previously proposed concept and also expand to include new important information, including for example the new identification of the post-stroke cfDNA source being primarily from circulating neutrophils and not from the injured brain site.

Specific comments

1) The images of plaque rupture are unconvincing as there is no thrombus contiguous to a plaque with breakdown of a fibrous cap. In Fig 1D what is the incidence of thrombus in the ACA? Is this thrombus specifically only seen in association with atherosclerotic plaque? Fig 1E. The vulnerability index does not predict plaque rupture in mice since this is generally a rarity. In this study the increase in this index may simply reflect increased myeloid cells in the plaque so it provides little additional information. In Suppl Fig 3 the images of plaque indicate that the major change is an increase in the content of CD68+ macrophages, without alteration in necrotic core or collagen content. This would be consistent with earlier studies showing an increase in myelopoiesis post MI leading to increased plaque. The images purporting to show ruptured fibrous caps (Fig 3E) rather are showing irregularities of the endothelium without assessment of the fibrous cap.

Response: We agree with the reviewer, that in the interest of brevity, the morphological features of plaque destabilization have not been sufficiently clearly described in the original manuscript. Therefore, we include several new analyses in the revised manuscript to address all the reviewer's specific concerns listed in this comment, which we want to address individually below.

In response to the statement that "The vulnerability index does not predict plaque rupture in mice since this is generally a rarity", we want to clarify that while this is surely true for common atherosclerosis models (e.g. ApoE- or LDL-R.- deficient mice on high-fat diet), we specifically established for this study an animal model of rupture-prone plaques in the common carotid artery (CCA). This was achieved by a tandem ligation which leads to turbulent blood flow with exacerbated concentric plaque growth and large necrotic cores – resembling very closely high-risk, rupture prone plaques in humans. A detailed characterization of this animal models is provided in the revised Extended data Figure 2. Yet, to further confirm that the phenomenon of plaque rupture and secondary infarction after stroke is specific for our novel animal model, we repeated the same experimental layout and systematic screening for secondary infarctions in conventional atherosclerotic ApoE-KO mice on 8 weeks of high-fat diet. We neither observed in these animals stenotic CCA plaques (macroscopic image in **A**) and none of the animals displayed secondary cerebral infarctions (**B**). This additional data set has been included in the revised manuscript in Extended data Figure 2J.

i) Reduction in cap thickness: We measured fibrous cap thickness using picro sirius red stainings on CCA sections from *ApoE*^{-/-} mice with tandem stenosis one week after stroke or control surgery. Fibrous cap thickness was defined as the mean distance cap thickness measurements according to Silvestre-Roig et al. Nature, 2019. Both, fibrous cap thickness and the ratio to the vascular lumen area

were significantly decreased after stroke. This additional data has been added to the revised Extended data Figure 3.

ii) Fibrous cap rupture and atherothrombosis at CCA plaques: As suggested by the reviewer, we quantified the occurrence of physical plaque rupture and intraluminal thrombus apposition. Physical plaque ruptures were defined as discontinuities in the picro sirius red-labeled fibrous cap (A) with additional confirmation using collagen I staining (B); intraluminal thrombus formation (E) was identified by staining for CD41 (platelets, C) and von Willebrand Factor (vWF, D). Only if all these four histological criteria were fulfilled, we rated this as an event of plaque rupture with atherothrombosis. We found in 80% of mice after stroke with a secondary ischemic brain lesions signs of atherothrombosis, while this was detected only in one case of the animals without secondary brain infarction (F). This highly significant correlation between atherothrombosis and secondary brain ischemia clearly supports the concepts of inflammatory plaque rupture as the cause of recurrent stroke events in our animal model. This new data set was included in the revised Figure 1d.

iii) Fibrous cap collagen fiber orientation: To further study morphological features of fibrous cap destabilization, we performed morphological analysis of collagen fiber orientation. Using picro sirius red stainings of CCA sections, we visualized fibrillar collagen I which greatly contributes to the fibrous cap stability (A). Next, we quantified the aspect ratio (AR) of the fiber orientation based on the

anisotropy where an AR > 1 shows predominant orientation and values closer to AR ~ 1 indicate random orientation (according to: Tsai et al., J Biomed Opt, 2010 and Cicchi et al., J Biophoton., 2010). 7d after stroke, we detected a significant reduction in AR values compared to fibrous caps of sham-operated animals, indicating remodeling and destabilization of the fibrous caps after stroke. This new data set has been added to the revised Figure 4a.

iv) PCA analysis of vulnerability index factors: We believe that the reviewer's perception of myeloid cell accumulation in plaques after stroke being the main feature contributing to the plaque vulnerability index might be due to our lack of presenting more clearly the individual features contributing to this index in the original manuscript.

In a principal component analysis (A) comparing the contribution of the five different parameters to the vulnerability index between sham and stroke-operated animals (with and without secondary lesions), we observed that collagen and smooth muscle actin (SMA) content of the intima area were the strongest predictors of group allocation in the principal component 1, followed by CD68⁺- necrotic core (NC)- and fibrous cap (FC)- area in approx. similar weight to the component loading (B). Hence, while CD68 macrophage accumulation is indeed an important factor, it is neither the most dominant variable and definitely not the exclusive feature driving the vulnerability index. This additional analysis has been included in the revised Extended data Figure 3f.

v) In vivo imaging of atherothrombosis: Additionally, we aimed to detect in vivo the process of atherothrombosis at the site of the rupture-prone CCA plaque. Therefore, we made use of the recently established in vivo imaging modality of intravascular thrombi using MPIO-based matrix microparticles (Lizzaroni et al., Science Advances, 2022). We performed in vivo MRI imaging of mice undergoing our CCA TS model and acquired TOF sequences to visualize vascular structures (MR-angiography of the cerebral vasculature) and T2*-weighted sequence for iron-sensitive imaging (i.e. MPIO-bearing intravascular thrombi) before and after i.v. DNA injection. A detailed description of the methodology is provided in the revised methods section.

With this imaging modality, we were able to confirm intra-stenotic atherothrombosis in the CCA resulting in vascular occlusion in the angiography (red circle is indicating hypoperfused right MCA territory); additionally, we detected more distal thrombotic material in MCA branches (arrowheads in right images). This new data set has been included in the revised Figure 4i.

vi) Distribution of secondary infarctions in vascular territories: As suggested by the reviewer, we analyzed in a cohort of 24 mice secondary ischemic lesions in the contralateral hemisphere to the primary stroke the distribution of secondary lesions to the cerebrovascular territories. As expected, the majority of lesions were found in the middle cerebral artery (MCA) territory, with only 3 lesions in the anterior cerebral artery (ACA) and only one in the watershed area between middle and posterior cerebral artery territories.

2) The evidence for inflammasome activation is weak. Increased inflammasome priming is shown but may simply reflect increased entry of myeloid cells into plaque and may not be increased relative to the increase in these cells. The AIM2 inhibitor data is interesting but needs to be backed up with a

genetic model. Fig 2A, E and H show increased inflammasome priming (increased IL-1b and pro-caspase-1) in the carotid lesions, which may be explained simply by an increased content of inflammatory myeloid cells in the plaque. However, the active form of caspase-1 (p20) does not appear to be increased relative to the pro-form, therefore increased inflammasome activation has not been shown. P20 amount should not be normalized to actin but rather shown as a ratio of p20/pro-caspase-1. The direct evidence for involvement of AIM2 is based on use of an inhibitor that is described in a preprint. This needs to be supported by experiments in mice with genetic deficiency of AIM2 in myeloid cells.

Response: We provide in the revised manuscript several additional data sets and a re-analysis of the caspase-1 WB analysis to validate post-stroke inflammasome activation and specificity of the AIM2 inflammasome. Normalization of p20/pro-Casp1 as well as the cell-based analysis using FAM-FLICA FACS analysis confirmed increased inflammasome activation independent of increased myeloid cell infiltration. Using genetic deficiency for AIM2 (as well as ASC, cGAS, NLRP1 and NLRP3) for in vitro studies and in vivo bone marrow experiments, we confirmed that the AIM2 inflammasome is the main inflammasome in the observed phenomenon. We additionally confirmed direct inflammasome assembly by demonstration of AIM2-dependent ASC oligomerization. The used AIM2 inhibitor [calix(6)arene] has by now been published; additionally, we present novel (unpublished) data on its dose-dependent displacement of DNA from the AIM2 DNA-binding site. Please see the detailed replies to each of the listed comments below including the newly generated data sets:

i) Increase in p20 active Caspase-1: As suggested by the reviewer, we re-analyzed all western blot quantifications for caspase-1 and present results now as p20/pro-caspase-1 ratios throughout the revised manuscript. None of the results and their interpretation has been changed by this modification in the analysis; in contrast, stroke effects on inflammasome activation is even more pronounced in this analysis than the previous p20/actin normalization. This analysis shows a significant increase in caspase-1 cleavage after stroke (A), and treatment with the caspase-1 inhibitor VX765 after stroke led to a decrease in caspase-1 cleavage (B). Pharmacological inhibition with an AIM2 inhibitor (calix[6]arene) and to a lesser degree with a NLRP3 inhibitor (MCC950) also significantly reduced the p20/pro-caspase-1 ratio. Conversely, a single bolus of i.v. DNA administration significantly increased this ratio (D). Results of this revised analysis are presented in Figure 2g, j and Extended data Figure 6. Original uncropped western blots for all data presented is displayed in Suppl. information 1.

ii) Single-cell based analysis of inflammasome activation by FACS: To further validate that inflammasome activation (i.e. caspase-1 cleavage) is indeed functionally increased in CCA plaques after stroke and results are not skewed by changes in cellular composition, we performed an

additional FACS-based analysis of active Caspase-1 using the intracellular staining with the fluorescent inhibitor probe 660-YVAD-FMK (FAM660). From CCA plaque cell suspensions at 24h after sham or stroke surgery we analyzed equal cell numbers of CD45+CD11b+ myeloid cells (100 cells per sample) from all samples and analyzed the proportion of FAM660-positive cells and the mean fluorescence intensity for FAM660. Both, the percentage of myeloid cells with active caspase-1 as well as caspase-1 activity (i.e. MFI) was significantly increased in the stroke compared to the sham group. These findings have been added to the revised manuscript in Figure 2b.

iii) Inflammasome activation in human monocytes by stroke serum:

[REDACTED]

By treating primary human monocytes with serum from stroke or control patients, we observed that serum from stroke patients was sufficient to lead to caspase-1 cleavage in human monocytes (see below, from Figure 3b in above-referenced manuscript).

[REDACTED]

iv) ASC oligomerization depends on AIM2: Additionally, we analyzed ASC oligomerization as direct evidence of physical inflammasome formation beyond caspase-1 cleavage. Here, we primed WT or AIM2^{-/-} BMDMs with 100ng/ml LPS for 4h and subsequently treated the cells with 250ng DNA for 1h. WT and AIM2-deficient BMDMs were equally efficiently primed as represented by a prominent ASC monomer band in both genotypes. However, the formation of ASC dimers, tetramers and larger oligomers was observed only in WT cells and was completely absent in AIM2^{-/-} BMDMs. These findings support the hypothesis that AIM2 is a key inflammasome involved in sensing cell-free DNA release after stroke and are presented in the revised manuscript in Figure 2k.

v) Inflammasome activation in BMDMs by stroke serum depends on AIM2 and ASC: We characterized the stimulation of bone marrow-derived macrophages with stroke or sham serum for different inflammasome sensor deficiencies using NLRP1, NLRP3, cGAS, ASC and AIM2 knockouts. We acquired caspase-1 cleavage in the macrophage lysates. Furthermore, we checked NET DNA stimulation in WT vs AIM2-deficient macrophages. We observed that only ASC and AIM2 deficiency prevented inflammasome activation (i.e. increase in p20/pro-caspase-1 ratio) after stroke serum treatment. Correspondingly, caspase-1 cleavage was completely blunted in AIM2^{-/-} BMDMs in response to DNA treatment. These new findings are now presented in the revised manuscript in Extended data Figure 8c, d.

vi) In vivo inflammasome activation is AIM2 dependent: In addition to the genetic in vitro studies described above, we also performed an in vivo genetic validation experiment by performing bone marrow transfer from WT or AIM2^{-/-} animals to atherosclerotic recipient mice. For this, we used a well-established genetic depletion model (Mx1^{Cre}c-myb^{fl/fl}; Stremmel et al. 2018, J Immunol Methods) on LDLr^{-/-} background with high-fat diet. Due to tamoxifen induction of c-myb under the MX1 promoter, the bone marrow of these mice was depleted and WT or AIM2^{-/-} bone marrow was transplanted. 10 weeks later mice received an i.v. DNA challenge and inflammasome activation was characterized 24h after the stimulus. Flow cytometry for active caspase-1 (FAM660) revealed that the adoptive transfer of AIM2^{-/-} bone marrow to the LDLr^{-/-} mice led to a pronounced decrease in caspase-1 activation of myeloid cells (WT: 13.1% caspase-1+; KO: 1.57-2.2% caspase-1+) – confirming that myeloid cells' AIM2 inflammasome is required for the post-stroke systemic inflammasome activation. These new results are now presented in the revised manuscript in Figure 2l.

We acknowledge the reviewer's concern regarding the mention of "support by a genetic model" in the context of assessing the impact of monocyte-specific AIM2 deficiency on CCA plaque rupture. To conduct such an investigation, a monocyte-specific, inducible AIM2 deficiency model would be necessary to avoid affecting primary plaque development before stroke or sham surgery. This would entail the development of an inducible, triple-transgenic mouse strain on a homozygous ApoE or LdlR knockout background. While it is theoretically feasible to create such a genetic animal model, extensive inquiries and discussions within the scientific atherosclerosis community have revealed that, as of now, such a model does not readily exist. Furthermore, there is currently no suitable inducible cre-driver line established on an atherogenic background. Consequently, to pursue this research avenue, we would have been required to initiate the generation of a completely new triple-transgenic line in our facility. This process, including generation, back-crossing, and conducting experiments, would demand approximately 18-24 months. Ultimately, this substantial effort would have yielded only a limited amount of additional information compared to the combined experiments we have already conducted for this revision, as outlined above. We maintain our confidence that the cumulative results from various aspects of our research, including AIM2-dependent ASC oligomerization (iv), in vitro stimulation experiments with specific inflammasome-deficient BMDMs (v), AIM2-deficient BM transplant (vi), and the rigorous validation of the AIM2 inhibitor's specificity (vii, as detailed below), collectively demonstrate the critical involvement of the AIM2 inflammasome in myeloid cells in post-stroke inflammation and plaque rupture.

vii) characterization of calix[6]arene as an AIM2 inhibitor: We understand the skepticism of the reviewer towards the previously unpublished AIM2 inhibitor calix[6]arene for the original manuscript submission. The compound and its detailed characterization has by now been published (Green et al., iScience, 2023). Additionally, we performed new experiments to particularly address the specificity of calix[6]arene to inhibit DNA binding to AIM2. We performed electrophoretic mobility shift assays (EMSA) for detecting AIM2 – DNA interactions. Here, 72bp FAM-tagged fluorescent dsDNA was incubated with recombinant human AIM2-GST for 5min and then calix[6]arenes were added. Finally, a TBE gel was run and fluorescently imaged. In the blot below (A), "Free DNA" represents dsDNA which has been displaced from AIM2, migrating faster through the gel due to its smaller size. "AIM2:DNA" is dsDNA complexed with AIM2 and trapped at the top of the gel. We observed a dose-dependent effect of calix[6]arene to displace dsDNA from AIM2. This finding has been reproduced in three individual experiments and its quantification is shown below in panel B. These new results are presented in the revised manuscript in the Extended data Figure 7a, b.

3) Although the idea that increased circulating DNA can activate the macrophage AIM2 inflammasome has been proposed previously the evidence is lacking and the conjecture may not be completely credible. The activation of AIM2 inflammasome in cultured macrophages requires introduction of double stranded DNA (such as polydA:dT) in liposomes in order to access the cytosolic AIM2 sensor. Can the authors show that dsDNA such as they injected or isolated from patient plasma can activate AIM2 in cultured macrophages?

Response: We thank the reviewer for this important comment. Indeed, to our knowledge the uptake mechanisms of cell-free DNA into monocytes/macrophages for sensing by AIM2 is currently still largely unclear.

We are confident that we were able to demonstrate in this study – supported by numerous new data sets presented in detail above in the response to comment 2 – that cell-free DNA can lead to AIM2-dependent caspase-1 activation in macrophages. In a detailed characterization of the source of cell-free DNA after stroke, we newly identified that most cell-free DNA is derived from neutrophils as a result of post-stroke NETosis. Please see the response and several new data sets on this aspect in reply to comments by Reviewer 2. Therefore, it is likely that dsDNA complexed with nuclear proteins (histones, HMGB1, etc.) in such NET-DNA can engage active endocytosis-mechanisms to enter the macrophage cytosol as it has been shown for example that HMGB1 can represent such a shuttling molecule for extracellular ligands (e.g. Deng et al., *Immunity*, 2018).

Nevertheless, we also performed an additional experiment in which we aimed to directly visualize the intracellular translocation of fluorescently labeled NET-DNA into BMDMs (to reproduce the same culture and stimulation conditions for which we have proven inflammasome activation, see above). For this, we labeled NET-DNA derived from activated neutrophils with the fluorophore draq5 and added to the culture medium of BMDMs (A). Next, we visualized NET-DNA uptake by co-imaging draq5-labeled NET-DNA with nuclear DNA (Hoechst) and ASC speck formation using BMDMs from an ASC-citrine reporter mouse strain (B6.Cg-Gt(ROSA)26Sor^{tm1.1(CAG-Pycard/mCitrine*, -CD2*)Dtg/J}; Tzeng et al., *Cell Reports*, 2016). Here, we observed intracellular NET-DNA in colocalization with ASC specks in the cytosol (C-G). Importantly, no liposomes or other transfectants were needed for this intracellular uptake of NET-DNA by macrophages. This new data set clearly demonstrates the efficient and rapid translocation of extracellular NET-DNA into macrophages and its co-localization with the inflammasome. This data set has been added to Figure 3e of the revised manuscript.

Moreover, we analyzed NET DNA uptake in a live cell imaging setup. After adding NET DNA to the BMDMs, we detected a rapid increase in draq5-labeled exogenous NET DNA in WT BMDMs, illustrating the efficient uptake of extracellular NET DNA. Approximately 60min after NET DNA addition, we detected a steep increase in ASC citrine fluorescence with a cytosolic speck confirmation (i.e. ASC speck = physical inflammasome formation). Shortly after speck formations, cells showed signs of pyroptotic cell death including blebbing/bursting and loss of membrane integrity. This pyroptotic cell death can be noticed in the graph as the sharp decrease of fluorescence in all channels (CD11b, NET DNA and ASC Citrine). Please find a representative video sequence of the described DNA uptake dynamics between the 50-120 minute timeframe from the graph below here: <https://youtu.be/3QNI32dpFbw>.

4A) The data in Fig 3 suggesting that increased MMP activity contributed to plaque rupture would be more compelling if the authors had shown a breakdown of collagen in the surface of the plaque.

Response: We performed a more detailed analysis of the fibrous plaque composition as described in the reply to comment 1. Particularly, we performed an analysis of collagen fiber orientation in the fibrous cap, demonstrating increased randomness of fiber orientation as a sign of extracellular matrix remodeling after stroke. Please see above for detailed explanation and data set.

4B) IL-1b has a small effect on MMP expression – was IL-1b made in bacteria and perhaps contaminated with bacterial products.

Response: We believe that the notion of IL-1 having only a small effect on MMP expression is based on the results we had shown in the original manuscript of a dose-dependent effect on MMP2 and MMP 9 mRNA expression by RT-PCR (A). However, MMP activity is only partially regulated by its transcription, a main effect is its activation under inflammatory tissue conditions. Correspondingly, the enzymatic activity of MMP9 as measured in this study by gel zymography or by western blot analysis of cleaved (activated) MMP2/9 isoforms is more informative about MMP “activity”. Of note, inflammasome inhibition as well as DNase treatment significantly reduced these readouts of MMP activity (B, C). These data sets are presented in Figure 4j, l and Extended data Figure 9f, g.

The *in vitro* experiment the reviewer is referring to was performed with a commercial product (Recombinant Mouse IL-1 beta/IL-1F2 Protein; R&D systems, Cat.: 401-ML-005/CF) which was tested to be endotoxin free (<0.01 EU per 1 μ g of the protein by the LAL method) and used in previous studies for similar purposes. Nevertheless, the recombinant cytokine was produced in *E. coli*, so potential additive effects by bacterial contaminants cannot be fully excluded.

To fully address this issue and test the disease-modifying effect of IL-1 β in our model, we performed an additional *in vivo* experiment in which we neutralized circulating IL-1 β by administration of neutralizing monoclonal antibodies and analysis of the treatment effects on CCA plaques 1 week later (A). IL-1 β -neutralization was sufficient to reduce MMP2/9 activity in CCA plaques (in situ zymography, B) and prevented fibrous cap thinning (C). In summary, these findings clearly confirm that IL-1 β is a key effector enzyme in mediating plaque destabilization due to increased MMP activity after stroke. These new results are presented in the revised manuscript in Figure 4d, e.

5) Again, Fig 4D shows an increase in inflammasome priming but not activation. If dsDNA were activating the AIM2 inflammasome, p20 would be increased relative to pro-caspase-1.

Response: The reviewer is referring here to the Western blot analysis of caspase-1 from human CCA samples. As for the corresponding analyses from our mouse model (please see detailed reply and revised results in reply to comment 2), we have performed the requested analysis of cleaved caspase-1 to pro-caspase-1 which did not change the interpretation of the results. For human caspase-1 processing we quantified the transient caspase-1 cleavage species (p33) and final cleavage product p20 (Boucher et al. 2018; JEM) and normalized to full-length pro-caspase-1. With this analysis we observe a significant increase in p33/p20 caspase-1 not only normalized to beta-actin (original analysis) but also normalized to pro-caspase-1. The revised analysis has been added to the revised Figure 5f.

Referee #2:

Specific Comments

1) The characterization of “cell-free DNA” is not technically convincing. First the quantification is performed by different methods (“total DNA” and “ssDNA” by nanodrop, “dsDNA” by Qubit). But none of these methods are completely specific to the targeted stranding. Also, viewing the jaggedness of cfDNA, it is unclear how accurate are the cfDNA measurements. Finally, because no structural or size analysis is performed, the authors cannot confirm that they are detecting “cfDNA” (this could be contaminating genomic DNA or EV DNA in the bloodstream).

Response: Based on the reviewer’s suggestion we used an established automated electrophoresis system (Agilent Bioanalyzer) to acquire the size and distribution of isolated (NORGEN cfDNA isolation Kit) cfDNA from mice after stroke. We acquired cfDNA size 6h after stroke and 6h after stroke in mice treated in vivo with i.v. DNase I (1,000 U). By gel electrophoresis, we confirmed a significant increase in blood cfDNA after stroke (A) as previously already shown in the original manuscript by Nanodrop and Qubit quantification. Moreover, we detected that this increase in total cfDNA was mainly driven by short-fragment cfDNA of less than 400bp (B). Correspondingly, the in vivo DNase treatment significantly reduced the <400bp cfDNA concentration also in this assay (C). This new data set was added to the revised Extended data Figure 7f, g and 8e.

As per the reviewer’s comment, we additionally analyzed the potential contribution of DNA contained in extracellular vesicles (EV) to the total cfDNA increase in the blood after stroke. We performed an extracellular vesicle (EV) spin down and acquired spectrophotometrically the concentration of EV-associated DNA and vesicle-free DNA from blood samples obtained 6h after stroke or sham surgery. We detected an only very minor contribution of EV-associated DNA to total cfDNA (less than 3% after stroke). Moreover, the concentration of EV-DNA remained unchanged between sham and stroke ($p = 0.18$), while the concentration of vesicle-free DNA concentration was significantly increased (sham vs stroke $p = 0.0079$). This new data set was added to the revised Extended data Figure 7h.

2) Numerous studies have previously investigated cfDNA for the diagnostic and management of stroke (or myocardial infarction), and the potential impact of nuclease treatment. This is reducing the novelty of the work related to the observations. See for example: PMID: 34991335; PMID: 36928073; PMID: 31978408; PMID: 29691397; PMID: 31876653 (among many other).

Response: We fully agree with the reviewer that numerous previous studies have already studied various aspects of cell-free DNA in stroke, myocardial infarction and other tissue injuries mainly as a diagnostic biomarker. However, this was neither the focus nor the main novel aspect of this manuscript and was never claimed as such. In fact, we have ourselves reported in a previous manuscript (Roth et al., *Immunity*, 2021) the increase of post-stroke cfDNA and described in detail its role in mediating the systemic inflammatory response to stroke.

The focus of this study is not primarily on cfDNA but on the inflammatory mechanisms promoting atherosclerotic plaque destabilization and discovering a previously unrecognized cause of the high risk of recurrent vascular events after an incident stroke. Specifically, this is the first study using a novel animal model of rupture-prone atherosclerotic plaques, in which we observed secondary brain infarctions after inflammatory CCA plaque rupture after stroke. While cfDNA release after stroke could have been assumed from previous studies (although also countless other biomarkers are reported to be increased after stroke, including cytokines, sympathetic innervation, brain-derived alarmins, etc.), the inflammatory mechanisms mediated by cfDNA leading to atherosclerotic plaque destabilization (IL-1-mediated MMP expression, plaque rupture and atherothrombosis) were completely unknown and constitute the main novelty of this manuscript. Additionally, the use of DNase for this indication (i.e. inflammatory comorbidities after stroke) is novel and was not demonstrated before, highlighted by the fact that we received a patent on the use of DNase I for treating post-stroke inflammatory comorbidities based on the work presented in this manuscript (Patent number: US20230241185A1 / EP4138881A1).

Moreover, additional experiments performed to address the reviewer's comments derived unexpected and novel information on the source of cfDNA after stroke, which is explained in more detail below in the reply to comment 4. Indeed, we identified that NET-DNA—and not DNA derived from the necrotic brain tissue—is the main source of the increased concentration of circulating cfDNA after stroke in stroke patients and in experimental stroke. These findings considerably expand the information provided on post-stroke cfDNA in the original manuscript and add novel aspects to the proposed mechanisms.

3) The link between the mechanistic paragraphs (corresponding to Fig1-3) and translational paragraph (corresponding to Fig4) is unclear. The demonstration that stroke increases plaque inflammation and MMP activity in patients is unrelated to their potential response to a DNase treatment (which is potentially the major novelty of the work).

Response: We are surprised that the link between the data presented in our novel experimental mouse model of rupture-prone plaques and results derived from human blood and carotid artery plaques samples was not apparent and excuse for the potential lack of clarity in the manuscript.

The human patient samples were obtained from symptomatic and asymptomatic atherosclerotic plaques in the carotid artery, i.e. highly stenotic plaques from patients that either already had an ischemic stroke or not. Thereby, these human samples represent exactly the situation which we mimicked in our novel animal model of highly stenotic, and rupture prone plaques of mice in which we induced a stroke or only a sham surgery.

Importantly, we were able to confirm on the human blood and plaque samples all of the key features of the inflammatory mechanisms leading to plaque destabilization that we discovered in our mouse model: inflammasome activation, MMP activity, myeloid cell accumulation and activation of the F.XII-dependent coagulation cascade. As these were the exact same key readout parameters which we characterized in the animal model, the link between the experimental findings and human validation seemed obvious to us. We have modified the text in the results section as well as in the discussion to explain this match in experimental and human findings more clearly.

Additionally, we performed methylome analysis of human blood-derived cfDNA to study the potential cellular source of circulating cfDNA after stroke in human patients. This analysis and unexpected identification of NET-DNA as the major cfDNA source led again to a series of additional experiments in our mouse model which confirmed this finding and introduced novel therapeutic means by targeting post-stroke NETosis. We expand on this issue below in our reply to comment 4.

Moreover, we thank the reviewer for acknowledging the novelty of the DNase treatment – yet, as stated above the major novelty lies in the description of the inflammatory mechanisms in the atherosclerotic plaque and not primarily in the use of DNase. Nevertheless, we are happy to inform the reviewer that based on this study and our previous report on cfDNA-mediated inflammation after stroke (Roth et al., *Immunity*, 2021), we have already obtained IRB approval and initiated a clinical trial in stroke patients receiving either DNase I or placebo and testing for biomarkers of inflammatory response and clinical outcome (ReScinD Trial, ClinicalTrials.gov ID: NCT05880524). However, conducting and reporting this clinical trial is of course vastly beyond the scope of this mechanistic study but we discuss and reference this trial now in the revised discussion section.

4) As a large fraction of DNA in bloodstream is contained in vesicles (e.g. exosomes, apoptotic bodies, etc thus protected from nucleases), how can a simple DNase treatment lead to a clinical effect? Could this suggest that the treatment is actually acting on a subpopulation of “extracellular DNA” population, more than the bulk “cell-free DNA population”? Could it be neutrophil extracellular traps? In general, this could be more detailed as it could lead to misinterpretation in the mechanism involved.

Response: We thank the reviewer for this knowledgeable remark which prompted us to perform a series of additional experiments, leading to important novel findings and conceptually expanding the manuscript.

First, we performed an analysis of EV- versus -Non-EV-associated cfDNA as reported already above in reply to comment 1, in which we observed that after stroke particularly the EV-free cfDNA fraction was substantially increased while the very minor EV-associated cfDNA fraction remained unchanged (please see above).

Next, we performed a methyloome analysis of isolated cell-free DNA from stroke patient samples in the acute phase after the ischemic event. This analysis is based on the unique DNA methylation pattern in different cell types and enables to identify the tissue and/or cell origin of circulating cell-free DNA as previously reported and used in other indications (Moss et al. 2018 Nature Comm; Loyfer et al. 2023 Nature). Surprisingly, we observed in this analysis that barely any of the blood cfDNA of stroke patients showed the methylation signature of brain cells but was mainly derived from myeloid cells (predominantly neutrophils and to a lesser extent monocytes). These findings indicated that the major source of the acute increase in cfDNA blood concentration after stroke might be due to neutrophil NETosis, which has previously been reported to occur in response to ischemic brain injury (De Wilde, RPTH, 2023). This new data has been included in the revised manuscript in Figure 3a.

Based on these findings, we further studied the potential formation of NETs in the experimental stroke model and its relevance to the observed inflammasome-dependent plaque destabilization after stroke. First, we analyzed NET formation after experimental stroke by flow cytometric quantification of citH3-positive neutrophils and detected an approx. 3-fold increase in stroke compared to sham-operated animals (**A**). Likewise, treatment of primary cultured (naïve) neutrophils with serum obtained from mice after stroke was sufficient to induce massive NETosis in vitro in comparison to treatment with the serum of sham-operated animals, confirming the induction of NET formation by a serum mediator released after ischemic brain injury (**B**). These new data sets were added in Figure 3b-d.

Consequently, to investigate if neutrophils are indeed the major source of cell-free DNA after stroke, we next aimed to deplete neutrophils using i.p. injection of Ly-6G-specific antibodies (clone 1A8) or by inhibition of NETosis using administration of a PAD4 inhibitor (GSK484). Here we found that *in vivo* depletion as well as inhibition of NETosis is sufficient to decrease cfDNA levels after experimental stroke (A). Moreover, both therapeutic approaches, neutrophil depletion and inhibition of NETosis, were equally effective in preventing post-stroke inflammasome activation in circulating monocytes after experimental stroke (B), confirming the key role of NETosis in DNA-mediated inflammasome activation. These new results have been included in the revised Figure 3f, g.

Finally, we performed NETosis inhibition in our novel plaque rupture model and could also confirm here that cfDNA (A) and caspase-1 activation in the carotid artery plaque (B) are decreased once neutrophil extracellular trap formation is functionally inhibited. These results are presented in the revised Figure 3h-j.

Taken together, these additional experiments which were prompted by the reviewer’s comment substantially expanded our description and biological understanding of cfDNA-mediated inflammasome activation. Particularly, we were able to identify neutrophil degranulation as the main and unexpected source of the rapid increase in cfDNA after stroke. These new findings are conceptually

relevant as they provide explanation for the cellular uptake mechanisms of cfDNA in macrophages and propose neutrophil NETosis as a potential immunotherapeutic target after stroke.

5) Because the cfDNA methods used to evaluate differences between asymptomatic and stroke patients are unspecific, they can be biased by the pre-analytical conditions or experimental conditions used. Based on the method section in the supplementary documents it is unclear if such conditions could alter the observations made in Figure 2G and Figure 4. The authors should clarify this.

Response: For all cell-free DNA acquisition (human and murine) only plasma was used.

Murine sample procedure: Venous blood from cardiac puncture was drawn in EDTA tubes. Samples were then centrifuged twice, plasma was collected and frozen at -80°C until further processing. For cfDNA isolation, we used the plasma/serum cell-free circulating DNA purification mini kit (#55100, NORGEN Biotek, Canada), according to the manufacturer's instructions. Per mouse, 250µl plasma was diluted 1:1 with nuclease-free water and used for cfDNA isolation. The isolated cfDNA was dissolved in 30µl elution buffer and total, ssDNA and dsDNA were acquired. Total cfDNA and ssDNA was acquired using spectrophotometry (nanodrop ND1000). For dsDNA we used the commercially available "Qubit dsDNA-HS (high sensitivity) Assay" (#Q32851, Thermo Fisher, US) to determine concentrations by fluorophotometric analysis.

Human sample procedure: Blood collection was performed at the Medizinische Hochschule Hannover, Germany. Here, venous blood was collected and sampled in EDTA tubes. EDTA full blood was then transferred to the Institute for Stroke and Dementia Research (ISD), Munich, Germany. Samples were centrifuged for 10min at 1500 x g and plasma was collected and stored at -80°C until further processing. For cfDNA isolation, we used the same kit as for murine samples (#55100, NORGEN Biotek, Canada), according to the manufacturer's instructions. 500µl plasma was used for cfDNA isolation. The isolated cfDNA was dissolved in 30µl elution buffer and total, ssDNA and dsDNA were measured as described above for the murine samples.

6) Most of the mechanistic work is demonstrated (logically) on animal models. This is a limitation of the work conclusions, which is not clearly stated in the text. The author should also discuss how a translation into clinic from their observation could be achieved (and if this is possible).

Response: We report in this study novel results of a large epidemiological analyses from 1798 patients from two ongoing clinical cohorts on the incidence of early recurrent vascular events after stroke – the so far largest clinical cohort addressing this question in the past two decades (Figure 1a and Extended data Figure 1). We then indeed go on to study the inflammatory mechanisms contributing to this high risk of early recurrent events – for which we generated a novel animal model, performed numerous highly complex in vivo experiments, used novel genetic animal models and multiple therapeutic approaches. Finally, we use prospectively collected fresh CCA atherosclerotic sample material collected over a period of more than 2 years from two academic centers to validate our novel mechanistic information from the experimental stroke model in human samples. Of note, this is the first analysis of inflammasome activation and downstream inflammatory mechanisms in human CCA plaque samples so far. With this human material we were able to confirm all key mechanistic findings, previously unknown and unrecognized to contribute to post-stroke recurrent events, from animal models in the human samples. Vice versa, in new experiments

performed during this revision and presented in detail above, we characterized the cellular cfDNA source in human sample material and again performed a reverse translational approach to investigate the role of NETosis in detail in an experimental model. We believe and are confident that what we describe in this study is a highly efficient example of bi-directional translation from clinical epidemiological data to experimental mechanistic exploration and back to clinical validation. Therefore, we have to politely disagree with the reviewer on the notion that our detailed mechanistic studies would represent a “limitation” of this study. Quite contrarily, our mechanistic studies have discovered a key pathomechanistic pathway (i.e. inflammasome-dependent plaque destabilization) which is likely to lead to novel therapeutic developments for secondary prevention in stroke patients.

Of note, a recent study from our author group (Grosse et al., *Stroke*, 2022) has independently described in a cohort study unfavorable outcome of stroke patients with increased cfDNA levels in the acute phase, which confirms the potential disease relevance of the pathway described herein. Moreover, based on the results reported in this study, we have initiated a randomized, controlled clinical trial (ReScinD Trial, ClinicalTrials.gov ID: NCT05880524) in which we aim to test the efficacy of DNase I in a proof-of-concept approach on systemic inflammation and clinical outcome after stroke. This study was approved by the national and local authorities in May 2023 and has currently started recruiting patients. However, the scope of such a prospective clinical trial is of course vastly beyond the scope of this already very comprehensive mechanistic study and is anticipated to recruit participants over a period of 3 years involving 3 clinical centers.

Following the reviewer’s suggestion, we added a paragraph in the discussion section in which we discuss the potential translational relevance of the discovered inflammatory mechanism of plaque rupture and now also reference our newly recruiting clinical trial as a translational example of how to further develop findings from this study for clinical development (discussion section, page 15).

7.) Because alterations in the nuclease activity leads to specific marks in the cell-free DNA structure, could this be used as an early marker for stroke detection (cf <https://doi.org/10.1016/j.tig.2021.04.005>)? In general, the generation of cfDNA involve a group of nucleases, and the potential use of a group of nucleases could further increase the treatment efficiency. This could maybe be further discussed or investigated.

Response: We thank the reviewer for this important comment which we did not sufficiently consider before. In our animal model the use of DNase I at the indicated dose and given as a single bolus was sufficient to decrease cfDNA concentration and prevent the inflammatory response to levels nearly reaching baseline levels in sham-operated animals. Therefore, we cannot exclude synergistic effects of combination with other nucleases or somewhat higher potency of other nucleases which we did not test systematically in this study. However, putative additive effects of other nucleases are probably only minor because the DNase I efficacy already nearly ceiling with a nearly complete block of the inflammasome-dependent effects on systemic inflammation and atherosclerotic plaque destabilization. As suggested by the reviewer, this issue is now discussed in the revised discussion section (page 15).

Referee #3:**Specific Comments**

1) Some added controls may be required. Experiments compare CCA-TS only vs CCA-TS plus stroke mice. What about stroke only mice? Are there thrombi that also occur days or weeks later? Where? What happens to inflammasomes, caspases, MMPs, circulating free cfDNAs etc? What happens to the carotids in mice subjected to stroke-inducing surgery alone?

Response: In this study we investigated the effect of stroke on destabilization of common carotid artery (CCA) plaques, particularly the mechanisms of post-stroke inflammation on plaque rupture and secondary cerebrovascular events. In order to study the causal role of a surgically induced stroke on CCA plaques we needed to develop an animal model of rupture prone plaques which do not develop spontaneously in mice, not even in the commonly used atherosclerosis mouse models (ApoE^{-/-} or LdlR^{-/-} mice) which develop atherosclerotic plaques predominantly in the aorta but not in the CCA (A). Moreover, aortic plaques in common atherosclerosis models are also not prone to rupture and therefore do not lead to spontaneous vascular events unlike high-risk human plaques which we aimed to mimic in our novel animal model of rupture-prone CCA plaques. For this model, we induced a tandem stenosis (TS) of the CCA leading to blood flow turbulences which induced concentric plaque growth with features of high risk, rupture-prone plaques (see Extended data Figure 2 for detailed characterization).

Please excuse this lengthy explanation, which is meant to justify why we did not include a “Stroke only” group in our original manuscript, as such mice without a tandem stenosis do not display CCA plaques and from our >15 years of experience in experimental stroke models (with several thousand animals analyzed) do never show secondary brain infarctions.

However, we of course fully acknowledge and understand this concern and the need for a formal control group. Therefore, as suggested by the reviewer, we included a stroke only group in the revised manuscript highlighting the lack of CCA plaque formation (A) and absence of secondary brain infarction (B). This additional data has been added to the revised manuscript in Extended data Figure 2j and we hope it further clarifies the rationale for the used CCA-TS model in this study.

Moreover, we compared key inflammatory readouts, as suggested by the reviewer, between CCA-TS+stroke and stroke only groups and analyzed caspase-1 cleavage as a marker of inflammasome activation and the expression of MMP2/9 (C-E). However, we feel that this analysis does not add important biological information to the manuscript as it basically only validates again that without tandem stenosis induction in the CCA atherosclerotic plaque formation and vascular inflammation are basically absent. We therefore present this new data only for the reviewer’s information but have so far not included it in the revised manuscript. If the reviewer deems it important, however, we are of course happy to include it in the Supplementary information.

Finally, we refrained from presenting data of further in-depth analysis between the remaining control groups, which is the comparison between naïve/sham mice to stroke only induction. This comparison would only reflect the effect of stroke on systemic inflammation (in the absence of atherosclerosis), which we and others have previously characterized in detail. Specifically, we have previously described systemic AIM2 inflammasome activation in response to stroke (Roth et al., Immunity, 2021), which was in fact the basis for this current study and is referenced in the introduction section.

2) Seems like the secondary strokes occur in the ACA territory? Is this surprising since the major flow routes should come into MCA territories? What are the percentage distributions of secondary stroke locations? The authors show secondary thrombi in CCA-TS plus stroke mice. Should one see these in the more-upstream ACA territories instead of the large vessels in the base of the brain? Importantly, these data should be quantified in terms of amount, location of thrombi in various vascular regions and how they might correspond to location of severity of the secondary strokes. It may also be helpful to compare these numbers between all three experimental groups (CCA-TS, stroke, CCA-TS plus stroke).

Response: We thank the reviewer for this excellent remark and believe that the perception of secondary strokes occurring primarily in the ACA territory was biased by our misleading demonstration of an ACA territorial secondary lesion which was not representative for most recurrent infarctions and without providing a quantification of such. Therefore, we have performed as per the reviewer's suggestion a quantification of the distribution of secondary lesions in the cerebral vascular territories (see below). As expected, the majority of lesions was found in the middle cerebral artery (MCA) territory, with only 3 lesions in the anterior cerebral artery (ACA) and only one in the watershed area between middle and posterior cerebral artery territories.

However, a corresponding systematic analysis of thrombi localization in the supplying arteries was not possible due to technical limitations of the serial sectioning approach, which does not cover the complete vascular arborization of the large intracranial arteries to allow such an analysis. Moreover, arterio-arterial emboli leading to secondary brain injuries are even expected to be already spontaneously lysed or resorbed by the time of histological analysis. For analysis of secondary infarction incidence, we studied brains at 7d after the primary lesion. However, analyses of the inflammatory response and secondary plaque rupture indicate that a substantial number of secondary lesions due to plaque rupture occur already in the acute phase within 24h (on this issue, please see also the detailed reply to comment 4 below) – hence, it is not surprising that at 7d after the primary

lesion we observe only the resulting secondary lesion but not anymore the causative intravascular embolism.

However, to further address this aspect in vivo and avoid the limitations of the histological analysis as described above, we made use of a recently established in vivo imaging modality of intravascular thrombi using MPIO-based matrix microparticles (Lizzarondo et al., Science Advances, 2022). We performed in vivo MRI imaging of mice undergoing our CCA TS model and acquired TOF sequences to visualize vascular structures (MR-angiography of the cerebral vasculature) and T2*-weighted sequence for iron-sensitive imaging (i.e. MPIO-bearing intravascular thrombi) before and after i.v. DNA injection. A detailed description of the methodology is provided in the revised methods section. With this imaging modality we were able to confirm intra-stenotic atherothrombosis in the CCA resulting in vascular occlusion in the angiography (red area is indicating hypoperfused right MCA territory); additionally, we detected more distal thrombotic material in MCA branches (arrowheads in right images). This new data set has been included in the revised Figure 4i.

3) The authors try to make a distinction between NLRP3 vs AIM2, stating that only inhibition of AIM2 but not NLRP3 prevented inflammasome activation. However, in the western blot shown in Fig 2E, it seems that inhibition of NLRP3 also reduces caspase-1 activation.

Response: We fully concur with the reviewer's observation. Indeed, in the referenced exemplary western blot gel it seems that NLRP3 inhibition also decreases inflammasome activation. However, in the quantification of the western blot gels for the Caspase-1 p20 / Actin ratio from 6 replicates (Fig. 2f of original manuscript) we did not observe a statistically significant reduction ($p=0.83$). However, to more accurately distinguish potential effects on inflammasome activation (independent of priming, i.e.

pro-Caspase-1 expression), we modified the analysis according to another reviewer's suggestion by quantifying the p20/pro-caspase 1 ratio (see modified Fig. 2g in revised manuscript). With this analysis, indeed, also NLRP3 inhibition by MCC950 significantly reduced inflammasome activation. We corrected the interpretation of these results in the revised manuscript's results section (page 6).

Since these results were obtained using a (high dose) *in vivo* NLRP3 inhibitor MCC950, we aimed to further clarify the role of individual inflammasome sensors using a more standardized and specific *in vitro* system. For this, we characterized the stimulation of bone marrow-derived macrophages with stroke or sham serum for different inflammasome sensor deficiencies using NLRP1, NLRP3, cGAS, and AIM2 knockouts and in inflammasome-deficient ASC knockout macrophages. We acquired caspase-1 cleavage in the macrophage lysates. Furthermore, we checked NET DNA stimulation in WT vs AIM2-deficient macrophages. We observed that only ASC and AIM2 deficiency prevented inflammasome activation (i.e. increase in p20/pro-caspase-1 ratio) after stroke serum treatment (A). Correspondingly, caspase-1 cleavage was completely blunted in AIM2^{-/-} BMDMs in response to DNA treatment (B). These new findings are now presented in the revised manuscript in Extended data Figure 8C,D.

In summary, based on our *in vivo* pharmacological NLRP3 inhibition, we cannot exclude some additional/additive effects of the NLRP3 inflammasome on post-stroke plaque inflammation. Yet, AIM2 seems to be the predominant sensor for serum-derived inflammatory cues after stroke, which we identified to be cell-free DNA which is the specific ligand for AIM2. Correspondingly, AIM2 inhibition and neutralization of cfDNA by DNase I treatment was sufficient to largely prevent the systemic inflammatory response to stroke and inflammatory plaque rupture.

4) There may be important questions about timing. cfDNA is very rapidly increased but this is transient, and levels essentially returns back normal by 24 h post-stroke. In contrast, the recurrent events were detected at 7 days post-stroke. How does a very early transient signal cause such a delayed response?

Response: We fully agree with the reviewer's remark and his observation of a very rapid spike in cfDNA concentration is absolutely correct. Moreover, also the inflammatory response to the acute cfDNA release is occurring already within the first 24h as we have previously characterized in detail (Roth et al., *Immunity*, 2021) and confirmed also on tissue-level at the CCA in this study. Of note, all data in Figure 1g-i on exacerbation of plaque inflammation after stroke is already at 24h after stroke. Hence, we agree with the reviewer and expect plaque rupture and secondary infarctions to occur predominantly in the acute phase already within the first 24-48h. In fact, this corresponds also to the risk profile of recurrent events in humans with the highest recurrence rate within the first 72h (e.g. Mono et al., *J Neurosurg*, 2013).

The rationale to study recurrent event rate in our animal model at 7d after stroke or sham surgery was twofold:

First, we aimed to maximize the sensitivity to detect early recurrent events, which was based on several parameters (MRI detection, cell death (FJC & TUNEL) and microglial reactivity). As infarct maturation and histological detection of cell death and particularly microglial reactivity requires in our experience at least 12h to allow for sufficiently sensitive detection of also small parenchymal lesions on consecutive serial sections, we were concerned to “miss out” on some lesions which might have occurred just very recently before brain sampling when analyzing too early. In contrary, experience with also very small brain lesions (e.g. stab wound injury, photothrombotic single arteriole occlusion) has shown that such microlesions do not “resolve” or are being absorbed within 7d and remain still detectable by the used parameters in this study.

Second, in the clinical stroke literature, secondary brain ischemia in the first week after the incident lesion is most commonly termed as “early recurrent events” which were the focus of this study. Moreover, clinical trials investigating early recurrent event rates often use MRI at 7d after incident lesion as the primary endpoint (e.g. Kang et al., Annals of Neurology, 2003). Hence, we aimed in our study design to mimic this clinical trial situation and detect early recurrent events in contrast to later recurrent events in the chronic phase which most likely have a different etiology.

Nevertheless, to specifically address this reviewer question, we analyzed the recurrent event rate in a cohort of 10 animals with tandem stenosis at the CCA already at 24h after stroke. At this early time point we already detected secondary lesions fulfilling the same histological criteria as at 7d for 3 out of 10 animals (see below). This additional data set has been added in the revised manuscript in Supplementary Figure 3H.

5) The authors try to prove cfDNA causality by injecting this into mice. More details are needed. How much, exactly how, and when are these injections performed? Importantly, how can one equate the in vivo DNA challenge treatment (50 ug) to the endogenous levels of stroke-induced upregulation of cfDNA (600-800 ng/ml)?

Response: For the referenced in vivo experiment (revised Figure 2j), we isolated NET DNA from cultured murine neutrophils. For this, we isolated neutrophils from WT bone marrow (humerus, femur and tibia) and 2×10^7 cells were cultured in 150 mm tissue culture dishes. Cells were then stimulated with 100 nM PMA overnight. The next day, the supernatant was collected from the culture dish and centrifuged at 500 xg for 10 min, then the supernatant was collected and centrifuged again at 15,000 xg for 10 min. After that, the supernatant was decanted and the pellet (i.e., the NET DNA) was dissolved in nuclease-free water.

We then injected 5 μg of the isolated NET DNA per mouse (not 50 μg , as stated by the reviewer). This dose was established in previous experiments and verified to result in comparable blood cfDNA concentration to the stroke-induced endogenous levels. For this, we measured blood concentrations of cfDNA after i.v. NET DNA injection; the resulting blood concentrations after i.v. injection of 5 μg NET DNA was at the very same levels as observed in the acute phase after experimental stroke, which in both cases is approx. 900 ng/ml (compared Figure 2h and Extended data Figure 8a).

Moreover, additional lines of evidence further confirmed the causal role of cfDNA in mediating the inflammatory response to stroke:

- the i.v. DNA challenge was sufficient to lead to a manifold increase in plaque IL-1 β concentration, which is the key cytokine leading to MMP activation and plaque degradation (Extended data Figure 8)
- the i.v. DNA challenge increased MMP activity and plaque coagulation (F.XIIa expression) at the CCA plaque (Figure 4h and Extended data Figure 10)
- highly specific degradation of cell-free DNA by DNase I was sufficient to prevent post-stroke plaque inflammation, rupture and secondary ischemic events (Figures 2j, k and 3i, j)

6) In Fig 3, besides using serum, does DNA challenge also induce MMP activation? Moreover, does DNase eliminate stroke-induced upregulation of MMP and Factor XIIa?

Response: As suggested by the reviewer, we additionally analyzed the effect of NET DNA stimulation on BMDMs – correspondingly to the stroke serum stimulation demonstrated in the original Figure 3. We observed that 250 ng NET DNA stimulation of primed (LPS 100 ng/ml for 4h) WT BMDMs was sufficient to induce inflammasome activation and release of IL-1 β (**A, B**). Moreover, transfer of supernatant of such stimulated BMDMs onto naïve BMDMs was sufficient to induce MMP release without prior priming (**C**). In contrast, stimulation of primed AIM2^{-/-} BMDMs with NET DNA did not induce IL-1 β release (**B**) and its supernatant transfer to WT BMDMs failed to induce MMP release (**C**). These new results have been included in Extended data Figure 9h-j.

The effect of DNase treatment of MMP and F.XIIa upregulation was already shown in the original manuscript; these results are shown in Figure 4I of the revised manuscript.

7) Finally, there is a critical question regarding the proposed cfDNA mechanism. Many other factors are known to induce plaque instability, including cytokines. Many of these cytokines (e.g. IFN, IL6 etc) are known to be upregulated after stroke. How can the authors prove that cfDNA is the trigger and not these other cytokines?

Response: We propose a causal chain of inflammatory events which are triggered by the rapid increase in cfDNA after the brain injury, leading to AIM2 inflammasome activation, resulting in IL-1 β secretion (which requires inflammasome activity) and subsequent IL-1-mediated plaque degradation (via MMP activation) and expression of secondary cytokines such as IL-6 (see scheme below). Therefore, we do not propose that cfDNA itself has effector function to directly induce plaque degradation but its pro-inflammatory effects are mediated by exactly the effector cytokines mentioned by the reviewer (predominantly IL-1 β), which are downstream of cfDNA-mediated inflammasome activation.

This concept is based on numerous data sets provided already in the original manuscript as well as additional data from experiments performed during this revision:

- i) Experiments using in vivo NET-DNA challenge and therapeutic neutralization of cfDNA using DNase I have demonstrated a causal role of cfDNA in mediating post-stroke inflammasome activation (caspase-1 cleavage and IL-1 β secretion) – please see reply to comment 5 for more detailed explanation.
- ii) The new in vitro experiments described in the comment 6 above demonstrate that AIM2-inflammasome dependent IL-1 β secretion can mediate MMP activation in naïve macrophages
- iii) We have demonstrated a causal role of inflammasome activation in leading to atherosclerotic plaque rupture by several lines of evidence: genetic (caspase-1 deficiency) or pharmacological (caspase-1 inhibitor VX765) inflammasome deficiency nearly completely blunts the systemic inflammatory response to stroke (blood and CCA plaque cytokine expression) and prevents inflammatory plaque rupture and secondary brain ischemia after experimental stroke.
- iv) In additional experiments performed during this revision (reviewer 1, comment #4), we were able to confirm in vivo the effector function of IL-1 β on MMP activation and plaque rupture. More specifically, we have already demonstrated in the original manuscript the IL-1 β secretion in response to cfDNA-mediated inflammasome activation as well as the dose-dependent effect of IL-1 β on MMP expression by macrophages.

In new experiments, we neutralized circulating IL-1 β by the administration of neutralizing monoclonal antibodies and analysis of the treatment effects on CCA plaques 1 week later. IL-1 β neutralization was sufficient to reduce MMP2/9 activity in CCA plaques (in situ zymography, **A**) and prevented fibrous cap thinning (**B**). These new results were added to the revised Figure 4d, e.

In summary, we fully agree with the reviewer's assessment and provide further evidence that IL-1 β is a key effector enzyme in the proposed mechanisms of cfDNA-mediated inflammasome activation and inflammatory plaque rupture. However, results from our experiments unequivocally demonstrate that cfDNA-mediated inflammasome activation is upstream the post-stroke release of IL-1 β . This causal directionality is most clearly highlighted that cfDNA neutralization or inflammasome inhibition nearly completely prevents post-stroke systemic inflammation (i.e. cytokine secretion), yet, neutralization of IL-1 β is also highly effective in preventing post-stroke atherosclerotic plaque degradation.

Reviewer Reports on the First Revision:

Referees' comments:

Referee #1 (Remarks to the Author):

The revised manuscript is definitely improved and contains several pieces of interesting additional data. However, it still falls short of clearly demonstrating the main proposed mechanism involving uptake of circulating DNA by plaque macrophages leading to activation of AIM2 is responsible for increased contralateral stroke formation.

1. There is now more convincing data for plaque breakdown, however, it is not completely clear if they are dealing with plaque rupture of plaque erosion/Netosis/thrombosis as described by Peter Libby's group in a similar model (Franck G et al, Circ Res 2017). The evidence for inflammasome activation in the plaques is now compelling, but involves both AIM2 and NLRP3 inflammasomes. Would the very specific NLRP3 inflammasome inhibitor MCC950 improve contralateral stroke?
2. The new finding that increased circulating DNA is derived from neutrophils is interesting. However, the experiments showing that injected DNA, or DNA added to cells can activate the AIM2 inflammasome does not prove that is what is occurring in the carotid plaque in vivo to mediate contralateral stroke. The new Figs 2 i,j,k do not directly address this issue. The beneficial effects observed for DNase-1 injection or Pad4 inhibition could be explained by effects in the brain and may not be due to effects on carotid plaque rupture.
3. The authors have used an inhibitor to address the in vivo role of AIM2. However, the recently published manuscript shows that this inhibitor also inhibits other DNA mediated responses mediated by TLR9 and cGAS. Thus, it is not highly specific. Most importantly the authors have declined to conduct experiments using AIM2 KO bone marrow transplantation experiments to test their model and prove more directly the in vivo role of AIM2 in contralateral stroke. Their argument is that mice with Aim2 deficient bone marrow may also have less developed atherosclerotic lesions and therefore they would have to develop a complex inducible model of Aim2 deficiency. However, the published data (Paulin, O. Soehnlein et al Circ 2018) shows a mild impact of Aim2 deficiency in Apoe^{-/-} mice with no effect on lesion area. Even if there are modest changes in the plaque from Aim2 deficiency (this is in fact not known for their model), the authors could still compare the impact of sham operation or stroke on plaque rupture and contralateral stroke formation. If Aim2 deficiency removes the difference between these 2 groups, it would support the essential role of Aim2 in contralateral stroke.

Referee #4 (Remarks to the Author):

Most of the problems have been clarified. There are still critical problems that need to be clarified.

Specific comment 1:

This reviewer felt that the manuscript should also include the figures and experimental results shown in the reply (page 22) These results are interesting; however, it remains to be clarified where and how in the body neutrophil activation and netosis occur, and circulating cfDNA seemed to activate macrophages throughout the body, but why did they invade into the plaques? Or did NetDNA enter the plaque and activate the infiltrated macrophages in an AIM2-dependent manner? This is unnatural though.

At this point, the authors should be aware of two issues.

- 1) In Fig.3g, cfDNA increases by only 50 % compared to sham, but is this enough to activate whole body macrophages? Are these roughly equivalent to 5 ug/ml (as replied to comment 5)? Or was macrophages' AIM2 activated even with sham?
- 2) Was plaque rupture confirmed in all cases of stroke recurrence (Fig.4m, etc.)? When CBF decreases in CCA stenosis, blood clots are more likely to occur in cerebral blood vessels, and netosis and AIM2 activation within cerebral blood vessels may directly cause blood clots in local cerebral blood vessels, causing stroke recurrence (as shown in Fig.4i). Even in clinical cases, recurrence due to plaque rupture should not be so common.

Specific comment 8:

IL-6 and IFN can be examined by administering neutralizing antibodies. The significance of IL-1b is recognized, but don't inflammatory cytokines also contribute to plaque rupture? In the summarized figure (reply page 27), macrophages activated by AIM2 subsequently released inflammatory cytokines, but has this been confirmed? If the macrophage goes into pyroptosis, it may not be able to produce cytokines.

Referee #5 (Remarks to the Author):

The authors are to be congratulated on their thorough investigation and coherent manuscript describing the mechanistic details underlying the vascular and immune players involved in the relationship between stroke and early recurrent vascular events. The authors' findings that neutrophil-derived cfDNA is a triggering or exacerbating factor in atherosclerotic plaque rupture in the inflammatory post-stroke environment and that it may serve as a druggable target are of considerable interest. These results are novel and relevant to a broad audience.

The comments review the technical details of the study, specifically those highlighted in the points of Reviewer #2 relating to cfDNA.

Point #1: In order to make claims about the involvement of "cfDNA" in a biological process, it is necessary to include size distribution evidence to support the idea that isolated nucleic acid fragments truly represent "cell-free" components, as opposed to contaminating high molecular weight genomic DNA derived from cells lysed during the biofluid collection or processing steps. The typical size profile of cfDNA is ~170 bp or multiples representing dimers/oligomers. The authors' inclusion of microfluidic gel electrophoresis (Agilent Bioanalyzer) data in this revision addresses this point. Because the probability of cell lysis (and subsequent contamination of the cell-free portion of the plasma sample by genomic DNA) increases with the time interval between sample collection and processing, the authors should include the exact time interval and interim preservation conditions (RT or on ice) in their supplemental materials ("Preparation of plasma samples for free nucleotide quantification"). This is important because long time intervals between collection of blood in EDTA tubes and plasma processing can result in the cleavage and breakdown of contaminating genomic DNA to smaller bp sizes that are difficult to distinguish from true cfDNA fragments on subsequent size distribution analyses. This point is also important to consider when working with low-volume blood samples from rodent models, where isolating plasma without buffy coat contamination following centrifugation requires great care. Additionally, the authors are advised to ensure that their use of the terms "serum" and "plasma" correspond between the supplemental materials describing the methods (which describe the collection of "plasma" for the experimental mouse model and human stroke patients) and the manuscript itself (which describes increases in "serum" cfDNA). Finally, the addition of the EV-derived cfDNA characterization, to support the authors' claims that the major contributor to stroke-associated cfDNA increase is extravesicular DNA, is appreciated.

Point #2: While cfDNA has been previously described as a DAMP, cfDNA has primarily been investigated as a diagnostic marker in the context of stroke. The mechanistic nature of this study provides new evidence of the role cfDNA can play as an exacerbating factor in the broader pathological process leading to atherosclerotic plaque destabilization and rupture post-stroke. The results of the methylation-based cell-of-origin analysis completed by the authors in this revision are intriguing. The analysis adds depth to the revised manuscript by characterizing the composition of cfDNA in the post-stroke environment in patients and the experimental model, adding detail to the overall mechanism being described.

Point #3: The links between the mechanistic findings in the experimental animal model and their correlates in human pathological specimens have been adequately clarified in the revised results and

discussion sections of the text. The authors' reference to the ongoing clinical trial of DNase I in stroke patients is a valuable addition to the manuscript discussion section, highlighting one potential method for therapeutically targeting the described mechanism.

Point #4: As mentioned previously, a deeper analysis of cfDNA origin in stroke helps to clarify the mechanistic details described in the manuscript and the potential role of DNase treatment in the downstream prevention of secondary ischemic events. The exact percentages of total cfDNA from each cell-type should be listed in Figure 3a. It is important to clarify that the total neutrophil-derived cfDNA is still <30% of the total circulating cfDNA, weakening the argument that this is the main factor driving inflammasome activation. Indeed, other organs might contribute cfDNA after stroke, such as the liver, heart, and kidneys. This possibility was not investigated in this study, which focused narrowly on brain-immune-vascular interactions. Vascular endothelial cells also represent an important but unexplored potential contributor to the total circulating cfDNA milieu. As indicated in the authors' response, serum was the blood fraction used to treat primary cultured neutrophils to induce NETosis (Figure 3c), though plasma is the fraction used to identify and characterize cfDNA. An explanation of the choice of one blood fraction over another for each experimental step is relevant to include, as the clotting process involved in serum separation involves white blood cell lysis that may lead to the spillage of additional genomic DNA contents that are not ordinarily present in the extracellular compartment in vivo (see PMID 32516802). In general, the authors should clarify the anatomical location of the NETs thought to be contributing to cfDNA release, whether circulating or in the plaque itself.

Point #5: As discussed above, the use of the terms "serum" vs. "plasma" in the manuscript and supplemental materials should be clarified, and the exact time interval between blood draw and sample centrifugation should be included. This is particularly relevant for the human plasma samples, where it is stated: "EDTA full blood was then transferred to the Institute of Stroke and Dementia Research..."). How long did it take for the transfer to happen, and what were the storage conditions for the EDTA tubes during transit? Additionally, blood draws from peripheral veins in living animals would provide a more representative cfDNA profile compared to those drawn via cardiac puncture, at which point additional perfusion-related drugs may have been administered. Indeed, death-associated cfDNA-release (especially considering the real-time nature of cfDNA profiling) has the potential to alter the pre-death cfDNA profile. This is mitigated by technical difficulties associated with small animal venous blood draws (awake or under anesthesia) and the need for larger sample volumes for cfDNA analysis.

Point #6: The combination of evidence derived from epidemiological analyses, experimental animal models, and prospectively collected human pathological specimens that is used to validate the mechanism presented in this study is valuable. The translational potential of this work has been clarified in the revised manuscript.

Point #7: The authors' addition of a sentence addressing this point on pg. 15 of the revised manuscript discussion section is appreciated and satisfactory.

In summary, after careful review of the revised manuscript, extended data, and supporting materials, the authors appear to have adequately addressed the previous comments of reviewer #2

regarding cfDNA. However, to truly confirm that the cfDNA mechanism occurs in vivo, the authors should repeat their animal model plasma collection experiments using cfDNA collection best practices to ensure that the high levels of neutrophil-derived cfDNA described in this study accurately represent levels/percentages of circulating neutrophil-derived cfDNA in vivo. Given the collection strategies described in the methods, it is possible that unintentional enrichment of neutrophil-derived DNA via WBC lysis may have occurred due to long (>2 hr) time intervals between blood collection into non-cell stabilizing EDTA tubes (as opposed to cfDNA stabilization tubes) and centrifugation and/or due to the absence of a second high-speed (14,000-16,000 g) centrifugation spin of the plasma. Quantification of total cfDNA using qPCR or ddPCR is also recommended. References that advise on cfDNA collection best practices include PMID 32122922 and PMID 37806433.

Author Rebuttals to First Revision:

REVIEWER #1

1a) There is now more convincing data for plaque breakdown, however, it is not completely clear if they are dealing with plaque rupture or plaque erosion/Netosis/thrombosis as described by Peter Libby's group in a similar model (Franck G et al, Circ Res 2017).

Response: We performed during the revision a systematic histological analysis, as suggested by this reviewer, to assess fibrous cap rupture, defined as discontinuation of Picro Sirius Red-labeled fibrous caps. This was detected in all animals that also had secondary contralateral lesions and was highly significantly associated with atherothrombosis (see Figure 1d). Therefore, we are confident that this model recapitulates CCA plaque rupture. In contrast, the referenced study by the Libby group used a different model which was specifically developed to induce erosion- but not rupture-prone lesions. For this, the authors first induced an endothelial damage followed by healing, which led to a matrix-rich fibrous neointima—but without a lipid or necrotic core and only few inflammatory cells. After that, flow perturbations were induced using a cone placed on the arteries with the “tailored fibrous intimal hyperplasia” (Franck et al., results section). Thereby, this is a clearly different model: while our model is characterized by large necrotic core lesions, high inflammatory burden and a thinned, rupture-prone fibrous cap, all these features are missing in the described model in Franck et al.

1b) The evidence for inflammasome activation in the plaques is now compelling, but involves both AIM2 and NLRP3 inflammasomes. Would the very specific NLRP3 inflammasome inhibitor MCC950 improve contralateral stroke?

Response: We tested the effect of NLRP3 deficiency as well as pharmacological inhibition using MCC950 in several experiments.

The only data set supporting a role of the NLRP3 inflammasome in this model is the finding on reduced caspase-1 cleavage in CCA plaques after in vivo treatment with MCC950 (Figure 2f+g, and below). However, while this was a statistically significant effect it was still less efficient than the AIM2 inhibition. Importantly, MCC950 treatment was not able to significantly reduce local IL-1 β concentrations in the CCA plaque in contrast to AIM2 inhibition (Ext. data Fig. 7c, and below). Moreover, a genetic screening experiment using macrophages of specific

In conclusion, these results suggest a potential minor involvement of the NLRP3 inflammasome in vivo either as a bystander or secondary mechanisms being deployed in response to the primary inflammatory stimulus; however, the NLRP3 inflammasome is not directly engaged by factors released in the stroke serum (see in vitro experiments). Moreover, this secondary involvement of

the NLRP3 inflammasome (potentially downstream of DNA sensing) is well in line with previous reports demonstrating that DNA-sensing inflammasomes can induce a cell death program that initiates NLRP3 activation via K^+ efflux (Gaidt et al., Cell, 2017).

2) The new finding that increased circulating DNA is derived from neutrophils is interesting. However, the experiments showing that injected DNA, or DNA added to cells can activate the AIM2 inflammasome does not prove that is what is occurring in the carotid plaque in vivo to mediate contralateral stroke. The new Figs 2 i,j,k do not directly address this issue. The beneficial effects observed for DNase-1 injection or Pad4 inhibition could be explained by effects in the brain and may not be due to effects on carotid plaque rupture.

Response: We performed the NET-DNA injections as an in vivo challenge experiment to establish the causality of DNA inducing plaque inflammasome activation, in response to a request by this reviewer in the first revision round. Indeed, these findings shown now in Fig. 2i-k did not demonstrate that DNase could not have additional effects on the primary infarct itself, as the experiments were not designed to address this question. However, we did assess this possibility and performed in vivo brain MRI 48h after stroke for infarct volumetry in control treated mice and animals receiving either DNase or the Caspase-1 inhibitor VX765 but did not observe a difference in infarct volumes between groups (see below). This new data set has been included in Extended data Figure 10e. Therefore, a direct effect on the brain in the sense of a neuroprotective function of these interventions can be excluded, while in contrast the inflammasome inhibition (VX765) did significantly reduce plaque vulnerability (Fig. 2e).

3a) The authors have used an inhibitor to address the in vivo role of AIM2. However, the recently published manuscript shows that this inhibitor also inhibits other DNA mediated responses mediated by TLR9 and cGAS. Thus, it is not highly specific.

Response: The compound calix-6-arene is an inhibitor of DNA sensors by competitively binding to the DNA binding site, as we have additionally demonstrated in the revision experiments shown in Ext. data Fig. 7a,b. As such, it is correct that calix-6-arene can theoretically also bind and inhibit cGAS sensing of DNA; however, we have observed in additional experiments performed during this revision that cGAS deficiency did not affect inflammasome activation in response to stroke serum stimulation (Ext. data Fig. 8c, and below). Therefore, this leaves AIM2 as the sole functional target of calix-6-arene in the context of post-stroke inflammasome activation.

3b) Most importantly the authors have declined to conduct experiments using AIM2 KO bone marrow transplantation experiments to test their model and prove more directly the in vivo role of AIM2 in contralateral stroke.

Response: In fact, we did perform the requested bone marrow transplantation experiments to test the involvement of macrophage-specific AIM2 in post-stroke inflammasome activation. These results were presented in the revised Figure 2I and clearly demonstrated that macrophage-specific AIM2 deficiency substantially reduces inflammasome activation. However, while these experiments demonstrate once again the causal role of AIM2 in mediating post-stroke inflammasome activation, these experiments are not suitable to study plaque rupture and recurrent vascular events for obvious methodological limitations which are further discussed below.

3c) Their argument is that mice with Aim2 deficient bone marrow may also have less developed atherosclerotic lesions and therefore they would have to develop a complex inducible model of Aim2 deficiency. However, the published data (Paulin, O. Soehnlein et al Circ 2018) shows a mild impact of Aim2 deficiency in Apoe^{-/-} mice with no effect on lesion area. Even if there are modest changes in the plaque from Aim2 deficiency (this is in fact not known for their model), the authors could still compare the impact of sham operation or stroke on plaque rupture and controlateral stroke formation. If Aim2 deficiency removes the difference between these 2 groups, it would support the essential role of Aim2 in controlateral stroke.

Response: We fully acknowledge the reviewer's very knowledgeable remarks, but we have to respectfully disagree with this assessment and their report on the referenced data on AIM2 deficiency in the Paulin et al. study from the Söhnlein group. The reviewer mentions "mild impact" and "modest changes" in the plaques between WT and the constitutive AIM2-KO mice, but in fact the differences are quite substantial!

[REDACTED]

[REDACTED]

In order to avoid this unnecessary long experiment which would require at least 18 months to be conducted, we performed alternative experiments including the above-mentioned bone marrow transfer experiment, the in vitro genetic screen for specific inflammasome knockouts (see response to 3a.), the ASC oligomerization experiment in WT and AIM2^{-/-} mice (Fig. 2k) and further characterization of the pharmacological inhibitor of DNA sensors. All these independent lines of evidence confirm the causal role of AIM2 in plaque inflammation and rupture, while the additional experiment in an inducible AIM2^{-/-} mouse strain would not generate substantial new evidence that would justify this experiment.

REVIEWER #4 (REF 2 REPLACEMENT)

Most of the problems have been clarified. There are still critical problems that need to be clarified. Specific comment 1: This reviewer felt that the manuscript should also include the figures and experimental results shown in the reply (page 22)

Response: We are of course happy to include the presented results from the control experiments in mice without CAA plaques in a revised manuscript version (now added in Extended data figure 2j-l). The data is shown below again for reference.

These results are interesting; however, it remains to be clarified where and how in the body neutrophil activation and netosis occur, and circulating cfDNA seemed to activate macrophages throughout the body, but why did they invade into the plaques? Or did NetDNA enter the plaque and activate the infiltrated macrophages in an AIM2-dependent manner? This is unnatural though.

Response: Indeed, we propose that stroke leads to systemic inflammasome activation and consequently also systemic macrophage inflammation throughout the organism, as we and others have already demonstrated in previous studies (e.g. Roth et al., Immunity, 2021; Grosse et al., Stroke, 2022). Likewise, NETosis after stroke seems to be a systemic process which might be initiated locally in the injured brain vasculature but propagates to the systemic increase in cell-free DNA in the circulation.

Moreover, we also fully concur with the reviewer in that we do not expect that DNA would “invade” into the atherosclerotic plaques, as we also do not claim or show such a finding in our manuscript. In contrast, we observe that after stroke the cell count of plaque macrophages increases dramatically, which is mainly due to the recruitment of circulating monocyte/macrophages to the atherosclerotic plaques (Fig. 1g,h). Additionally, we demonstrate that macrophages can readily internalize circulating NET-DNA, which leads to inflammasome activation in these macrophages (Fig. 3e). Therefore, our results indicate that DNA uptake and inflammasome activation in macrophages is a systemic process, with the subsequent recruitment of the activated macrophages to the inflammatory site in atherosclerotic plaques which further exacerbates plaque inflammation.

At this point, the authors should be aware of two issues.

1) In Fig.3g, cfDNA increases by only 50 % compared to sham, but is this enough to activate whole body macrophages? Are these roughly equivalent to 5 ug/ml (as replied to comment 5)? Or was macrophages' AIM2 activated even with sham?

Response: In comparison to naïve animals, the maximum increase in cell-free DNA after stroke is more than 2-fold—please see for absolute values the kinetics for cfDNA concentrations after experimental stroke shown in Fig. 2h. This increase in cfDNA is clearly sufficient for systemic inflammasome activation as demonstrated by multiple data sets throughout the manuscript, for example highlighted by data obtained from experiments using stroke serum for macrophage activation (Ext. data Fig. 8c, d), or the reduction of cfDNA concentration by DNase to Sham levels (Ext. data Fig. 8e) and its efficacy in preventing inflammasome activation (Fig. 2m).

Indeed, also Sham surgery can lead to a minor increase of cfDNA concentration, which is the reason why we normalized blood cfDNA concentration and Caspase-1 activation in Fig. 3g to Sham levels instead of only reporting the treatment effects for the stroke group only. However, Sham surgery itself neither induced systemic inflammation and also did not lead to any recurrent vascular events (see Fig. 1c). Therefore, we believe that Sham surgery, as well as minor vascular events such as a TIA or minor stroke, do not reach the inflammatory threshold to induce the systemic inflammatory changes as observed after major stroke injuries.

To address the question of comparability in blood concentration of NET-DNA injections to endogenous levels, we performed during the previous revision round already additional experiments. For this, we measured blood concentrations of cfDNA after i.v. NET DNA injection; the resulting blood concentrations after i.v. injection of 5 µg NET DNA was at the very same levels as observed in the acute phase after experimental stroke, which in both cases is approx. 900 ng/ml (please compare Figure 2h and Extended data Figure 8a).

2) Was plaque rupture confirmed in all cases of stroke recurrence (Fig.4m, etc.)? When CBF decreases in CCA stenosis, blood clots are more likely to occur in cerebral blood vessels, and netosis and AIM2 activation within cerebral blood vessels may directly cause blood clots in local cerebral blood vessels, causing stroke recurrence (as shown in Fig.4i). Even in clinical cases, recurrence due to plaque rupture should not be so common.

Response: Plaque rupture (and local atherothrombosis) was confirmed in 11 out of 12 cases of mice with contralateral, secondary lesions (See Fig. 1d and below). Moreover, we were further able to demonstrate local atherothrombosis and secondary CCA occlusion using in vivo MRI (see Fig 4i). In contrast, Sham surgery did not induce secondary lesions in a single animal despite the same degree of CCA stenosis and plaque development in these animals, clearly demonstrating that the stroke-induced inflammatory cascade is necessary to induce rupture of CCA plaques.

In our clinical study, and in accordance with previous epidemiological reports, the overall recurrence rate in the first 30 days was above 6% (1-year risk at approx. 12%). While the risk for recurrent stroke events in patients with a macroangiopathic (LAA) stroke etiology is more than 2-fold higher than in other etiologies. Moreover, the etiology for the recurrent stroke in LAA patients has been shown to be predominantly of the same etiology as for the index stroke (i.e. LAA = plaque rupture and atherothrombosis; see e.g. Shin et al., Arch Neurol, 2005; Kolmos et al., JCBFM, 2021; Rucker et al., Stroke, 2020).

Specific comment 8: IL-6 and IFN can be examined by administering neutralizing antibodies. The significance of IL-1b is recognized, but don't inflammatory cytokines also contribute to plaque rupture? In the summarized figure (reply page 27), macrophages activated by AIM2 subsequently released inflammatory cytokines, but has this been confirmed? If the macrophage goes into pyroptosis, it may not be able to produce cytokines.

Response: IL-1 is an early response cytokine under inflammatory conditions which is upstream of the clinically relevant IL-1—IL-6—CRP axis. As such, IL-1 is recognized as the most “apical” cytokine inducing a pro-inflammatory response in multiple innate and adaptive immune cells (see e.g. Dinarello, Blood, 2011 for a comprehensive overview). The mechanism leading to this very early secretion of active IL-1 β after stroke is the inflammasome activation described herein, as well as in previous reports from our group and others investigating inflammasome activation under conditions of sterile tissue injury (e.g. Roth et al., Immunity, 2021; Toldo & Abbate, Nature Reviews Cardiology, 2018). Of course, other pro-inflammatory cytokines and chemokines are also intricately involved in the pathogenesis of atherosclerosis and atherothrombosis, a phenomenon which is meanwhile commonly referred to as the “cytokine/chemokine network” (see e.g. Yan et al., Atherosclerosis, 2021). However, numerous reports have demonstrated that in acute infections as well as acute tissue injuries including acute ischemic stroke, IL-1 family cytokines are key initiators of this complex network of pro-inflammatory cytokines/chemokines as it enhances Nf κ B signaling via the IL-1 receptor, leading to the transcriptional upregulation of further cytokines including IL-6 (see e.g. Yamamoto et al., Nature, 2004; Orjalo et al., PNAS, 2009).

More specifically, we have demonstrated in a previous report that stimulating macrophages with stroke serum—using the same in vitro protocol as used also in this study for Extended Data Fig. 8c—we observed not only an increase in IL-1 β ,

[REDACTED]

The comment that pyroptotic cell death could contribute to macrophage loss and subsequent reduction in cytokine production is absolute accurate. However, this process is delayed and does rather contribute to the functional immunosuppression observed in the subacute phase after stroke and other acute injuries, as previously demonstrated by us and others (e.g. Roth et al., *Immunity*, 2021; Wen et al., *Front Immunol*, 2022; Croker et al., *Curr Opin Immunol*, 2014). In contrast, in the acute phase after inflammasome activation, the formation of Gasdermin pores in the process of pyroptosis has been described to be a major contributor to the early and massive secretion of IL-1 and IL-18 after inflammasome activation because the pore formation and cellular blebbing leads to rapid export of active IL-1 and IL-18 without inducing immediate cell death in macrophages (see e.g. Broz, *Semin Immunol*, 2023 for review).

REVIEWER #5 (REF 2 REPLACEMENT)

The authors are to be congratulated on their thorough investigation and coherent manuscript describing the mechanistic details underlying the vascular and immune players involved in the relationship between stroke and early recurrent vascular events. The authors' findings that neutrophil-derived cfDNA is a triggering or exacerbating factor in atherosclerotic plaque rupture in the inflammatory post-stroke environment and that it may serve as a druggable target are of considerable interest. These results are novel and relevant to a broad audience. The comments review the technical details of the study, specifically those highlighted in the points of Reviewer #2 relating to cfDNA.

1a) In order to make claims about the involvement of “cfDNA” in a biological process, it is necessary to include size distribution evidence to support the idea that isolated nucleic acid fragments truly represent “cell-free” components, as opposed to contaminating high molecular weight genomic DNA derived from cells lysed during the biofluid collection or processing steps. The typical size profile of cfDNA is ~170 bp or multiples representing dimers/oligomers. The authors' inclusion of microfluidic gel electrophoresis (Agilent Bioanalyzer) data in this revision addresses this point.

Response: We thank the reviewer for acknowledging that this issue has been resolved by the provided data using microfluidic gel electrophoresis.

1b) Because the probability of cell lysis (and subsequent contamination of the cell-free portion of the plasma sample by genomic DNA) increases with the time interval between sample collection and processing, the authors should include the exact time interval and interim preservation conditions (RT or on ice) in their supplemental materials (“Preparation of plasma samples for free nucleotide quantification”). This is important because long time intervals between collection of blood in EDTA tubes and plasma processing can result in the cleavage and breakdown of contaminating genomic DNA to smaller bp sizes that are difficult to distinguish from true cfDNA fragments on subsequent size distribution analyses. This point is also important to consider when working with low-volume blood samples from rodent models, where isolating plasma without buffy coat contamination following centrifugation requires great care.

Response: We fully agree with the reviewer that varying time intervals for cfDNA isolation can affect cell lysis and thereby accuracy of detected cell-free DNA. As requested, we now provide a detailed overview for time intervals and details of plasma and cfDNA isolation protocols utilized for the different sets of murine and human cfDNA isolation in this study (please see below). We further added the mean and range of quantified cfDNA within the different sample groups under “results” to the table below. This overview demonstrates that all samples from mouse experiments and human samples for methylome analysis have been processed within very short time frames of less than 30 minutes. Prolonged time periods of sample processing in the substudy of patients in which carotid artery samples and blood were analyzed in parallel (Fig. 5), was inevitable due to intraoperative sampling and shipping. This detailed information is now included for the methodological information as a table in the Supplementary material and methods section. The results for cfDNA concentrations between all indicated substudies has been included as a bar graph in Supplementary Information Figure 11r.

	Protocol according to Greytak et al. 2020	Stroke (mouse)	MI (mouse)	i.v.DNA (mouse)	Stroke (human; CCA samples)	Stroke (human; cfDNA methylation)	MI (human)
Time in EDTA (min)	≤120	5-15	15-30	5-15	≥180	15-30	≤15
1st centrifugation	800-1,600xg for 20min	3,000xg for 10min at 4°C	3,000xg for 10min at 4°C	3,000xg for 10min at 4°C	3,000xg for 15min	1,500xg for 10min at 4°C	1,600xg for 30 min
2nd centrifugation	14,000xg – 16,000xg for 10-20min at 4°C	3,000xg for 10min at 4°C	3,000xg for 10min at 4°C	3,000xg for 10min at 4°C	x	3000xg for 10min at 4°C	x
Storage	-80°C	-80°C	-80°C	-80°C	-80°C	-80°C	-80°C
Processing	Cat. 55100; NORGEN Biotek	Cat. 55100; NORGEN Biotek	Cat. 55100; NORGEN Biotek	Cat. 55100; NORGEN Biotek	Cat. 55100; NORGEN Biotek	QIAamp Blood DNA kit; Symphony robot	Cat. 55100; NORGEN Biotek
Results	X	HC 200(150-330) ng/ml AIS 486(335-745) ng/ml	HC 220(170-280) ng/ml MI 350(300-400) ng/ml	HC 350(230-480) ng/ml i.V. 800(650-1050) ng/ml	Asymp 450(350-520) ng/ml Syp 1200(500-2200) ng/ml	x	HC 900(600-1200) ng/ml MI 1700(1000-2800) ng/ml

Overview of human and murine cfDNA samples processing and acquisition

1c) Additionally, the authors are advised to ensure that their use of the terms “serum” and “plasma” correspond between the supplemental materials describing the methods (which describe the collection of “plasma” for the experimental mouse model and human stroke patients) and the manuscript itself (which describes increases in “serum” cfDNA).

Response: In response to the request for clarification, we have reviewed and revised the usage of the terms "serum" and "plasma" throughout the entire manuscript. Specifically, we utilize blood plasma for the quantification of cell-free DNA (cfDNA) and various inflammatory mediators. In contrast, blood serum, prepared by allowing blood to coagulate for 15 minutes at room temperature, is employed to stimulate neutrophils and bone marrow-derived macrophages. Our initial experiments involved the use of serum for macrophage stimulation to explore the role of cfDNA in mediating subacute immunosuppression, as detailed in Roth et al. (2021). The primary rationale behind this approach was to eliminate any potential cellular sources, such as thrombocytes, that could influence the stimulation of the inflammasome in cells. We have implemented these clarifications in the Methods section and throughout the manuscript, with all modifications distinctly highlighted for ease of identification. This adjustment ensures a precise distinction between the uses of serum and plasma in our study, aligning with the descriptions provided in the supplemental materials for both the experimental mouse model and human stroke patients.

1d) Finally, the addition of the EV-derived cfDNA characterization, to support the authors' claims that the major contributor to stroke-associated cfDNA increase is extravesicular DNA, is appreciated.

Response: We thank the reviewer for acknowledging the resolution of this issue through the addition of extracellular vesicle (EV)-derived cfDNA characterization.

2) While cfDNA has been previously described as a DAMP, cfDNA has primarily been investigated as a diagnostic marker in the context of stroke. The mechanistic nature of this study provides new evidence of the role cfDNA can play as an exacerbating factor in the broader pathological process leading to atherosclerotic plaque destabilization and rupture post-stroke. The results of the methylation-based cell-of-origin analysis completed by the authors in this revision are intriguing. The analysis adds depth to the revised manuscript by characterizing the composition of cfDNA in the post-stroke environment in patients and the experimental model, adding detail to the overall mechanism being described.

Response: We appreciate the reviewer's acknowledgment of how this analysis enriches the revised manuscript.

3) The links between the mechanistic findings in the experimental animal model and their correlates in human pathological specimens have been adequately clarified in the revised results and discussion sections of the text. The authors' reference to the ongoing clinical trial of DNase I in stroke patients is a valuable addition to the manuscript discussion section, highlighting one potential method for therapeutically targeting the described mechanism.

Response: We are grateful for the reviewer's recognition of the revised discussion section.

4) As mentioned previously, a deeper analysis of cfDNA origin in stroke helps to clarify the mechanistic details described in the manuscript and the potential role of DNase treatment in the downstream prevention of secondary ischemic events. The exact percentages of total cfDNA from each cell-type should be listed in Figure 3a. It is important to clarify that the total neutrophil-derived cfDNA is still <30% of the total circulating cfDNA, weakening the argument that this is the main factor driving inflammasome activation. Indeed, other organs might contribute cfDNA after stroke, such as the liver, heart, and kidneys. This possibility was not investigated in this study, which focused narrowly on brain-immune-vascular interactions. Vascular endothelial cells also represent an important but unexplored potential contributor to the total circulating cfDNA milieu.

Response: Following the reviewer's recommendation, we have incorporated a table detailing the exact percentages (mean and range) for each cell population into the revised Supplementary Information 1.

Indeed, in this analysis we primarily focused on the brain's role, as the organ acutely impacted by stroke, and the involvement of specific immune cell subsets in mediating the link between the brain and atherosclerosis post-stroke. Contrary to our initial hypothesis that cfDNA would predominantly originate from necrotic brain tissue, the new data compellingly challenge this assumption. Consequently, our analysis has not yet extended to additional organs and cell types, but such an exploration remains a promising direction for future research. We concur with the reviewer on the potential significance of endothelial and other remote organ contributions to the systemic cfDNA pool post-stroke, a topic we have now addressed and discussed in the revised discussion section on page 14. Despite the complex origins of total cfDNA as suggested by our human methylome analysis, we maintain that neutrophil-derived DNA plays a crucial role in driving post-stroke systemic inflammation. This assertion is supported by evidence from neutrophil depletion studies, pharmacological inhibition of NETosis, and in vitro NET-DNA stimulation, reinforcing the significance of neutrophil-derived DNA in this context.

	Mean of % cfDNA	Range of % cfDNA
Astrocytes	0.13	(0 - 0.27)
Neurons	0.04	(0 - 0.13)
Oligodendrocytes	0.02	(0 - 0.11)
Monocytes	8.36	(3.05 - 12.18)
Neutrophils	28.94	(12.29 - 56.62)
NK cells	1.19	(0.32 - 2.95)
B cells	0.81	(0.12 - 1.79)
CD4 T cells	3.34	(1.41 - 6.31)
CD8 T cells	0.40	(0.11 - 0.51)
Treg cells	0.13	(0 - 0.28)

As indicated in the authors' response, serum was the blood fraction used to treat primary cultured neutrophils to induce NETosis (Figure 3c), though plasma is the fraction used to identify and characterize cfDNA. An explanation of the choice of one blood fraction over another for each experimental step is relevant to include, as the clotting process involved in serum separation involves white blood cell lysis that may lead to the spillage of additional genomic DNA contents that are not ordinarily present in the extracellular compartment in vivo (see PMID 32516802). In general, the authors should clarify the anatomical location of the NETs thought to be contributing to cfDNA release, whether circulating or in the plaque itself.

Results: please see response to Figure 1b) above.

5) As discussed above, the use of the terms "serum" vs. "plasma" in the manuscript and supplemental materials should be clarified, and the exact time interval between blood draw and sample centrifugation should be included. This is particularly relevant for the human plasma samples, where it is stated: "EDTA full blood was then transferred to the Institute of Stroke and Dementia Research..."). How long did it take for the transfer to happen, and what were the storage conditions for the EDTA tubes during transit? Additionally, blood draws from peripheral veins in living animals would provide a more representative cfDNA profile compared to those drawn via cardiac puncture, at which point additional perfusion-related drugs may have been administered. Indeed, death-associated cfDNA-release (especially considering the real-time nature of cfDNA profiling) has the potential to alter the pre-death cfDNA profile. This is mitigated by technical difficulties associated with small animal venous blood draws (awake or under anesthesia) and the need for larger sample volumes for cfDNA analysis.

Response: please see our responses to comments 1b) and 1c) above on clarification of the terms “serum” and “plasma” and detailed information on cfDNA isolation protocols.

Additionally, we want to clarify that blood withdrawal in mice was not done post-mortem but in deeply sedated but alive mice through puncture of the right cardiac ventricle (of the beating heart). This procedure avoids post-mortem blood collection and the obvious caveats indicated by the reviewer, it also maximizes collectable blood volume in comparison to venous blood withdrawal and allows to maintain sterility of the samples.

6) The combination of evidence derived from epidemiological analyses, experimental animal models, and prospectively collected human pathological specimens that is used to validate the mechanism presented in this study is valuable. The translational potential of this work has been clarified in the revised manuscript.

7) The authors' addition of a sentence addressing this point on pg. 15 of the revised manuscript discussion section is appreciated and satisfactory.

Response to 6) and 7): we thank the reviewer for these positive assessments.

In summary, after careful review of the revised manuscript, extended data, and supporting materials, the authors appear to have adequately addressed the previous comments of reviewer #2 regarding cfDNA. However, to truly confirm that the cfDNA mechanism occurs in vivo, the authors should repeat their animal model plasma collection experiments using cfDNA collection best practices to ensure that the high levels of neutrophil-derived cfDNA described in this study accurately represent levels/percentages of circulating neutrophil-derived cfDNA in vivo. Given the collection strategies described in the methods, it is possible that unintentional enrichment of neutrophil-derived DNA via WBC lysis may have occurred due to long (>2 hr) time intervals between blood collection into non-cell stabilizing EDTA tubes (as opposed to cfDNA stabilization tubes) and centrifugation and/or due to the absence of a second high-speed (14,000-16,000 g) centrifugation spin of the plasma. Quantification of total cfDNA using qPCR or ddPCR is also recommended. References that advise on cfDNA collection best practices include PMID 32122922 and PMID 37806433.

Response: As recommended by the reviewer, we repeated plasma collection after experimental stroke and performed a parallel cfDNA isolation using either standard protocols published Greytak et al., 2020; i.e. PMID 32122922) or the slightly modified protocol used in this study – of note, the only notable difference between protocols was in regard to centrifugation speed of the second centrifugation step. Specifically, we collected fresh plasma samples from WT mice 4h after stroke (fMCAo) surgery. Blood was withdrawn intracardially from the right ventricle and was immediately placed into a 1,5 ml

EDTA tube and spined down first time 3,000 xg for 15min, then plasma (Supernatant) was collected and transferred for a second centrifugation step. 2nd Centrifugation was either performed as before in the study (**Protocol 1**) 3,000xg for 15min or with a second high-speed centrifugation step (**Protocol 2**) at 15,000xg for 15min (Greytak et al. 2020). cfDNA was then isolated using the above mentioned “Plasma/Serum Cell-Free Circulating DNA Purification Kit” from NORGEN. All cfDNA was isolated in the same purification run to minimize “processing bias”.

Importantly, resulting cfDNA concentrations (total cfDNA; spectrophotometry) did not differ between the two protocols (Wilcoxon matched-pairs signed rank test: P= 0.6396 – see below. Matching colors represent same animal samples that have been processed by the two described protocols). These new results have been included in the revised manuscript as Supplementary Information Figure 1r.

Reviewer Reports on the Second Revision:

Referee #1 (Remarks to the Author):

The revised MS is improved and responds well to prior critiques. Overall, this is a very interesting, thoroughly performed study with substantial mechanistic and therapeutic insights.

Referee #4 (Remarks to the Author):

Authors have clarified the questions raised by this reviewer.
No further comments.

Referee #5 (Remarks to the Author):

The authors have made several beneficial changes to the manuscript and supplemental materials. The authors' addition of an overview table describing the exact processing conditions prior to cfDNA isolation is sufficient to address previous concerns regarding cfDNA collection practices. The authors have clarified their usage of the terms "plasma" and "serum" in the methods. Their addition of a table specifying the exact percentages of total cfDNA arising from each tested cell type is helpful (Supplementary Information 1s), as is their comparison of cfDNA plasma concentrations in this experimental stroke model using slightly different processing protocols (Supplementary Information 1r). A typo was noted on manuscript pg. 14 ("sourse" should be "source"). In summary, the authors have adequately addressed the prior concerns regarding cfDNA handling and potential overstatement of the role of neutrophil-derived cfDNA in this mechanism.

Author Rebuttals to Second Revision:

Response to the reviewer

Referee #5:

[...]A typo was noted on manuscript pg. 14 (“sourse” should be “source”).

Thank you for your attentive reading of the manuscript. We corrected the Typo to “source”.